# Synergy processing of diverse ground-based remote sensing and in situ data using GRASP algorithm: applications to radiometer, lidar and radiosonde observations

Anton Lopatin[1], Oleg Dubovik[2], David Fuertes[1], Georgiy Stenchikov[3], Tatyana Lapyonok[2], Igor Veselovskii[4], Frank G. Wienhold[5], Illia Shevchenko[3], Qiaoyun Hu[2] and Sagar Parajuli[3]

[1]GRASP SAS, Villeneuve d'Ascq, 59650, France
[2]Laboratoire d'Optique Atmosphérique, CNRS/Université de Lille, Villeneuve d'Ascq, 59650, France
[3]King Abdullah University of Science and Technology, Thuwal, 23955-6900, Kingdom of Saudi Arabia
[4]Prokhorov General Physics Institute of the Russian Academy of Sciences, Moscow, 117942, Russia
[5]Eidgenössische Technische Hochschule, Zürich, 8092, Switzerland

*Correspondence to*: Anton Lopatin (anton.lopatin@grasp-sas.com) and Oleg Dubovik (oleg.dubovik@univ-lille.fr)

**Abstract.** The exploration of aerosol retrieval synergies from diverse combinations of ground-based passive sun-photometric measurements with co-located active lidar ground-based and radiosonde observations using versatile GRASP algorithm is presented. Several potentially fruitful aspects of observation synergy were considered.

First, a set of passive and active ground-based observations collected during both day and night time were inverted simultaneously under the assumption of temporal continuity of aerosol properties. Such approach explores the complementarity of the information in different observations and results in a robust and consistent processing of all observations. For example, the interpretation of the night-time active observations usually suffers from the lack of information about aerosol particles sizes, shapes and complex refractive index. In the realized synergy retrievals, the information propagating from the close-by sun-photometric observations provides sufficient constraints for reliable interpretation of both day- and night- time lidar observations.

Second, the synergetic processing of such complementary observations with enhanced information content allows for optimizing the aerosol model used in the retrieval. Specifically, the external mixture of several aerosol components with predetermined sizes, shapes and composition has been identified as an efficient approach for achieving reliable retrieval of aerosol properties in several situations. This approach allows for achieving consistent and accurate aerosol retrievals from processing stand-alone advanced lidar observations with reduced information content about aerosol columnar properties.

Third, the potential of synergy processing of the ground-based sun–photometric and lidar observations, with the in situ backscatter sonde measurements was explored using the data from KAUST.15 and KAUST.16 field campaigns held at King Abdullah University of Science and Technology (KAUST) in the August of 2015 and 2016. The inclusion of radiosonde data has been demonstrated to provide significant additional constraints to validate and improve the accuracy and scope of aerosol profiling.

The results of all retrieval set-ups used for processing both synergy and stand-alone observation data sets are discussed and inter-compared.

## 1 Introduction

The ground-based remote sensing is widely recognized as a valuable source of information about the details of the optical properties of ambient atmospheric aerosols (e.g. IPCC, 2013). The passive ground-based remote sensing may include the spectral observations of the direct Sun radiation as well as the multi-angular polarimetric spectral observations of diffuse Sun radiation transmitted through the atmosphere. Such observations have significant sensitivity to the atmospheric aerosol amount, its particles size, shape and morphology; however, they have practically no sensitivity to the vertical variability of aerosols.

The active lidar observation techniques on the other hand are usually used to obtain the information about vertical distribution of aerosol. Lidars emit a series of electromagnetic pulses and register the returned responses from the atmosphere. These responses registered as a function of the time delay are sensitive to the amount and properties of the aerosol at different atmospheric layers. At the same time, compared to passive observations, lidars have notably lower information content with respect to the detailed properties of aerosols such as particle sizes and composition. The most popular lidar systems measure attenuated elastic backscattering registered at the same wavelengths as emitted radiation. Such measurements are affected by variation of all aerosol properties including concentration, sizes and shapes distribution as well as particle composition. Quantitative interpretation of such data is challenging and requires significant a priori information about the aerosol properties (see for e.g. Klett, 1981). The more advanced systems with polarization capabilities emit the polarized light beams and register the state of polarization of the returned signal in addition to the intensity. The obtained depolarization measurement provides the information about the shape of aerosol particles. Additionally, some lidar systems are designed to use the non-elastic scattering, when laser beams trigger radiation emission by certain gases at different atmospheric layers (Wandinger, 2005). Such systems, together with the backscatter signal can directly measure the attenuation of the atmosphere that has direct sensitivity to the amount of aerosol below the level of induced emission. The above-mentioned and other more complex systems (for e.g., High Spectral Resolution Lidar (HSRL), Hair et al., 2008) help to increase significantly the information content of lidar observations about the properties of aerosols. Nonetheless, even the most recent and advanced lidar systems have generally inferior information content about the details of aerosol properties compared to passive multi-angular observations. Indeed, lidar systems usually use only few spectral channels (usually 1 to 5) and can register intensity and state of polarization of reflected signals amounting into generally less than 8 independent measurements even for the most advanced lidar systems. Additionally, the lidar measurements have some other limitations. For example, ground based lidar observations have a blind zone next to the ground due to incomplete geometrical overlap of a laser beam and telescope field of view (Freudenthaler et al., 2018) ranging from several hundred meters to several kilometers depending on the design and purpose of the system. Also, the signals measured by lidar are rather weak with strong distance dependence and lidar measurements suffer from significant registration noises especially during day time observations, limiting capabilities of inelastic (or so-called Raman) observations in the daylight. Therefore, information from co-located photometric measurements is always desirable for the interpretation of lidar observations and the complementarity of the passive and active measurement remains important even if the advanced lidar systems are used. There are many suggestions for joint processing of coincident photometric and lidar ground-based observations, which provide complementary information. For example, recently proposed LiRIC (Chaikovsky, et al., 2016) and GARRLiC/GRASP (Lopatin et al., 2013) algorithms use the joint data from a multi-wavelength lidar and an AERONET Sun/sky-scanning radiometer to derive vertical profiles of fine and coarse aerosol components as well as extra parameters of the column-integrated properties of aerosols. However, it should be noted that in order to maximize the community benefits from synergy, such retrievals should be efficient for processing data collected within the observational networks.

Indeed, the ground-based observations are often collected within a framework of extensive networks. Since the ground-based measurements have local characteristics, conducting such measurements using the network of similar instrumentation allows the generation of regionally and even globally representative data sets helpful for various climate studies, validation of satellite observations and other aerosol related research. The AERONET (Holben et al., 1998), Sky-Net (Nakajima et al., 2020) and SONET (Li et al., 2018) networks of Sun/sky-scanning radiometers are the most visible examples of the global networks of ground-based photometric observations. Similarly, the MPLNet (Welton et al., 2001) and EARLINET (Pappalardo et al., 2014) are the examples of global and regional European lidar networks. In general, the photometric instrumentation is well adapted for automated and even autonomous data collection and, therefore, the operational networks of ground-based photometers are rather extensive. At present, AERONET and Sky-Net have globally more than 600 and 200 sites respectively with SONET (http://www.sonet.ac.cn/index.php) network still in active deployment phase. The lidar systems are more complex in

development and substantially more demanding in operation, correspondingly, lidar networks, as a rule, have a significantly smaller number of sites, e.g., MPLNet has 70 sites globally (although only 20 are active at the moment) and EARLINET has about 27 active sites in Europe.

The complementarity of photometric and lidar data is well recognized by the research community and the creation of joint observations sites where both the photometric and lidar observations are available is highly encouraged, often by upgrading a photometric site with lidar instrumentation. For example, MPLNet sites are always co-located with AERONET sites. In these regards, European ACTRIS (Aerosols, Clouds, and Trace gases Research Infrastructure Network) infrastructure (https://www.actris.eu/Home.aspx) can be mentioned as one of the best examples of networks emphasizing the acquisition of diverse complimentary observations at each site. Specifically, all ACTRIS observational super-sites possess, not only both photometric and complex multi-wavelength lidar systems but also additional in situ data of different kinds.

It should be noted that due to the complexity of lidar systems, especially of advanced multi-wavelength systems, the unification of both lidar observations and their processing is very challenging. For example, EARLINET includes very different lidar systems with different data processing and different customized aerosol retrieval approaches. In these regards, there is significant progress in unification of lidar data processing within ACTRIS lidar network, even though de facto lidar systems remain different. In contrast, the unification of observations and subsequent processing is significantly more advanced for the photometric networks. For example, in the frameworks of AERONET, SkyNet, SONET and CARSNET (Che et al., 2019) the observations are obtained using the same instrumentation following the same observational protocol, while processing is centralized and implemented employing the same retrieval algorithm. Moreover, the observational set up used by the photometric networks seems satisfactory for the aerosol community and there are few rather limited efforts to improve it (Giles et al., 2017). Indeed, as shown in numerous studies the main aerosol properties including aerosol size distribution, complex refractive index, and information about particle shape can be successfully retrieved from the spectral direct Sun and sky-scanning ground-based observations of atmospheric radiation (e.g. Dubovik and King, 2000; Dubovik et al., 2000, 2006; Torres et al., 2014; Sinuyk et al., 2020; Nakajima et al., 2020). While it was shown that addition of polarized sky-scanning observations could provide some improvements in the retrieval accuracy of aerosol fine particles size distribution and refractive index (e.g., see Li et al., 2018; Fedarenka et al., 2016), due to the high complexity of polarimetric observations, such measurements are employed operationally only in SONET (Li et. al., 2018) network (Dubovik et al., 2019). Correspondingly, one of the main challenges of implementing synergy retrievals based on co-incident radiometric and lidar data is achieving a sufficient flexibility of the retrieval in using different lidar observations and assuring their adequate and consistent fusion with the passive measurements. One of rather successful examples of such retrieval tool is GARRLiC (Generalised Aerosol Retrieval form Radiometer and Lidar Combination) algorithm developed by Lopatin et al. (2013) integrated into the generalized approach by Dubovik et al. (2011) that is now named as GRASP (Generalized Retrieval of Atmosphere and Surface Properties, (Dubovik et al., 2014)). GARRLiC/GRASP inverts both the photometric and lidar observations and is currently being employed for operational processing of such combined data within the framework of the European ACTRIS infrastructure (https://www.actris.eu/Home.aspx). However, the original GARRLiC algorithm was developed for the application to the specific observational set of multi-wavelength elastic scattering lidar together with AERONET-like sun/sky-radiometer observations, and did not include the possibilities of utilizing other types of lidar observations (e.g., depolarization and non-elastic scattering).

This paper discusses the evolution of GARRLiC/GRASP approach and demonstrates a wide spectrum of the possibilities for realizing the processing of ground-based observations. Specifically, the current version of GRASP is useful not only for a synergetic retrieval using diverse radiometric and lidar observations but also for a stand-alone instrument processing. To be precise, the present version of GRASP allows new possibilities for inversion of lidar-only observations. The inversion of radiometer-only data is an inherent feature of GRASP (e.g. see Lopatin et al., 2013; Fedarenka et al., 2016; Torres et al., 2017) since it has evolved from AERONET retrieval developments (Dubovik et al., 2011, 2014). Moreover, the GRASP approach

allows for combining the remote sensing data with co-incident in situ observations. For example, this paper demonstrates the potential of synergy processing of the ground-based remote sensing observations together with advanced lidar or backscatter sonde data (that can be considered as a certain in situ analogue of lidar back-scattering measurements). Specifically, the data from SHADOW-1/SHADOW-2 and KAUST.15/KAUST.16 field campaigns held at Institut de Recherche pour le Development (IRD) at Dakar and King Abdullah University of Science and Technology (KAUST) in 2015–2016 were comprehensively analyzed using GRASP approach. Finally, the paper shows benefits of using the multi-pixel approach that has been introduced in GRASP for improving reliability of satellite data processing (Dubovik et al., 2011). This approach uses a priori knowledge of limited time or spatial variability of the parameters retrieved from coordinated but not fully co-incident and/or simultaneous observations. For example, it is used in processing of satellite observations where observations of a large group of different satellite pixels are inverted simultaneously. In this study, it is demonstrated below that this principle can be rather efficient for combining no-coincident but close in time observations, e.g. day- and night- ground-based measurements. The explanation of necessary methodological details is provided in Section 2, numerical tests and applications of the concept to real data are provided and discussed in Section 3.

## 2 Application of GRASP concept to lidar data and their combination with photometric observations

GRASP is a highly versatile algorithm that is developed based on very general principles of numerical inversion and atmospheric radiation modelling which allows utilization of the same algorithm in diverse applications, including processing of passive and active remote sensing observations from ground, space and aircraft including in situ measurements. One of the several objectives behind the development of such a generalized approach is a possibility of straightforward transfer of fruitful retrieval ideas identified in one area of applications to other domains. For example, the development of GRASP algorithm allowed the adaptation of several ideas proven to be useful in aerosol retrievals from AERONET observations (see description in Dubovik and King, 2000; Dubovik et al., 2000, 2006) to enhance retrieval of aerosol properties from satellite observations (see Dubovik et al., 2011). Lopatin et al. (2013) extended the application of GRASP concept for inversion of combined lidar and radiometric ground-based observations. Romain et al. (2017) illustrated the application of the algorithm for the interpretation of ground-based sky-camera observations. Torres et al. (2017) demonstrated the high potential of GRASP retrieval concept for inverting only direct Sun photometric observations. Espinosa et al. (2017, 2019) and Schuster et al. (2019) used GRASP approach for processing in situ aircraft and laboratory light scattering measurements. Most of these studies benefited from previous GRASP implementations and included certain new elements needed in specific applications. In these regards, in the description below the paper will focus on new elements developed for interpretation of ground-based active and passive observations, as well as interesting adaptations of previously developed concepts to these applications.

In this section, improvements accumulated during GRASP code development and methodological base that are crucial for the presented study are discussed. Other details on GRASP operational principles and application to different observation types could be found in (Dubovik et al., 2000, 2006, 2011 and Lopatin et al., 2013).

### 2.1 Modelling of aerosol optical properties

For applications to ground-based passive and active observations the aerosol in GRASP is usually modelled as external mixture of $K$ aerosol components:

$$\tau_{scat/ext}(\lambda; h) = \sum_{k=1}^{K} \int_{h_{min}}^{h_{max}} \int_{\ln \varepsilon_{min}}^{\ln \varepsilon_{max}} \int_{\ln r_{min}}^{\ln r_{max}} \frac{C_{scat/ext}(m_k(\lambda); h; \varepsilon; r)}{v(r)} \frac{dV_k(h)}{dh} \frac{dN_k(\varepsilon)}{d\ln\varepsilon} \frac{dV_k(r)}{d\ln r} \, d\ln r \, d\ln\varepsilon \, dh \quad (1)$$

and

$$\tau_{scat} P_{ij}(\lambda; \Theta; h) = \sum_{k=1}^{K} \int_{h_{min}}^{h_{max}} \int_{\ln \varepsilon_{min}}^{\ln \varepsilon_{max}} \int_{\ln r_{min}}^{\ln r_{max}} \frac{C_{ij}(m_k(\lambda); h; \varepsilon; r; \Theta)}{v(r)} \frac{dV_k(h)}{dh} \frac{dN_k(\varepsilon)}{d\ln\varepsilon} \frac{dV_k(r)}{d\ln r} \, d\ln r \, d\ln\varepsilon \, dh, \quad (2)$$

where $\lambda$ denotes wavelength, $\Theta$ – scattering angle, $h$ - altitude of the layer, $\varepsilon$ - axis ratios of spheroid, and $r$ denotes the radius of volume equivalent sphere, $v(r)$ is the volume of particle with radius $r$ and $C_{scat/ext}(m_k(\lambda); h; \varepsilon; r)$, $C_{ij}(m_k(\lambda); h; \varepsilon; r; \Theta)$ are cross-sections of scattering, extinction and directional scattering corresponding to matrix elements $P_{ij}(\Theta)$ of aerosol particle. Each of k-th aerosol components may have different size distribution $\frac{dV_k(r)}{d\ln r}$, shape distribution $\frac{dN_k(\varepsilon)}{d\ln \varepsilon}$, spectral complex index of refraction, $m_k(\lambda) = n_k(\lambda) - i\kappa_k(\lambda)$ and vertical profile $\frac{dV_k(h)}{dh}$.

Thus, aerosol properties in GRASP are retrieved in the form of size and shape distributions, vertical profile and spectrally dependent complex refractive index for $K$ components. In principle, all these characteristics are continuous functions that in actual retrieval are represented by a set of discrete parameters. For example, one of the most general representation of size distribution is a superposition of several base functions:

$$\frac{dV_k(r)}{d\ln r} = \sum_{i=1}^{N_r} c_i^k v_i(r), \tag{3}$$

where $v_i(r)$ are fixed functions (so-called "bins") and $c_i^k$ are the weights of corresponding bins that are retrieved. For example, in GRASP $v_i(r)$ can be represented by the rectangular or tri-angular functions centered in $N_r$ nodal points: $\ln r_{i+1} = \ln r_i + \Delta \ln r$ (e.g. see Dubovik et al., 2006) or by $N_r$ log-normal functions (e.g. see Dubovik et al., 2011). Similar approximations are used for shape distribution and vertical profile. Correspondingly, in total $N_{par} = K * (N_r + N_\varepsilon + N_h)$ parameters are retrieved to characterize size, shape and vertical distributions of these $K$ components. When such functional representations are employed, the size distribution are retrieved in absolute scale, while the shape distributions and vertical profiles are retrieved in relative scale using the following normalizations:

$$\int_{\ln \varepsilon_{min}}^{\ln \varepsilon_{max}} \frac{dN_k(\varepsilon)}{d\ln \varepsilon} = 1 \quad \text{and} \quad \int_{h_{BOA}}^{h_{TOA}} \frac{dV_k(h)}{dh} = 1. \tag{4}$$

In practice, if the number $N_{par}$ of sought parameters is large, the reliable retrieval of all parameters is challenging due to the limitations of information content. Therefore, the number of bins, and even chosen functional form of the characteristics, can be varied in different situations. For example, in AERONET retrieval, with rather high information content in respect to the size distribution, 22 size bins are used (Dubovik and King, 2000). In satellite retrievals, the information content of reflected radiation is lower and normally a smaller number of parameters is retrieved. For example, 16 size triangular bins were used in the initial considerations for PARASOL data inversion using GRASP algorithm (Dubovik et al., 2011), which later were reduced to only 5 log-normal bins in PARASOL/GRASP operational processing (e.g., Chen et al., 2020). In GARRLiC/GRASP two aerosol components (fine and coarse) are retrieved using 10 and 15 triangular bins for fine and coarse particle size distributions and $N_h$=60 (for each fine and coarse component) rectangular bins for vertical profiles from the combination of lidar and radiometric data (Lopatin et al., 2013).

For the shape distribution, the superposition of up to $N_\varepsilon = 13$ rectangular bins can be used in GRASP the inversions (e.g., in the processing of full phase functions by Dubovik et al. (2006) or by Espinosa et al. (2017, 2019)). However, the sensitivity of light scattering and especially remote sensing observations is rather limited to the particle shape. Therefore, in many applications, a function with very limited number of parameters is used to approximate the shape distribution. For example, in AERONET retrieval, POLDER inversions, and GARRLiC/GRASP, the shape distribution is represented by two components of purely spherical and non-spherical particles with assumed shape distribution as described in details by Dubovik et al. (2006). Taking into account the normalization to unity in Eq. (4), the Eq. (1) can be rewritten for columnar properties for each aerosol component as:

$$\tau_{scat/ext}^k(\lambda) = \sum_{k=1}^{K} \int_{\ln r_{min}}^{\ln r_{max}} \left( A_{sph} \frac{C_{scat/ext}^{sph}(\ldots; r)}{v(r)} + \left(1 - A_{sph}\right) \frac{C_{scat/ext}^{nons}(\ldots; r)}{v(r)} \right) \frac{dV_k(r)}{d\ln r} d\ln r$$

and                                                                                                                                    (5)

$$\tau_{scat/ext}^k(\lambda)P_{ij}^k(\lambda;\Theta) = \sum_{k=1}^{K} \int_{\ln r_{min}}^{\ln r_{max}} \left( A_{sph} \frac{C_{ij}^{sph}(\dots;r)}{v(r)} + (1 - A_{sph}) \frac{C_{ij}^{nons}(\dots;r)}{v(r)} \right) \frac{dV_k(r)}{d\ln r} d\ln r$$

Correspondingly, the aerosol scattering properties in different atmospheric layers are modeled as:

$$\tau_{scat/ext}^k(\lambda;h) = \tau_{scat/ext}^k(\lambda) \frac{dV_k(h)}{dh}$$

and                                                                                                                                    (6)

$$\tau_{scat/ext}^k(\lambda;\Theta;h)P_{ij}^k(\lambda;\Theta;h) = \tau_{scat/ext}^k(\lambda)P_{ij}^k(\lambda;\Theta) \frac{dV_k(h)}{dh}$$

The number of retrieved parameters for size, shape and vertical distributions can be also decreased using other functional approximations than Eq. (3). For example, size distribution in GRASP can be represented by a bi-modal log-normal distribution with the parameters of these log-normal distributions being retrieved (Dubovik et al., 2011; Torres et al., 2017). Similarly, for the passive ground-based or satellite observations that do not have sufficient sensitivity to the detailed vertical profile, a simple functional approximation for vertical profile like exponential or normal distributions are used in GRASP retrievals (e.g., see Torres et al., 2014; Dubovik et al., 2011).

In addition to particle size, shape and vertical distribution, the spectral complex index of refraction, $n_k(\lambda); \kappa_k(\lambda)$ is retrieved in many GRASP applications. As a rule, the values of complex refractive index $n(\lambda_i)$ and $\kappa(\lambda_i)$ are retrieved directly at the wavelengths $\lambda_i$ of the available measurements following the AERONET retrieval approach by Dubovik and King (2000), where complex refractive index was retrieved at each wavelength of the sky radiance observation. For example, such approach is used in GARRLiC/GRASP inversion of combined sky radiometer and lidar data, GRASP inversion of Nephelometer measurements (Espinosa et al., 2017, 2019), and satellite or air borne measurement of multi-angular polarimeter (Dubovik et al., 2011; Chen et al., 2020; Puthukkudy et al., 2020; etc.). Another possibility realized in GRASP is a utilization of modelling of n(λ) and κ(λ) by assuming aerosol as a mixture of several (K) components, i.e.,

$$n(\lambda) = \sum_{i=1}^{K} c_i n_i(\lambda) \text{ and } \kappa(\lambda) = \sum_{i=1}^{K} c_i \kappa_i(\lambda),$$                                    (7)

or

$$n(\lambda) = n_{mix}[c_1; c_2; \dots; n_1(\lambda); n_2(\lambda); \dots] \text{ and } \kappa(\lambda) = \kappa_{mix}[c_1; c_2; \dots; \kappa_1(\lambda); \kappa_2(\lambda); \dots],$$     (8)

where Eq. (7) represents a so-called volume mixture and Eq. (8) denotes more complex internal mixture of the components. The Eq. (7) illustrates the simple volume mixture of different components with known dependencies $n_i(\lambda)$ and $\kappa_i(\lambda)$ – the sum weighted by volume fractions $c_i$. The Eq. (8) illustrates a more complex (non-linear) internal mixture of refractive indices of different components. For example, Li et al. (2019, 2020) used the approximations of volume mixture (Eq. (7)) and Maxwell-Garnett internal mixture (Eq. (8)) in the aerosol retrievals from AERONET ground-based radiometers and POLDER/PARASOL satellite observations.

The Eqs. (1-8) generally describe the retrievals where all or several of such aerosol parameters as size, shape, spectral refractive index and vertical distribution are explicitly retrieved. However, in the situations with very limited information content such retrievals could be very challenging or even impossible. In such a situation, adding extra assumptions and reducing the number of the retrieved parameters might be desirable. For example, in the processing of MERIS and PARASOL satellite data by GRASP (Chen, et al., 2020; Dubovik et al., 2021), the detailed aerosol parameters were not retrieved explicitly, instead the aerosol single scattering properties were modelled as an external mixture of several aerosol components and the columnar properties of each component are defined as:

$$\tau_{scat/ext}^k(\lambda) = c_v^k \rho_{scat/ext}^k(\lambda)$$                                                                         (9)

and

$$\tau_{scat}^k(\lambda)P_{ij}^k(\lambda;\Theta) = c_v^k \rho_{scat}^k(\lambda)P_{ij}^k(\lambda;\Theta) \tag{10}$$

where $\rho_{scat/ext}^k(\lambda)$ and $P_{ij}^k(\lambda;\Theta)$ denote the scattering/extinction per unit of volume and phase matrix of each aerosol component that are pre-calculated using complex refractive index, size and shape distributions assumed for each aerosol component.

Correspondingly, only $K$ concentrations $c_v^k$ drive the modeling of columnar properties of aerosol. This approach allows significant reduction in number of the retrieved parameters, which is especially fruitful for the observations with limited sensitivity to the size, shape and refractive index of the aerosol particles. Such multi-component external mixture approach has already been proven to be efficient for MERIS (https://www.grasp-open.com/products/meris-data-release/) and POLDER applications (Chen et al., 2020). In those applications the vertical aerosol distribution was assumed the same for all the components and modeled as exponential function of only one scale height parameter. In contrast, below in Section 3.2, this multi-component model will be considered for application to the lidar and radiosonde data where components are characterized by a separate detailed vertical distribution.

Modelling aerosol as an external mixture of different component is rather common concept used in many remote-sensing and climate modelling application with some modifications (e.g. see Chin et al., 2002; Levy et al., 2007). Generally, the aerosol components are associated with optically distinct types of aerosol based on particle sizes, scattering and absorption capabilities, etc. The defined mixture may be composed from 2 (fine and coarse aerosol compounds) in the simplest case, and up to 12 or more. For example, a certain simulation of GEOS-Chem (Chin et al., 2002) proposes utilizing 5 aerosol classes, with coarse aerosols (dust and sea salt) having several sub types (7 and 2 correspondingly) of different size.

Thus, Eqs. (1–10) demonstrate the methodological concepts used for modelling aerosol single scattering in GRASP algorithm. It should be noted that the approaches discussed in this section for modelling aerosols were already effectively and extensively used in several GRASP applications. At the same time, the structural design of GRASP algorithm allows rather straightforward modifications of aerosol single scattering model, and therefore other approaches can be easily employed depending upon the need of proposed application. In addition, there is a possibility of reducing a number of retrieved parameters by assuming that some of the retrieved characteristics, for e.g., shape, vertical distribution, refractive index, etc., are the same for a set of aerosol components. Such possibility combined with the flexibility of parameter definition described above allows changing the number of parameters retrieved ($N_{par}$) within an impressive range, tailoring the complexity of the retrieval to the informational content provided by the set of observations being used.

For example, as demonstrated by Lopatin et al. (2013) distinguishing between complex refractive indices of fine and coarse modes can be a very challenging task that is feasible only in situations when fine and coarse aerosol components have different origin and are well separated in different vertical layers. In these regards assuming the same complex refractive index and vertical distribution both for fine and coarse components may be more adequate for the situations with well mixed aerosol layers. Such assumption allows a drastic decrease of the number of the parameters retrieved resulting in the improved stability of the solution. It is also evidently useful for situations when information content is limited, i.e. in the case of single wavelength elastic lidar with no polarization capabilities. Similarly, in interpretation of the passive spaceborne observations that have limited sensitivity to vertical variability of the atmosphere, a unique vertical distribution can be used for several aerosol components as it has been done for POLDER (Dubovik et al., 2011; Li et al., 2019; Chen et al., 2020; etc.).

The effects of multiple-scattering in the atmosphere are accounted for in GRASP using the successive order of scattering radiative transfer code (Lenoble et al., 2007) that utilizes the single scattering aerosol properties together with surface bidirectional reflectance distribution function (BRDF) and bidirectional polarization distribution function (BPDF). Additional details of atmospheric radiation calculations implemented in the GRASP forward model can be found in the papers by Dubovik et al. (2011, 2021).

## 2.2 Vertically resolved measurements in GRASP: lidars and airborne instruments

GRASP has been developed as a highly versatile algorithm that can be applied to diverse measurements including the observations of vertical structure of the atmosphere provided primarily by two types of instrumentation: lidar and airborne in situ sensors. The first possibilities of processing elastic ground-based lidar measurements were introduced as the GARRLiC concept by Lopatin et al. (2013). During the last years, the list of vertically resolved observations accepted by GRASP has been significantly expanded and nowadays includes observations of aerosol vertical structure such as extinction, backscatter, normalized elastic and inelastic lidar signals together with volume and particle depolarization. In addition, some other methodological changes for improving processing of vertically resolved observations were introduced.

This section focuses on the presentation of all above mentioned changes and modifications of GRASP. The major driving motivation for these developments was the desire of adapting new observation techniques, many of which have significantly matured and become wide spread over the last decade.

### 2.2.1 Enhanced vertically resolved observations by advanced lidars, airborne instruments and radiosondes

Present efforts on the accurate profiling of atmosphere notably favor the measurements by powerful and sophisticated observation techniques (Comeron et al., 2017) that include High Spectral Resolution Lidars (HSRL) (Hair et al., 2008) and inelastic or Raman lidars (Veselovskii et al., 2015). The possibility to directly obtain vertical profiles of aerosol extinction and backscatter is the main advantage of these instruments compared to elastic lidars. At the same time the information on aerosol scattering properties at different layers of atmosphere can be provided by airborne remote sensing and in situ data. For example, the vertical profiles of aerosol extinction could be provided by in situ airborne sun-photometer measurements of aerosol optical depth at different atmospheric layers (e.g., Karol et al., 2013) and the vertical backscatter profile could be provided by radiosonde observations. Specifically, in this study the data from the Compact Optical Backscatter Aerosol Detector (COBALD, see https://iac.ethz.ch/group/atmospheric-chemistry/research/ballon-soundings.html) are used.

Another example is the airborne nephelometer that can provide the measurements of aerosol total extinction, absorption and scattering, together with angular scattering and degree of linear polarization at different layers (Espinosa et al., 2017, 2019; Schuster et al., 2019). Finally, the elastic lidars emitting polarized signals can detect profiles of the signal depolarization that is a function of the elements $P_{22}$ and $P_{11}$ of the scattering matrix.

Thus, in order to have flexibility for inverting advanced vertical observations the profiles of scattering, extinction, all scattering matrix elements and some of their direct products were included in GRASP interface as possible input/output characteristics. Correspondingly, the GRASP software (https://www.grasp-open.com/products/) starting from version 0.8.1 allows simulation and inversion of the scattering characteristics for each atmospheric layer shown by Eqs. (6,9,10). It should be noted that in GARRLiC/GRASP the aerosol backscatter and extinction profiles were modeled as a part of the retrieval but remained encapsulated deeply in the calculations. In these regards, at present, various characteristics provided from diverse lidar observations can be modeled and processed by GRASP, i.e. could be used as a part of input for inversion and obtained as part of retrieval results output or forward simulations. In addition, several parameters that are simple functions of the scattering characteristics were included into the software input. For example, the elastic lidar measurements are described by the following lidar equation:

$$L(\lambda; h) = A(\lambda)\beta(\lambda; h)exp\left(-2\int_{z_{min}}^{z(h)} \sigma(\lambda; h)\, dz\right), \tag{11}$$

where $\sigma(\lambda; h)$ denote extinction and $\beta(\lambda; h)$ backscatter profiles. The extinction profile includes aerosol, molecular scattering and gaseous absorption components $\sigma(\lambda; h) = \sigma_a + \sigma_m + \sigma_g$ and backscatter includes aerosol and molecular components

$\beta(\lambda; h) = \beta_a + \beta_m$, $A(\lambda)$ is a constant estimated from lidar calibration, $z$ is lidar path related with the atmospheric altitudes of target $h$, ground level $h_{BOA}$ and zenith angle of lidar inclination $\Theta_L$ as $z(h) = \frac{(h-h_{BOA})}{\cos(\Theta_L)}$.

Generally, gaseous absorption and molecular scattering are rather stable and in most of lidar aerosol applications are usually accounted using climatology or ancillary data. The aerosol component of extinction $\sigma_a(\lambda, h)$ and backscatter $\beta_a(\lambda, h)$ profiles can be calculated as:

$$\sigma_a(\lambda; h) = \tau_{ext}^a(\lambda; h) = \sum_{k=1}^{K} \tau_{ext}^k(\lambda)\, v_k(h) \tag{12}$$

and

$$\beta_a(\lambda; h) = \frac{1}{4\pi} \sigma_{scat}^a(\lambda; h) P_{11}^a(\lambda; 180°; h) = \frac{1}{4\pi} \sum_{k=1}^{K} \tau_{scat}^k(\lambda)\, P_{11}^k(\lambda; 180°) v_k(h), \tag{13}$$

where $v_k(h)$ denotes $\frac{dV_k(h)}{dh}$.

In lidar applications, the aerosol backscatter is also often expressed via so-called lidar ratio $S_a(\lambda, h) = \frac{\sigma_a(\lambda,h)}{\beta_a(\lambda,h)}$.

$$\beta_a(\lambda; h) = \frac{\sigma_a(\lambda; h)}{S_a(\lambda; h)} = \frac{\tau_{ext}(\lambda) v_k(h)}{S_a(\lambda; h)}, \tag{14}$$

where the total lidar ratio for aerosol $S_a(\lambda; h)$ is defined as:

$$S_a(\lambda; h) = \frac{4\pi}{P_{11}(\lambda; 180°; h)\omega_0(\lambda)}, \tag{15}$$

where $P_{11}$ could be defined following Eqs. (2), (6) or (10) and $\omega_0(\lambda)$ is the aerosol single scattering albedo.

The lidar ratio is determined by aerosol microphysical composition (size distribution, refractive index, shape, etc.) only and doesn't depend on the amount of aerosol. Therefore, in situations when aerosol microphysics can be considered vertically constant, using a priori assumption about lidar ratio allows deriving a vertical profile of aerosol extinction directly from backscatter profile measurements. Similarly, if the aerosol is represented as an external mixture of K components, the backscatter profile can be expressed via the lidar ratios of its components:

$$\beta_a(\lambda; h) = \frac{\sigma_a(\lambda; h)}{S_a(\lambda; h)} = \sum_{k=1}^{K} \frac{\sigma_a^k(\lambda; h)}{S_a^k(\lambda; h)} = \sum_{k=1}^{K} \frac{\tau_{ext}^k(\lambda; h) v_k(h)}{S_a^k(\lambda; h)}, \tag{16}$$

where the lidar ratio for k-th component of aerosol $S_a^k(\lambda; h)$ is defined similarly as shown in Eq. (14).

Indeed, as discussed above and seen from Eq. (11), the attenuated aerosol backscatter measured by elastic lidars is a function depending on both aerosol backscattering at specific layer and aerosol extinction profile. The ambiguity in separation of the backscattering by the layer and extinction of the lidar signal by underlying layers is considered as the main challenge in the interpretation of elastic lidar signals. As mentioned earlier, the lidar systems using inelastic scattering address that ambiguity by measuring the following signal:

$$L_{nel}(\lambda; h) = A(\lambda; \lambda')\beta_{nel}(\lambda; h) \exp\left(- \int_{z_{min}}^{z(h)} \left(\sigma(\lambda; z) + \sigma(\lambda'; z)\right) dz\right), \tag{17}$$

where $\lambda'$ – wavelength of exciting impulse that triggers inelastic backscatter at wavelength $\lambda$, $\beta_{nel}(\lambda; h)$ is inelastic backscattering of atmosphere, $\sigma(\lambda; z)$ is atmospheric extinction and $A(\lambda; \lambda')$ is a constant estimated from lidar calibration. The shift $\lambda' \to \lambda$ in inelastic backscattering $\beta_{nel}(\lambda; h)$ could be a result of gaseous molecules emission frequency shifts due molecular rotations and vibrations or Rayleigh scattering and can be rather accurately estimated based on known characteristics of emitted lidar impulse and atmospheric gases. Therefore, the $\sigma_a(\lambda; h)$ is the only fully unknown characteristic in Eq. (17) and can be obtained from $L_{nel}(\lambda; h)$ by rather straightforward transformations.

It should be noted that the measurements of such advanced lidar systems as HSRL usually are converted to the measured backscatter and extinction profiles (Hair et al., 2008; Rogers et al., 2009, etc.) that can be used as an input to GRASP algorithm for conducting full aerosol retrieval (following Eqs. (12–16)).

Another characteristic measured by advanced lidars with polarimetric capabilities is the profile of volume and aerosol particle depolarization. The profile of particle depolarization can be estimated from the lidar returns of emitted polarized light beams following (Freudenthaler et al., 2009):

$$L_{\perp,\parallel}(\lambda; h) = A_{\perp,\parallel}(\lambda)\beta_{\perp,\parallel}(\lambda; h)exp\left(-2\int_{z_{min}}^{z(h)} \sigma(\lambda; h)\, dz\right), \tag{18}$$

where the subscripts "$\perp$" and "$\parallel$" indicate cross- and co-polarized components correspondingly. The atmospheric volume depolarization ratio can be estimated as:

$$\delta_v = \frac{L_\perp(\lambda; h)}{L_\parallel(\lambda; h)} = \frac{\beta_\perp(\lambda; h)}{\beta_\parallel(\lambda; h)}, \tag{19}$$

where the simple assumption of $A_\perp = A_\parallel$ has been used. The depolarization ratio from the atmospheric layers can be estimated via phase matrix elements as follows:

$$\delta_v = \frac{\beta_\perp(\lambda; h)}{\beta_\parallel(\lambda; h)} = \frac{I_\perp(\lambda, 180°; h) - Q_\perp(\lambda, 180°; h)}{I_\parallel(\lambda, 180°; h) + Q_\parallel(\lambda, 180°; h)} = \frac{P_{11}(\lambda, 180°; h) - P_{22}(\lambda, 180°; h)}{P_{11}(\lambda, 180°; h) + P_{22}(\lambda, 180°; h)}, \tag{20}$$

where the following relationships were used $I_\perp = P_{11} + P_{22}$, $Q_\perp = P_{12} + P_{22}$, $I_\parallel = P_{11} - P_{22}$, $Q_\parallel = P_{12} - P_{22}$, as well as assumption that $P_{12}(\lambda, 180°; h) = 0$.

The volume depolarization of light is a result of both aerosol and molecular scattering effects:

$$\delta_v = \frac{\sigma_{scat}^a P_{11}^a + \sigma^m P_{11}^m - (\sigma_{scat}^a P_{22}^a + \sigma^m P_{22}^m)}{\sigma_{scat}^a P_{11}^a + \sigma^m P_{11}^m + \sigma_{scat}^a P_{22}^a + \sigma^m P_{22}^m}, \tag{21}$$

where the simplified notations $\sigma_{scat}^{a,m}(\lambda; h) = \sigma_{scat}^{a,m}$ and $P_{ij}^{a,m}(\lambda; h) = P_{ij}^{a,m}$ were used. The molecular scattering properties including molecular backscatter $\beta_m$ and depolarization ratio $\delta_m$ are rather stable and well known. Therefore, in many lidar applications, depolarization ratio of aerosol $\delta_a$ is derived from lidar measurement, using $\beta_m$ and $\delta_m$, and provided for further interpretation. Specifically, using the identity $P_{22} = P_{11}(1 - \delta)/(1 + \delta)$ and definition $\beta = \sigma_{scat} P_{11}(180°)/4\pi$, Eq. (21) can be transformed as:

$$\delta_v = \frac{\beta_a + \beta_m - \beta_a(1 - \delta_a)/(1 + \delta_a) - \beta_m(1 - \delta_m)/(1 + \delta_m)}{\beta_a + \beta_m + \beta_a(1 - \delta_a)/(1 + \delta_a) + \beta_m(1 - \delta_m)/(1 + \delta_m)} =$$
$$\frac{\beta_a \delta_a + \beta_m \delta_m + (\beta_a - \beta_m)\delta_a \delta_m}{\beta_a + \beta_m + \beta_a \delta_a + \beta_m \delta_a} = \frac{R\delta_a(\delta_m + 1) - \delta_a + \delta_m}{R(\delta_m + 1) + \delta_a - \delta_m}, \tag{22}$$

where $R$ denotes so-called backscattering ratio $R(\lambda; h) = \frac{\beta_a(\lambda;h) + \beta_m(\lambda;h)}{\beta_m(\lambda;h)}$ that can be estimated directly from lidar measurements using known atmospheric density profile. In most of practical lidar applications, the particle depolarization is derived directly from observations and considered as one of principal characteristics for further analysis. At the same time, the derivation of particle depolarization from observed volume depolarization profiles relies on rather scrupulous data selection and require the knowledge of backscattering ratio. In these regards, direct inversion of the volume depolarization is an interesting alternative because comprehensive forward model utilized in the retrieval simulates depolarization using almost no or only very general assumptions. The examples of using volume depolarization in GRASP retrieval can be found in the earlier paper by Hu et al. (2019).

It should be emphasized that the observation of depolarization is a very powerful tool for detecting the presence of the non-spherical or irregular shaped particles such as desert dust aerosols or crystal clouds. Indeed, the change of the polarization state of the emitted light observed in the signal returned from different aerosol layers provides fundamentally new information about aerosol properties that is unavailable from other observation techniques. For example, none of the advanced polarimetric

passive instruments can be compared in sensitivity to the shape of aerosol or cloud particles with the lidar measurement of depolarization (Dubovik et al., 2019). Although polarimetric lidar observations remain significantly more challenging than conventional intensity inelastic observations, the recent developments and improvements have made depolarization observations widely available in many advanced lidar systems as those employed by MPL and ACTRIS networks (Pappalardo et al., 2014).

Thus, as a result of refining the GRASP forward model capabilities, diverse vertically resolved atmospheric characteristics can be used at present as the input of GRASP. Specifically, in addition to inversion of inelastic lidar observations described by Lopatin et al. (2013), last version of GRASP can invert inelastic lidar observations and depolarization as well as profiles of extinction and diverse single scattering characteristics.

In addition to adaptation of advanced vertically resolved measurements, several convenient modifications in handling actual lidar measurements were realized. For example, processing of lidar signal in GARRLiC/GRASP (Lopatin et al., 2013) relied on the conventional technique that estimates the lidar constants ($A(\lambda)$) using signal at some predefined or manually selected reference altitude with presumably negligible aerosol presence. The value of the signal at this altitude is used to normalize the attenuated backscatter profile in order to exclude a hardware dependent coefficient ($A(\lambda)$) present in lidar equations (see Eqs. (11), (17), (18)), and therefore to calibrate the profiles. This approach was adapted from earlier synergy retrieval Lidar and Radiometer Inversion Code (LiRIC) by Chaikovsky et al. (2016) that also processed combined radiometric and lidar data. However, LiRIC used the results of inversions from ground-based radiometers to constrain stand-alone lidar retrievals. In these regards, GARRLiC/GRASP proposed a more profound synergy approach by inverting joint lidar and photometric data set and simultaneously retrieving both columnar and vertical aerosol properties. This concept of the joint fitting allows for using an approach for addressing calibration uncertainties denoted by the constant $A(\lambda)$ in the lidar equations that is simpler and more straightforward compared to the conventional procedures used by Lopatin et al. (2013), Chaikovsky et al. (2016), etc. Specifically, the robust calibration of both elastic and inelastic lidar signals could be performed using the following normalization:

$$L^*_{norm}(\lambda, z) = \frac{L^*(\lambda, z)}{\int_{z_{min}}^{z_{max}} L^*(\lambda, z) dz}, \tag{23}$$

where $z_{min}$ and $z_{max}$ are minimum and maximum lidar observation distances, respectively, and $L^*$ denotes observed lidar signal that can be either elastic or inelastic. This normalization approach is used in the current version of GRASP. It excludes an operation of manual selection of the reference points from the lidar data treatment. The realization and application of the approach is described in earlier work by Bovchalyuk et al. (2016). This normalization not only eliminates the possible biases in the calibrated signals that could be introduced due to the incorrect selection of reference altitude, but also opens possibilities for adequate and simple lidar data processing on a significantly larger scale of signal variability. Indeed, correct selection of reference altitude, which in many ways depends upon the experience of the lidar operator and his/her ability to detect the presence of aerosol at higher atmospheric layers, makes centralized operational processing of the data coming from different sites and instruments a challenging and time-consuming task.

It should be noted that this paper is focused on the utilization of only ground-based vertically resolved observations. At the same time, modeling satellite or airborne vertically resolved lidar observations is rather similar and also has been implemented in GRASP and been used for feasibility analysis and selected applications to real data (e.g. Espinosa et al., 2020). The detailed discussion of GRASP applicability to active satellite observation will be provided in a separate publication in the future.

**2.3 Numerical inversion and retrieval constraints**

The numerical inversion in GRASP relies on the so-called multi-term least squares method (LSM), that has been introduced in previous papers (Dubovik, 2004; Dubovik and King, 2000; Dubovik et al., 2011; etc.). The details of numerical inversion implementation can be found in the papers by Dubovik et al. (2011, 2021). The strength of multi-term LSM approach is a

rather transparent methodology that allows an inversion of various observation data using multiple a priori constraints. Namely, several smoothness constraints can be used to retrieve continuous unknown characteristics together with direct a priori estimates for any set of parameters. For example, in AERONET retrieval, independent smoothness constraints were applied for retrieved aerosol size distributions, spectral dependence of complex index of refraction (e.g. see Dubovik and King, 2000; Dubovik et al., 2000) and particle shape distribution (Dubovik et al., 2006). In inversion of satellite observations, smoothness

constraints were also used for spectral dependencies of simultaneously retrieved parameters of surface BRDF and BPDF (see Dubovik et al., 2011). Lopatin et al. (2013) additionally applied smoothness constraints on the retrieved vertical profiles in GARRLiC/GRASP simultaneous inversions of co-located lidar and Sun/sky-radiometer data. Direct a priori constants are utilized in AERONET-like retrievals for the concentrations of particles at extremes of size distributions (for the smallest and largest size bins). Torres et al. (2017) used direct a priori estimates for the refractive index in GRASP inversion of spectral

AOD measurements.

In addition, an advanced feature of multi-pixel inversion has been introduced by Dubovik et al. (2011) and realized in GRASP algorithm. This concept allows benefiting from a priori knowledge about spatial or temporal variability of any of the retrieved parameters when a group of coordinated observations is inverted. For example, in inversion of large groups of POLDER image pixels, the application of a priori limitation on temporal variability of land reflectance and spatial variability

of aerosol parameters have been proven to be very useful for improving accuracy of the aerosol and surface retrievals (see Dubovik et al., 2011; Chen et al., 2020; etc.). Although the multi-pixel retrieval concept was initially introduced for satellite observation, it will be shown below that it can also be efficiently used for improving the retrieval from ground-based observations. This is specifically beneficial when co-located but not coincident lidar and radiometric observations are processed simultaneously. The details of application of the concept will be discussed in Section 3.1.

**3 Advanced applications of GRASP algorithm for interpretation of vertically resolved observations**

This section demonstrates the enhanced capacities of GRASP to process the vertically resolved ground-based observations. The focus of the demonstrations will be on the outlining of novelties of GRASP in comparisons with earlier GARRLiC/GRASP approach introduced by Lopatin et al. (2013): (i) new possibilities for synergy processing of combined radiometric and vertically resolved observations and (ii) recently introduced option of single instrument processing of vertically resolved

observations. Specifically, the following three aspects will be considered:

1. Utilization of observations by advanced lidar systems and airborne backscatter sonde;
2. Application of multi-pixel retrieval concept to the multi-instrument observations;
3. Realization of stand-alone instrument retrievals using vertically resolved observations by diverse lidars and

backscatter sonde.

Lopatin et al. (2013) have proposed a new synergy approach for enhancing retrieval by using simultaneous complementary radiometer and lidar data. The GRASP updates discussed here allow the simultaneous synergy inversions of a much wider variety of complementary observations if they are available. In these regards, many extensive field campaigns held in recent

years were focused on performing observations from a wide range of available measurement techniques and on efforts designed to guarantee high quality and continuity of observations. This paper will focus on four datasets provided by such campaigns. Thus, measurements by sun/sky radiometer, backscatter sonde and lidars with advanced capabilities such as polarization or multi-wavelength registration of inelastic backscatter will be used. The details of the measurements from each dataset used in the study are provided below.

**Sun-Sky photometer**

A Cimel Electronique 318 sun-photometer is used as a standard instrument in the AErosol Robotic NETwork (Holben et al., 1998) that provides accurate information about detailed columnar properties of aerosols in over 500 sites around the globe. Regular calibration procedures are employed within the network; the deployed sun-photometers provide aerosol optical thickness with the accuracy of 0.01 and sky-scanned radiances with the accuracy of 5% at a number of wavelengths covering at least visible and near infrared spectrum range, notably 440, 670, 870 and 1020 nm. All instruments operating within AERONET perform daily a pre-programmed measurements sequence that consists of a series of direct sun and sky radiance measurements at fixed solar elevations (almucantars) or azimuth angles (principal plane) during the day. Direct sun measurements are performed every 15 minutes and sky radiances are acquired almost every hour both for almucantar and principal plane configurations.

**Advanced lidars**

Most lidars measuring elastic backscatter observations use Nd:YAG laser, which provides measurement at 532 nm in case of single-wavelength instruments. The multi-wavelengths models provide measurements in additional 355 and 1064 nm channels (for e.g. Comeron et al., 2017). The lidar systems with polarimetric capabilities have an additional channel with a polarizer in front of the detector and provide depolarization ratio at one or several wavelengths. Raman lidars are additionally equipped with one or two channels that register inelastic backscattering signal from vibrational Raman scattering at 387 and 607 nm. Power of Raman backscatter can be increased, by using a group of nitrogen and oxygen rotational lines at 530 nm (Veselovskii et al., 2015). All lidars provide observations within a certain distance range, which varies from instrument to instrument and is limited by emitter/receiver field of view overlap in the lower part as well as by the signal-to-noise ratio in the upper part.

**Air borne backscatter sonde**

The Compact Optical Backscatter Aerosol Detector (COBALD) has been developed at the ETH (Eidgenössische Technische Hochschule) in Zürich, Switzerland. It is equipped with two high power light emitting diodes (LEDs) driven to ~700 mW radiant flux at 455 (blue light) nm and 940 (infrared) nm. A silicon photodetector that is placed between the LEDs, measures the light scattered back from particles and air molecules at a range extending from 0.5 to 5 m in distance from the instrument. It is typically installed on a standard radiosonde platform alongside with other in situ instruments, and provide a profile of aerosol backscatter at two mentioned wavelengths.

Due to the sensitivity limitations, the device can be used only at night time (https://iac.ethz.ch/group/atmospheric-chemistry/research/ballon-soundings.html), but the accuracy of the profiles provided is expected to be within the error interval of 5%, while precision along the profile is reported to be better than 1% in the upper troposphere and lower stratosphere region (Vernier et al., 2015).

The COBALD instrument has been used to detect aerosol layers (for e.g. Brunamonti et al., 2018, 2020; Vernier et al., 2015, 2018) or cirrus clouds (Brabec et al., 2012). It should be noted the COBALD-like instruments provide the data in the near-ground layer that are usually masked by overlap and, therefore, not available in lidar measurements. As a result, the effective altitude range of the backscatter sonde could stretch from the ground up to the stratosphere.

Two functionally very different GRASP retrieval approaches will be used in the demonstrations:

- **The advanced multi-instrumental retrieval** exploits multi-pixel retrieval approach described in (Dubovik et al., 2011) to combine photometric, lidar and radiosonde data benefiting from the complementarity information from the various measurements even in the situations when the different observations are not fully coincident. The details of the results of such retrievals will be discussed in the Sections 3.1.1 and 3.1.2 correspondingly.

• The application of GRASP for aerosol retrieval from **single instrument** to derive vertically resolved data only including the stand-alone retrievals from the multi-wavelength polarized Mie-Raman lidar and radiosonde observations will be presented in Sections 3.2.2 and 3.2.1 correspondingly. A series of numerical sensitivity tests and application to real data will be presented.

**3.1 Multi-instrumental retrievals**

The multi-pixel approach initially developed in GRASP by Dubovik et al. (2011, 2021) for inverting groups of coordinated satellite observations (image pixels) is apparently a fruitful concept for synergetic processing of ground-based observations. Here, we demonstrate the application of this concept for simultaneous inversion of co-located but not fully coincident backscatter profiles registered by advanced lidar systems or radiosonde, and radiation measured by sun-photometer. Usually,
the intense field campaigns include collocated observations by diverse techniques that provide complementary information but may not be fully co-located and simultaneous due to various reasons. For example, Raman and depolarization channels, lidars and backscatter sondes usually show a better performance (in terms of signal to noise ratio) under a condition when the background signal from the sky is low. Consequently, such measurements are often conducted during night time and cannot be combined with day time sun photometry observations. While nighttime photometry, yet promising (Roman et al., 2017), is
still in its early stages of development (Barreto et al., 2016) and provide only extinction measurements that have lower information content than standard daytime AERONET observations of both direct-Sun and diffuse sky radiation (Dubovik and King, 2000; Dubovik et al., 2000). As a result, combining all co-located data in a single retrieval is not possible because the synergy approaches as those by Lopatin et al. (2013) and Chaikovsky et al. (2016) were introduced for co-incident observations. At the same time, it is clearly seen in numerous studies that aerosol columnar properties do not change drastically
both temporally and spatially, and their temporal and spatial continuity can be used for joint processing of non-coincident (Benavent-Oltra, et al., 2019) or non-collocated (Herreras et al., 2019) multi-instrument observations. In these regards, the multi-pixel retrieval introduced by Dubovik et al. (2011) in GRASP is clearly appropriate for combined inversion of such data. Indeed, the approach realizes rigorous statistically optimized fitting of a group of observations under a priori constraints applied on aerosol time and space variability imposed using limitations on the correspondent derivatives of aerosol parameter
variability in the respect of time or coordinates.

Below, the application GRASP multi-pixel approach will be demonstrated for several sets of different observations. The knowledge about limited time variability of the columnar aerosol properties including complex refractive index, size and shape distributions will be used. Each inverted data set has several time segments, usually 2 or 3. Each of the segments contains a set of coincident and co-located observations. The coincident data in these sets contain photometric measurements including
almucantars combined with elastic lidar and/or depolarization at one or several wavelengths, while other time segments contain only observations of aerosol vertical profiles acquired from inelastic lidar channels or radiosonde data.

The set of retrieved aerosol parameters is exactly the same and is similar to the one proposed in (Lopatin et al., 2013), and successfully used in multiple combined radiometer-lidar data treatments (e.g. Tsekeri et al., 2016, 2017a, 2017b; Bovchaliuk et al., 2016; Benavent-Oltra et al., 2019; Roman et al., 2018; etc.). In contrast with AERONET retrieval (Dubovik and King,
2000, Dubovik et al., 2000, 2006), the aerosol is modeled as bi-component mixture: 25 parameters are used to define size distribution in each time segment (following Eq. (3)), which includes 10 triangular size bins for fine mode in the radius range of 0.05– 0.58 μm and 15 for coarse in the range of 0.33–15.0 μm. The positions of size bins are exactly the same as in AERONET, while there is an intercept of 3 size bins within the range of 0.33–0.578 μm (i.e. for these three size bins, fine and coarse size distribution may have different values). The values of complex refractive index are retrieved for every available
wavelength in the combined set of inverted observations. The retrieved values of complex refractive index could be considered the same for both fine and coarse modes, or retrieved as separate values, effectively doubling the number of parameters used

to describe this aerosol property. Additionally, in the same way as AERONET retrievals (Dubovik et al., 2006), each aerosol mode is modeled as a mixture of two components – spherical and non-spherical with common sphericity fraction retrieved for both fine and coarse modes (see Eq. (5)). Only the concentrations of fine and coarse modes change vertically. The full list of microphysical properties retrieved for each time segment varies depending on the set of observations and is presented in Table 1.

The temporal a priori constraints are applied on the retrievals, specifically on the variability of aerosol volume size distribution (both fine and coarse modes), complex refractive index and spherical particles fraction (for the methodological details of application of such constraints see (Dubovik et al., 2011)). These constraints limit temporal variability of the parameters and stabilize the retrieval of these properties for the nighttime segments that do not contain enough information to robustly retrieve all of them. In contrast with columnar properties, time variability of vertical profiles of fine and coarse modes is not limited. At the same time, additional a priori constraints are applied on smoothness of all the retrieved parameters (except sphericity fraction) within every time segment, similarly to the GARRLiC/GRASP approach, which did not include any multi-temporal constraints and was applied only to simultaneous data. Therefore, a generalized multi-pixel approach realized in GRASP (Dubovik et al., 2011) that allows for applying constraints on variability of aerosol and surface parameters in three dimensions (latitude, longitude and time) is reduced to an application of only multi-temporal constraints in this study, as all provided observations are considered to be spatially co-located, and no spatial variability constraints were used in the retrievals.

**Table 1: Summary of the aerosol properties retrieved and provided by GRASP for each inverted time segment and the parameters of the applied constraints.**

| Aerosol characteristic | | Constraints | | | |
|---|---|---|---|---|---|
| | | Smoothness | | Multi-temporal | |
| (each for evening, night and morning observations) | | Order of finite difference | Lagrange parameter | Order of finite difference | Lagrange parameter |
| $\frac{dV_k(r_i)}{d\ln r}$ | ($i_f$ = 1, …, 10; $i_c$=1, …,15) values of volume size distribution in size bins of fine and coarse aerosol component | 2 | 5.0e-1 | 1 | 5.0e+0 |
| $\frac{dV_k(h_i)}{dh}$ | vertical distribution of aerosol concentration of fine and coarse aerosol component, normalized to 1 ($i$=1, …,100) | 3 | 1.0e-5 | — | — |
| $C_{sph}$ | fraction of spherical particles of aerosol assumed the same for fine and coarse aerosol component | — | — | 1 | 1.0e+0 |
| $n(\lambda_i)$ | the real part of the refractive index for a set of observations wavelengths that is the same for fine and coarse aerosol component | 1 | 1.0e+4 | 1 | 1.0e+1 |
| $\kappa(\lambda_i)$ | the imaginary part of the refractive index for a set of observations wavelengths that is the same for fine and coarse aerosol component | 2 | 1.0e+1 | 1 | 1.0e+1 |

### 3.1.1 Advanced multi-temporal retrievals of COBALD, AERONET and MPLNet

This section describes simultaneous inversion of three data sets AERONET, MPLNet (Micro Pulse Lidar NETwork) and COBALD collected during KAUST.15 and KAUST.16 field campaigns conducted in August 2015 and 2016. These campaigns include rather unique observations by COBALD backscatter sonde performed using balloon flights. The KAUST site is located at the campus of King Abdullah University of Science and Technology, Thuwal, Saudi Arabia (22.3º N, 39.1º E) on a seashore of the Red Sea next to a relatively big city (Jeddah). The site observes strong dust activity due to its proximity to Arabian desert (Parajuli et al., 2020). Altogether 10 radiosonde flights were performed during the campaigns. In addition, observations with AERONET and MPL instruments are performed on a regular basis, covering the campaign periods.

The data acquired during COBALD radiosonde flights were inverted by GRASP in combination with closest-in-time (evening prior and morning after), coincident Sun-photometer/MPL measurements. The details of the data combinations used are summarized in Table 2. AERONET Total Optical Depth (TOD), combined with atmospheric scattering observation in almucantar geometry at four wavelengths 440, 670, 870 and 1020 nm were used in this study. The cloud-screening and other operational quality checks usually implemented for AERONET Level 1.5 retrieval products (Giles et al., 2019) were used in data preparation in order to assure the highest quality of input data for GRASP retrievals. The Micro-pulse lidar (MPL) from MPLNet lidar network (Welton et al., 2001) provided attenuated backscatter (Campbell et al., 2002) and volume depolarization profiles at 532 nm during the campaigns. Since MPL provides a time continuous observation, the signal was accumulated during the time window 15 minutes prior and after the time of AERONET measurement in order to coincide it with radiometric observation. This gives 31 minutes of time accumulation in total, which at the claimed repetition rate of 2500 Hz, results in equivalent averaging of 77500 profiles. Both MPL provided profiles were cropped to the same 100 altitudes grid with the native resolution of 75 m beginning from 280 m (lowest point available) till 7800 m. The upper limit was chosen from adequate levels of signal-to-noise ratio based on the preliminary analysis of the available data of the KAUST field campaigns. This analysis showed that most of the profiles contain very low (close to zero) response values starting from the altitudes higher than 7000 m. Attenuated backscatter signal from lidar was additionally normalized following Eq. (23).

**Table 2: Summary of the data and their combinations used by GRASP multi-temporal retrieval scheme during KAUST campaign.**

| Instrument | Measurement type | Measurement accuracy | Wavelength, nm | Observation set diurnal period | | |
|---|---|---|---|---|---|---|
| | | | | Evening | Night | Morning |
| Sun photometer | Total optical thickness | 0.01 | 440, 670, 870 and 1020 | + | — | + |
| | Almucantar | 5% | 440, 670, 870 and 1020 | + | — | + |
| MPL | Normalized backscatter profile | 30% | 532 | + | — | + |
| | Volume Depolarization Ratio | 0.015 | | + | — | + |
| COBALD | Backscatter ratio profile | 1% | 455, 940 | — | + | — |

COBALD radiosonde data were supplied in the form of vertical profiles of backscatter ratio ($R(\lambda; h)$) at 455 and 940 nm, accompanied with the data of atmospheric pressure $p(h)$ and temperature $t(h)$. The aerosol backscatter was extracted as follows:

$$\beta_a(\lambda; h) = (R(\lambda; h) - 1) \cdot \beta_m(\lambda), \tag{24}$$

where molecular backscatter $\beta_m(\lambda)$ in sr$^{-1}$km$^{-1}$ is calculated from the profiles of pressure $p(h)$ and temperature $t(h)$, provided at the same levels as backscatter ratio by the radiosonde equipment:

$$\beta_m(\lambda; h) = K(\lambda) \cdot \frac{3}{8\pi} \cdot \frac{p(h)}{(t(h) + 273.15)}. \tag{25}$$

The values for the molecular extinction $K(\lambda)$ of 2.6035x10$^{-2}$ km$^{-1}$ at 455 nm and 1.3084x10$^{-3}$ km$^{-1}$ at 940 nm are interpolated from Table 2 of Bucholtz, (1995) and scaled to the reference temperature and pressure (273.15K and 1000 hPa).

The resulting aerosol backscatter profiles were smoothed using a 5-point vertical sliding window for random noise suppression. The resulting aerosol backscatter profiles were downscaled using a linear interpolation to the 100 sample points from 280 till 7780 m with constant altitude increment of 75 m to correspond to the altitude range and resolution of the MPL signals. It should be noted that average time that took a balloon carrying COBALD instrument to reach the altitude of ~8 km, was

estimated to be close to half an hour. Profile measured within this time period is considered to be measured simultaneously at the time, corresponding to the middle of the flight time, i.e. 15 min after launch.

Thus, an observation set used in multi-temporal AERONET/COBALD/MPL retrievals consists of three diurnal sets including two (evening and morning) co-located observations of AERONET and MPL (see Table 2) and backscatter profiles provided by a single COBALD flight at night. All observations are considered to be instant and observing the aerosol properties averaged within a timeframe not exceeding 30 minutes.

### 3.1.1.1 Sensitivity study demonstrating the application of multi-temporal variability constraints

A limited set of sensitivity studies was conducted in order to evaluate limitations and capabilities of the multi-temporal retrievals of combined AERONET, MPL and COBALD observations. All the tests were done according to the following scheme. First, a set of aerosol properties (the same set of components as used in the retrievals) with predefined time dynamics was used to simulate the combined AERONET/MPL/COBALD observations as described in detail in Table 2. Then, these observations were inverted and the obtained results for the retrieved columnar properties were compared to the ones assumed

in the initial simulation. Such methodology allows a rather transparent approach to assess the feasibility of the retrieval and tune the inversion set-up if needed. Indeed, the utilization of the same forward calculation in generation of the data and inversion helps one to eliminate possible uncertainties that may exist in the real data and to focus on fundamental limitations. Additionally, the sensitivities of the retrieval to random and systematic noises can be checked in a controlled environment, that is often difficult to realize with the real data due to lack the detailed information about measurements accuracy.

For keeping the description focused, only one example of sensitivity tests will be demonstrated and discussed in details. Specifically, the study will outline the impact of multi-temporal constraints on the retrieval results when a set of aerosol columnar properties are derived simultaneously from evening, night and morning time observations. A mixture of two distinct aerosols including fine mode dominated absorbing smoke-like and coarse mode dominated non-spherical dust-like aerosols located in separated layers of the atmosphere, were used for the simulation. The values of the parameters used for the

simulations are provided in Tables 5 and 6. Two particular cases are shown: with monotonic and non-monotonic overnight change of coarse mode concentration. No change to other retrieved parameters (listed in Table 1) including refractive index and vertical distribution of both modes were introduced. It should be noted that in the presented sensitivity study the complex refractive indices of fine and coarse mode are retrieved separately. Such approach was chosen in order to assess the feasibility of such complex retrievals in a controlled environment, when the substantial presence of distinct aerosol modes could be

assured. The study was performed under noise free conditions in order to isolate well the impact of multi-temporal constrains on retrievals.

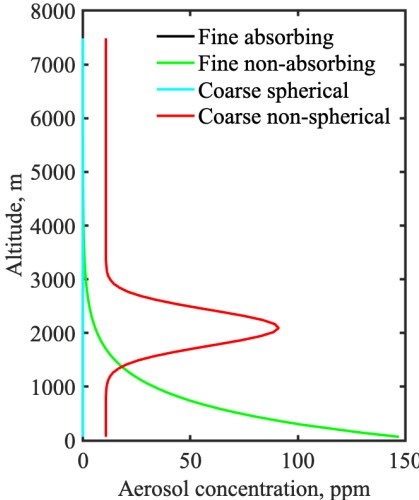

**Figure 1: Aerosol vertical distribution used for the simulation in the COBALD sensitivity test.**

A two-layer aerosol situation with a fine non-absorbing layer close to the ground and a coarse non-spherical layer above was used in forward simulations for modeling vertical structure of aerosol in the atmosphere (see Fig. 1). An exponential distribution with the scale height of 1000 m and Gaussian distribution with mean altitude of 2000 m and geometrical standard deviation of 500 m were used to simulate the aerosol layers correspondingly. The used observation geometries, altitude ranges and vertical resolution were assumed similar to the real observations as described above. The total concentration of each aerosol type was selected as the follows. The fine mode AODF(455) is ~ 0.45 was constant during the observation period. For AOD of coarse mode two scenarios were used: first, AODC(455) varying monotonically overnight from 0.5 in the evening to 0.8 in the morning; and second non-monotonic where AODC(455) rises from 0.5 in the evening to 1 at night and going down to 0.8 in the morning. The second non-monotonic scenario represents a case with a sharp change in the aerosol properties during the night that is not fully consistent with the assumed multi-temporal smoothness constraints on variability of aerosol properties (e.g. for change in size distribution). This test is expected to demonstrate the performance of the retrieval in such scenario that is unlikely but possible in the reality. The results of the retrievals of size distribution and complex refractive indices for fine and coarse modes corresponding to the case of monotonic and non-monotonic change of coarse mode concentration are presented in Figs. 2 and 3 correspondingly.

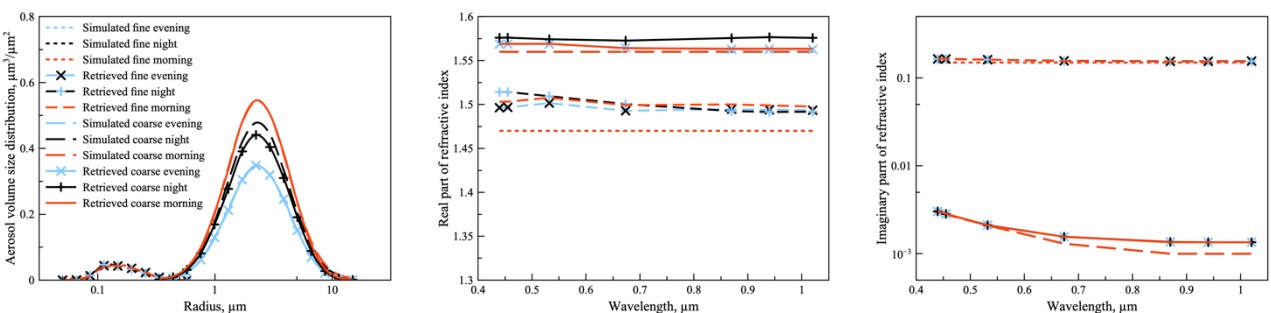

**Figure 2: Aerosol volume size distributions (left), real (center) and imaginary part (right) retrieved from a simulated data depicting monotonic change in aerosol coarse mode concentration.**

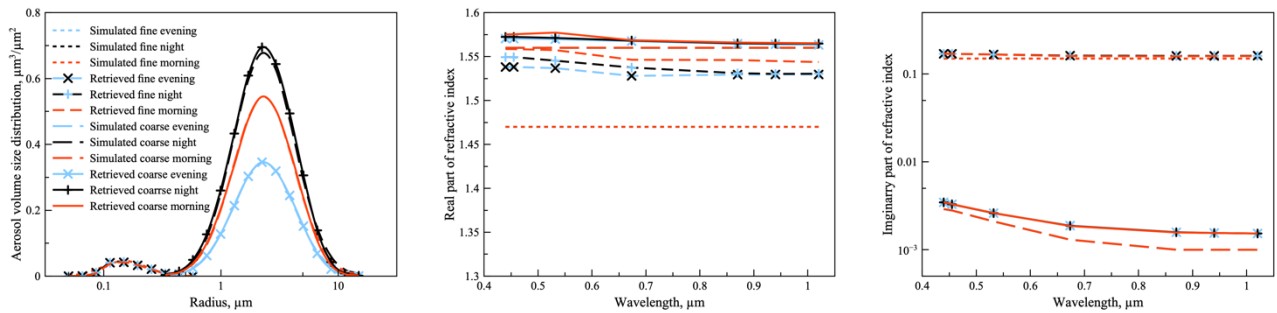

**Figure 3: Aerosol volume size distributions (left), real (center) and imaginary part (right) retrieved from a simulated data depicting non-monotonic change in aerosol coarse mode concentration.**

Figures 2 and 3 illustrate the effect of the multi-temporal limitations applied on aerosol properties from combined AERONET/COBALD/MPL retrievals. Apparently, the retrieval was able to reproduce the non-monotonic change overnight in coarse mode concentration. At the same time, it is obvious from the results (see central panels of Figs. 2 and 3) that distinguishing the refractive indices of fine and coarse mode is a challenging task even in noise free conditions and in a simplified scenario when the refractive indices remain constant overnight. Evidently, the retrieval would suffer even more if variability of real or/and complex refractive index would also be inconsistent with the assumption of smooth temporal variability of aerosol. In these regards, it also worth mentioning the importance of finding an adequate balance of multi-temporal constraints applied to different aerosol properties. For example, as it can be seen from Fig. 3 the retrieval of evening and morning aerosol may be affected negatively if evening, night and morning observations are inverted simultaneously in the case with such unlikely sharp temporal variability of aerosol, when a perturbation in temporal variability of one parameter may manifest in another while ideally fitting the observation data. For example, compare the retrieval of real part of the refractive

index in central panels of Figs. 2 and 3. At the same time, it should be emphasized, that the more elaborate sensitivity study (not shown here) suggest that the retrieval of aerosol in the scenarios with more challenging non-monotonic changes of multiple columnar parameters could be achieved if more advanced observations, such as inelastic or HSRL lidars are deployed at night. Moreover, the extra retrieval test for the similar scenario as in Fig. 3 showed that using an extra assumption of common complex refractive index for fine and coarse aerosol modes may significantly improve the retrieval in the cases when aerosol

is dominated by one of the modes.

Thus, the results of the tests conducted suggest that simultaneous inversion of combined AERONET/MPL/COBALD observations under constraints on the temporal variability of aerosol is feasible and meaningful especially for the situation with smooth and monotonic change of aerosol columnar properties overnight.

### 3.1.1.2 Application to real observations

One of the significant limitations of the multi-instrument synergetic retrievals is the data availability. The analysis of data availability and quality is summarized in the Table 3. As one can see that only 4 observation sets are suitable for the robust processing with the proposed multi-temporal retrieval approach (marked bold) when both evening and morning sets of almucantars and AOD measurements are available. The lack of quality COBALD data on 12/08/15 and photometric data on 08/08/16 makes the combined retrieval impossible since information about vertical aerosol distribution is missing. It is worth

mentioning that a large-scale dust storm swept over KAUST site on August 8–9, 2015, which affected AERONET measurements (Parajuli et al., 2020). Other cases (marked in italic) were processed using only one sun-photometer dataset available but their analysis is less interesting and not included in the article.

**Table 3: The analysis of data availability and quality for different instruments and for different dates of the COBALD flights. N/A denotes data absence, QA indicates absence of quality assured data, bold font indicates presence of full set of data required for the**
**multi-temporal retrieval.**

| Date/period | Evening | | Night | Morning | |
|---|---|---|---|---|---|
| | Photometer | MPL | COBALD | Photometer | MPL |
| **5 Aug 2015** | OK | OK | OK | OK | OK |
| *8 Aug 2015* | **Q/A** | OK | OK | OK | OK |
| *9 Aug 2015* | **N/A** | OK | OK | OK | OK |
| **10 Aug 2015** | OK | OK | OK | OK | OK |
| **11 Aug 2015** | OK | OK | OK | OK | OK |
| 12 Aug 2015 | OK | OK | **Q/A** | OK | OK |
| 8 Aug 2016 | **N/A** | OK | OK | **N/A** | OK |
| *9 Aug 2016* | OK | OK | OK | OK | QA |
| *11 Aug 2016* | OK | OK | OK | **N/A** | OK |
| **12 Aug 2016** | OK | OK | OK | OK | OK |

AERONET observations used for the combined retrieval on 5 of August 2015 were performed at 13:21 UTC (evening) and 05:16:12 UTC (morning, next day). COBALD radiosonde flight was executed at ~19:15 UTC.

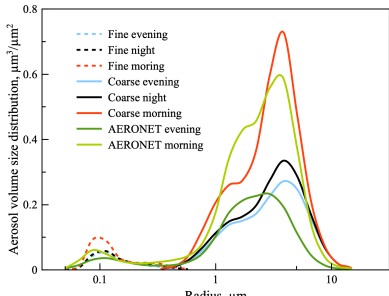 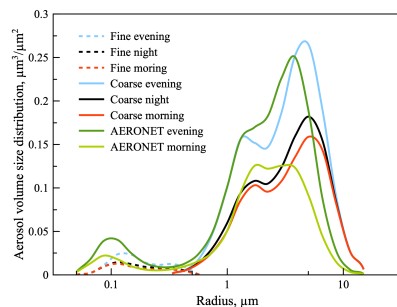 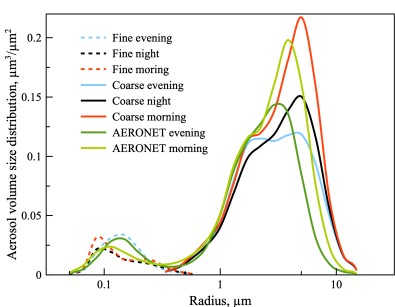

**Figure 4: Aerosol volume size distributions for fine (dashed) and coarse (solid) aerosol components retrieved during COBALD sound flight on 5 of August 2015 (left), 11 of August 2015 (center) and 12 of August 2016 (right). Distributions retrieved before and after the radiosonde flight with use of combined sun photometer and MPL data are shown in blue and red, retrieved by AERONET — in dark and light green, respectively.**

The volume size distributions for fine and coarse aerosol modes retrieved from the observations performed in the evening (blue), at night (black) and in the morning (red) 6 of August 2015 are shown in Fig. 4 overlaid with AERONET retrievals (green) for the same period. All distributions have significant domination of coarse particles, with very limited fine mode. The size distribution changes significantly overnight with higher concentration of coarse particles in the morning than in the evening. This could be explained by the diurnal-scale sea breezes becoming stronger by late morning, which mobilizes dust locally over the study site (Parajuli et al., 2020). At the same time, nighttime retrieval, which largely depends on inter-pixel temporal constraints due to the low sensitivity of backscatter observations to a detailed size distribution, shows a shape that is somewhat between evening and morning retrievals, although closer to the evening one.

Such behavior could be explained by the effect of temporal variability restrictions applied to these distributions as nighttime observations were assumed to be performed in the middle of the radiosonde flight at 19:30 which is ~6 hour after the evening ones and almost 10 hours prior to the observations in the morning. There is a small overestimation of concentrations of coarser particles provided by GRASP combined multi-temporal retrievals in comparison with AERONET. At the same time, it should be mentioned that to understand better the reason of such significant differences between results of GRASP multi-temporal and AERONET standard retrieval, GRASP inversion of almucantar + TOD only data was performed, as well as MPL/AERONET retrievals for evening and morning data, both with no use of temporal constraints. These retrievals (not shown here for keeping size of the paper reasonable) demonstrated a significantly better agreement in volume size distributions of GRASP almucantar + TOD results with standard AERONET results, allowing one to conclude that changes observed above are caused mainly by the inclusion of lidar observations.

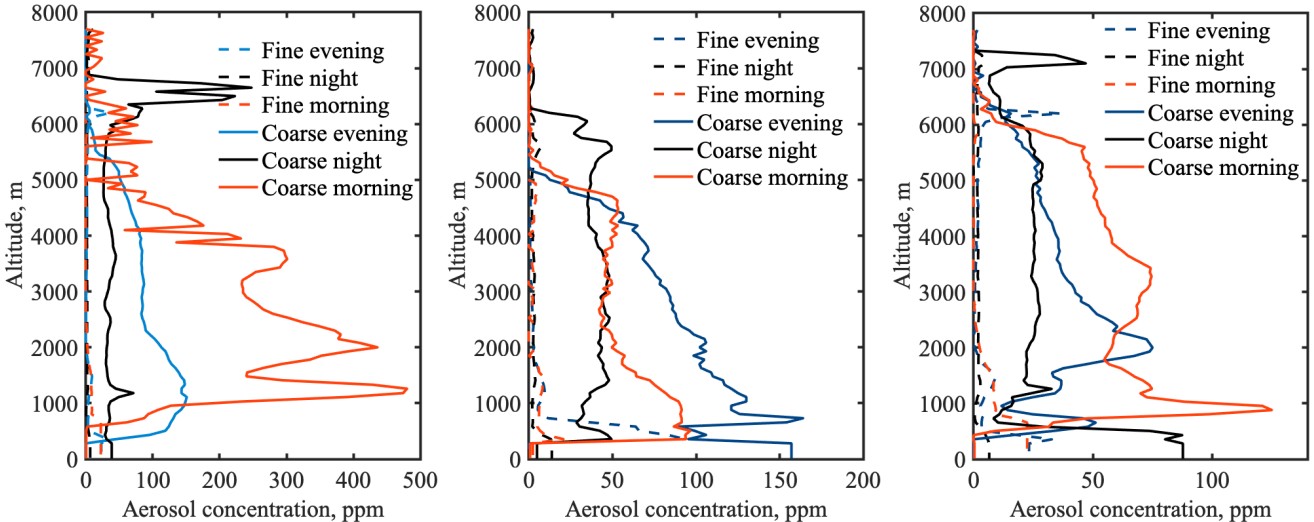

**Figure 5: Aerosol vertical distributions for fine (dashed) and coarse (solid) aerosol components retrieved during COBALD sound flight (black) on 5 of August 2015 (left), 11 of August 2015 (center) and 12 of August 2016 (right). Distributions retrieved before and after the radiosonde flight with use of MPL data are shown in blue and red, respectively.**

The retrieved percentage of spherical particles is almost constant overnight and is close to 1%. This, together with the dominating presence of coarse particles indicates the presence of desert dust, a typical aerosol type for the campaign's region and season.

The retrieved vertical distributions for fine and coarse aerosol modes are shown in left panel of Fig. 5. All distributions are dominated by coarse particles and change significantly overnight. Morning retrieval demonstrates not only the change in particle size, but also in the aerosol concentration in the layers below 4000 m. At the same time vertical profiles that were obtained several hours before and rely only on COBALD observations demonstrate a significant presence of aerosol in the layer between 6000 and 7000 m. This layer is not observed in the evening observations, but is still probably present in the morning profile, although with at slightly lower altitude (5500–6500 m) and with significantly smaller concentration. It should be noted that a similar behavior of nighttime high-altitude layers was also observed on a regular basis during summer time in

a previous study by Parajuli et al., (2020). The authors analyzed two years of MPL data retrieved at night with a similar multi-temporal approach at KAUST site, and concluded that the high-altitude dust was possibly associated with long-range transport of dust from distant sources. Unlike the size and vertical distributions, the complex refractive index values remained almost the same overnight. Real and imaginary parts of the refractive indices are shown in Fig. 6 and 7 correspondingly. All retrievals including AERONET shown in the left panel of Fig. 7 demonstrate a notable decrease of imaginary part with wavelength. The corresponding single scattering albedos are shown in Fig. 8, and demonstrate acceptable agreement with both retrievals indicating no significant change of these parameters overnight. AERONET observations used for combined retrieval on 10 August 2015 were performed at 14:56 UTC (evening) and 04:31 UTC (morning, next day). COBALD radiosonde flight was estimated to be performed at ~19:15 UTC. Although all quality indicators of MPL and sun-photometer data had suggested a high quality of observations, no stable retrieval was achieved, i.e., the resulting fits could not reach noise level expected for each inverted data set. Specifically, combined MPL/sun-photometer data acquired around 14:56 on 10 August 2015 could not be fitted together having significant misfits both in MPL (more than 40% in normalized backscatter) and almucantars (more than 8%) data. For this reason, the data analysis is skipped for this period.

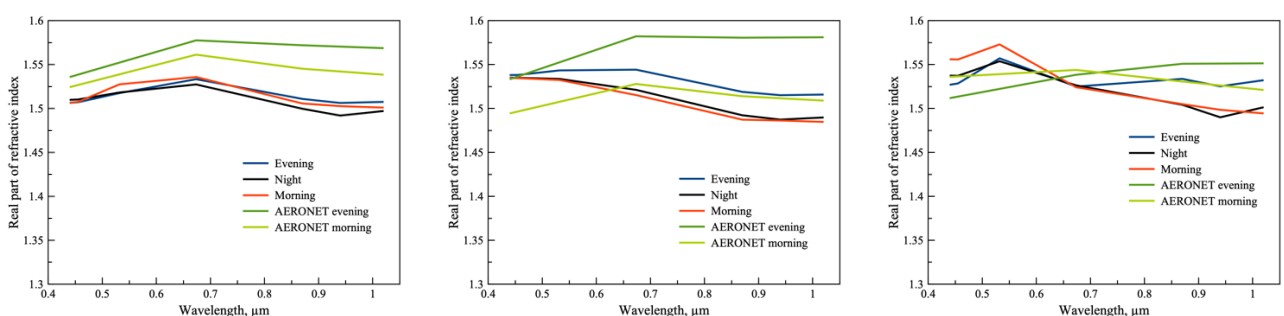

**Figure 6: Real parts of aerosol complex refractive indices retrieved during COBALD sound flight on 5 of August 2015 (left), 11 of August 2015 (center) and 12 of August 2016 (right). Values retrieved before and after the radiosonde flight with use of combined sun photometer and MPL data are shown in blue and red, retrieved by AERONET — in dark and light green, respectively.**

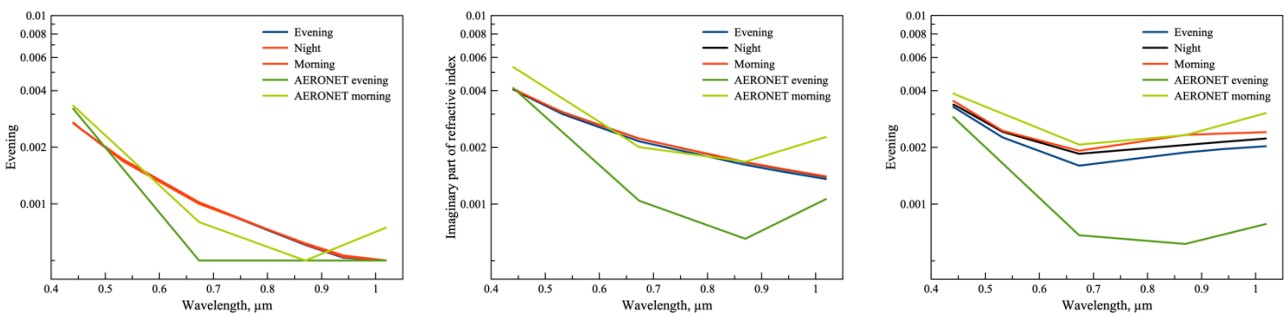

**Figure 7: Imaginary parts of aerosol complex refractive indices retrieved during COBALD sound flight on 5 of August 2015 (left), 11 of August 2015 (center) and 12 of August 2016 (right). Values retrieved before and after the radiosonde flight with use of combined sun photometer and MPL data are shown in blue and red, retrieved by AERONET — in dark and light green, respectively.**

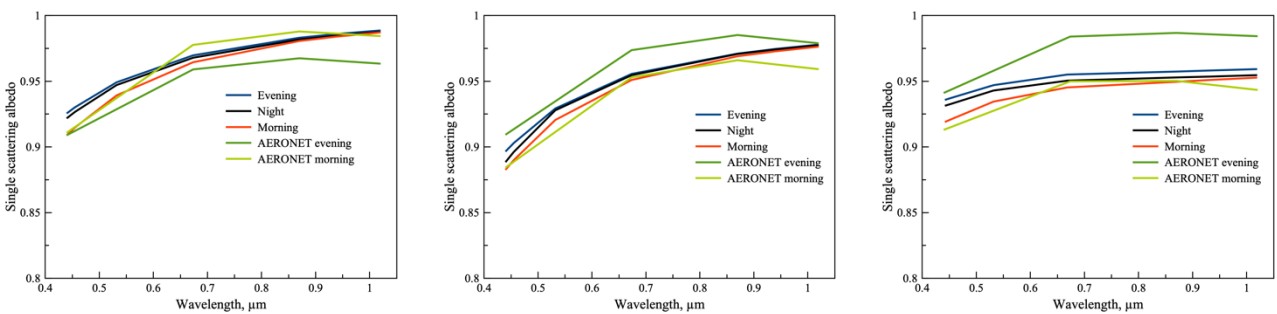

**Figure 8: Aerosol single scattering albedo retrieved during COBALD sound flight on 5 of August 2015 (left), 11 of August 2015 (center) and 12 of August 2016 (right). Values retrieved before and after the radiosonde flight with use of combined sun photometer and MPL data are shown in blue and red, retrieved by AERONET — in dark and light green, respectively.**

The closest observations used for the retrieval on the night of 11 Aug 2015 were performed at 12:31 UTC (evening) and 04:31 UTC (morning, next day). COBALD radiosonde flight was estimated to be executed at ~19:15 UTC. The volume size distributions for fine and coarse aerosol modes retrieved from the observations performed at mentioned time period are shown in central panel of Fig. 4 and compared with AERONET retrievals. All distributions demonstrate significant presence of coarse

particles, with almost no fine mode. Size distribution shows some changes overnight which could be mostly attributed to the change of total concentration, rather than evolution of particle sizes. Note that a large dust storm took place on August 8-10, 2015, which could be the reason why the size distributions show higher concentrations in the evening than in the morning, unlike in left panel of Fig. 4. The tendency of having bigger particles, than AERONET estimations could be observed, similar with left panel of Fig. 4.

The retrieved percentage of spherical particles is almost constant during the period and is close to 1%.

The retrieved vertical distributions for fine and coarse mode aerosols are shown in central panel of Fig. 5. All distributions are dominated by coarse particles of somewhat similar shape showing most of their differences in the total particle concentration. At the same time COBALD provided profile suggest a significantly higher (almost 6000 m) layer of aerosol (black lines), while both lidar profiles (blue and red lines) show no aerosol higher than 5000 m. Such discrepancy could be explained by a

significant diversion of the balloon from the launch point, making it possible for the radiosonde to be affected by aerosol layers unobserved by lidar.

Unlike the retrievals of complex refractive index on 5 August 2015, which remained almost constant (see left panel of Fig. 6), the slight changes could be observed in this parameter overnight 11 August 2015 (central panel of Fig. 6). Again, this could be because of the effect of transported dust particles during the dust storm, as mentioned earlier. The imaginary part has a

notable increase at shorter wavelengths, and remains constant during the observation period. AERONET retrievals performed at the same time demonstrate notable variability of this parameter overnight. The results of multi-temporal retrieval show smaller variability of the retrieved parameter even though some variability could be seen in the corresponding values of SSA (central panel of Figs. 6–8) with intermediate values of both imaginary part of refractive index and SSA obtained for the night compared to the evening and morning AERONET retrievals.

The closest AERONET observations to the COBALD radiosonde flight at ~22:15 UTC on 12 of August 2016 were performed at 14:53 UTC (evening) and 04:34 UTC (morning, next day 13 of August 2015).

The volume size distributions for fine and coarse aerosol modes retrieved overnight of 12 August 2016 are shown in the right panel of Fig. 4. All distributions are dominated by coarse particles and demonstrate similar shapes, showing most of their differences in the particle concentration.

The estimated percentage of spherical particles is almost constant during the observation period and is close to 3.5%.

The retrieved vertical distributions on 12 of August 2016 in the right panel of Fig. 5 show significant similarities with the profiles retrieved on 11 of August 2015 (see central panel of Fig. 5). All overnight profiles show almost uniform aerosol distribution below 6000 m, with COBALD retrieved profiles indicating aerosol presence up to 7500 m, with a distinct aerosol layer on the top of the profile. The aerosol layer on the top (~7500 m) is similar to those observed on 5 and 11 of August 2015,

which is not identified by MPL data but consistently shown by the radiosonde data in all three days. It should be noted that a similar aerosol layer (~6000-8000 km) was reported by Parajuli et al., (2020) in summer while analyzing the nighttime MPL data and which was attributed to the dust transported from remote inland deserts. Unfortunately, the amount of available data does not allow us to conduct a robust conclusion about the reason for such differences or a nature of the high-altitude layers observed at night.

The retrievals of complex refractive index are shown in the right panels of Fig. 6 and 7 correspondingly. The imaginary part behaves quite similarly to the cases described above (right and central panels correspondingly), demonstrating no significant change overnight, and increase of absorption in blue (440 nm) channel – a feature usually associated with desert dust (e.g. Dubovik et al., 2002). However, the absorption in the other channels is slightly higher than that in other presented cases. Slight

temporal changes of the real part of the refractive index are observed, with nighttime retrieval having similarities to the spectral features of refractive indices of both retrieved in the evening and in the morning. AERONET retrievals performed at the same time demonstrate stronger variability of imaginary part of refractive index overnight together with SSA retrievals (right panel of Fig. 8), with both imaginary part of refractive index and SSA values provided by multi-temporal retrieval located in between the evening and morning AERONET retrievals, being generally closer to the estimations from morning observations.

It should be emphasized that the changes in real part of the complex refractive index observed in 2015 (see left and center panels of Fig. 6) is persistent in the data of 2016 as well (right panel), despite the significant constraints applied on temporal variability of these parameters, even though only rather limited sensitivity to refractive index is expected (Dubovik et al., 2000b). Similarly, for the case of 12 August 2016, even though COBALD instrument could not provide enough information for detailed retrievals of aerosol size distributions, the influence of the measurement on these retrieved parameters was strong enough to introduce a notable change.

The measurement fits (the comparison between observed data and their representation by forward model at final iteration) achieved during the above-mentioned retrievals presented in Figs. 9–11 are grouped and combined by five different measurement types used in the retrieval (see Table 2). It should be mentioned that GRASP allows one to assign to each measurement type its own weight, proportional to the estimated accuracy of the observation, that is accounted during the optimization process (see Dubovik and King, 2000 and Dubovik et al., 2011). In these regards, the total optical thickness and sky radiances are expected to be performed with rather high accuracy of 0.01 in absolute and 5% in relative scale, respectively. This results in almost ideal fits achieved (see Fig. 9), slope close to 1 with RMSE and bias close to 0.

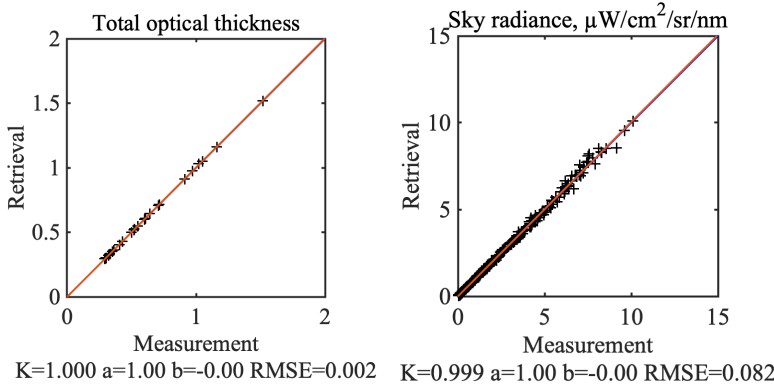

**Figure 9: Achieved total optical thickness (left) and sky radiances (right) fits for the multi-temporal retrievals performed at 5, 11 of August 2015 and 12 of August 2016.**

At the same time, lidar observations, which are subject to multiple technological challenges, are much stronger affected by random noise, specifically, the observations performed during daytime, which results in higher levels of expected noise and, therefore, less accurate fitting results. The results of lidar data fits are presented in Fig. 10. The regression for normalized attenuated backscatter (combined for all evening and morning observations at 532 nm, 600 measurements in total) is presented in log scale due to the high dynamic range of the attenuated signals. Both statistics show in overall good fits, although some challenges still can be observed. For example, the slope of the depolarization regression could be improved, and its fits have a notable yet small bias of ~1%, which is still within the expected accuracy of the observation (1.5%). The notable misfits of attenuated backscatter correspond to the near-ground observations, where the signal is highest due to the lower attenuation, and could be related to MPL overlap correction issues. The misfits observed for depolarization could be related to significant challenges in modeling the properties of non-spherical particles, which dominate the selected retrievals. It is also worth mentioning that 5 outliers with volume depolarization values higher than 35% were removed from regression of depolarization profiles fits. Such high values, never reported in other studies, were detected in the upper atmosphere (undergoing sharp and rapid changes with altitude) where low signal-to-noise ratio can sometimes produce unrealistic estimations of volume depolarization. Although representing less than 2% of the dataset, these outliers greatly complicated the statistical analysis of the rest of the data.

The achieved aerosol backscatter profiles fits provided by COBALD are shown in Fig. 11 and demonstrate almost perfect fits with no bias, good slope (1.01) and minuscule RMSE (0.052).

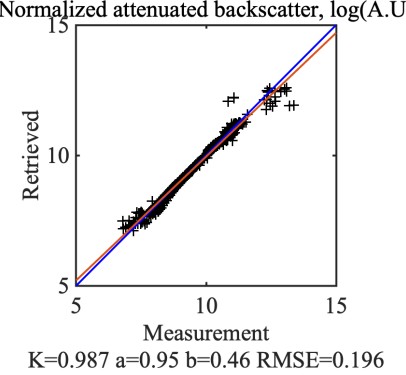 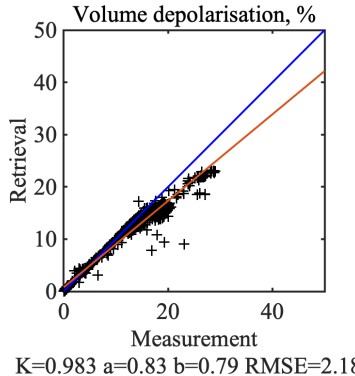

K=0.987 a=0.95 b=0.46 RMSE=0.196        K=0.983 a=0.83 b=0.79 RMSE=2.180

**Figure 10: Achieved attenuated normalized backscatter (left) and volume depolarization (right) fits for the multi-temporal retrievals performed at 5, 11 of August 2015 and 12 of August 2016.**

It should be specifically mentioned that in many cases the observation data usually were fitted even better than their expected accuracy. This can be considered as an indication of good agreement between GRASP forward model and the observations high quality. At the same time, the weights of a priori constraints are dynamically relaxed if residual continues to decrease while the improving of fit is still feasible (see Dubovik et al., 2011) that can create minor artificial features in the retrievals due to overfitting.

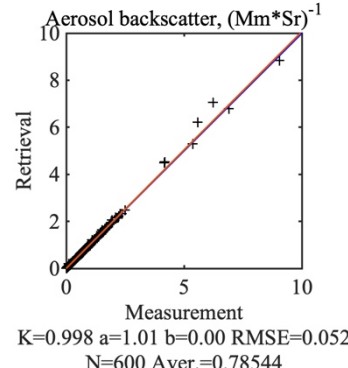

K=0.998 a=1.01 b=0.00 RMSE=0.052
N=600 Aver.=0.78544


**Figure 11: Achieved backscatter fits for the multi-temporal retrievals performed at 5, 11 of August 2015 and 12 of August 2016.**

**3.1.2. Synergetic processing of multi-temporal observations by advanced lidar and radiometer**

    This section describes simultaneous inversion of the AERONET and LIlle Lidar AtmosphereS (LILAS) — the advanced lidar system that incorporates the majority of recent developments in the lidar technique and instrumentation, including multi

wavelength volume depolarization observations and such improvements as inclusion of rotational Raman channel (Veselovskii et al., 2015).

    LILAS data selected for demonstration in this study were collected during SHADOW (SaHAran Dust Over West Africa) field campaign. The study of SHADOW campaign (Veselovskii et al., 2016) was performed in March – April 2015 and December 2015 – January 2016. The SHADOW site is located at the Institute for Research and Developmen (IRD) in Mbour, Senegal

(14° N, 17° W). This site has some similarities with KAUST as being located on a seashore 80 km from Dakar, close to Sahara Desert.

    Table 4 provides the details of the data used for inversion. The GRASP retrieval was applied to the data set of combined nighttime lidar observations and the Sun-photometric AERONET data collected in the evening prior and the morning after. Unlike the MPL system that was used in multi-instrument retrieval in the previous section, the capabilities of LILAS vary

strongly depending on the sky illumination conditions. During the daytime, LILAS provides profiles of attenuated backscatter at 355, 532 and 1064 nm, and profiles of the volume depolarization at 532 nm. At the nighttime, additional inelastic channels can be robustly used to measure inelastic backscatter at 387 and 530 nm. Using these supplemental nighttime observations,

particle extinction $\sigma(\lambda, h)$ (often also denoted as $\alpha$ in lidar studies) and backscattering $\beta(\lambda, h)$ coefficients at 532 nm were derived from a combination of elastic and inelastic channels using the methodology described in (Ansmann et al., 1992). Thus, overall nighttime lidar observations included 7 profiles of different origin: 3 elastic backscatter plus 1 depolarization profiles that are also available at daytime, complimented with inelastic attenuated backscattering profile at 387 and aerosol extinction and backscatter at 532 nm (see Table 4 for details).

**Table 4: Summary of the data and its combinations used by GRASP implying advanced multi-temporal retrieval scheme for SHADOW campaign.**

| Instrument | Measurement type | Measurement accuracy | Wavelength, nm | Observation set diurnal period | | |
|---|---|---|---|---|---|---|
| | | | | Evening | Night | Morning |
| Sun photometer | Total optical thickness | 0.01 | 440, 670, 870 and 1020 | + | — | + |
| | Almucantar | 5% | 440, 670, 870 and 1020 | + | — | + |
| LILAS | Normalized Backscatter Profile | 10% | 355, 532* and 1064 | + | — | + |
| | Volume Depolarization Ratio | 0.015 | 532 | + | + | + |
| | Raman shifted normalized backscatter profile | 10% | 387 | — | + | — |
| | Aerosol extinction profile | 10% | 532 | — | + | — |
| | Aerosol backscatter profile | 10% | 532 | — | + | — |

It should be noted that extinction and backscatter profiles could be also provided at 355 nm by combining elastic and inelastic signals at 355 and 378 nm. However, such profile has a certain limitation: a knowledge of aerosol extinction that is required to extract aerosol backscatter from elastic signal (at 355 nm) using extinction extracted from inelastic channel (at 387 nm) (following Ansmann et al., 1992). Utilizing a prior assumption about aerosol angstrom exponent addresses this issue, but may introduce a bias in the cases when this parameter is estimated incorrectly. This limits the application of the Raman lidar observation techniques mostly to situations when the spectral variations of the aerosol properties are negligible, e.g. for the cases dominated by desert dust particles. In order to avoid all possible inconsistencies in data treatment, the signals at 355 and 387 nm were used for the GRASP retrievals in the form of attenuated backscatters, as opposed to extracted extinction and backscatter profiles at 532 nm. It must be mentioned that LILAS registering rotational Raman scattering lines at 530 nm (Veselovskii et al., 2015), doesn't suffer from such limitation since the difference of emitted and registered frequency in such technique is small enough to be considered negligible.

During SHADOW campaign LILAS observations were performed at a zenith angle of ~43 degrees. The backscattering coefficients and depolarization ratio were obtained with range resolution of 7.5 m with corresponding height resolution of ~5.5 m. The profiles were cropped within the sounding distance range 1500–6500 m (1–4.7 km altitude range). Such a high minimum altitude was chosen due to a big overlap region of LILAS, meanwhile no aerosol was observed higher than 4.5 km during the campaign, and signal-to noise ratio of Raman channels was unacceptably low over distances of 6800 m. The measured aerosol profiles were downscaled to 100 altitude points using a logarithmical altitude grid, similar to the one used by Lopatin et al. (2013). As a result, altitude resolution in the inverted profile varied from ~15 m at 1 km to 70 m at 5 km altitude. The size of the altitude averaging window increases with altitude, which has proven to be effective for additional

diminishing of the measurement noise. In addition, such signal permutations agree with the known tendency of significant decrease of aerosol vertical variability with increase of altitude, therefore no significant loss of information about aerosol vertical variability is expected. Each of the profiles registered by LILAS has a temporal accumulation of 2 to 4 hours in the vicinity of AERONET observation, accumulating approximately 150000 laser pulses. Cropped and accumulated attenuated backscatter signals at 355, 387, 532 and 1064 were normalized as defined by Eq. (23).

The set of aerosol parameters to be retrieved is almost the same as was in the joint inversion of temporal records of COBALD, AERONET and MPLNet described in Section 3.1.1. The only change is that the values of complex refractive index are retrieved at additional wavelengths 355, 387 and 1064 nm. In addition, it is assumed that the value of the refractive index is the same at the close channels of 530 and 532 nm. The parameters that define aerosol size and shape distributions and vertical profiles are exactly the same as in Section 3.1.1.

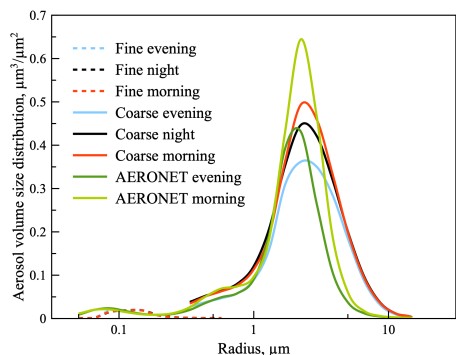


**Figure 12: Aerosol volume size distribution for fine (dashed) and coarse (solid) aerosol components retrieved on 15–16 of April 2015. Distributions retrieved before and after the advanced lidar observation in a combination with sun photometer data are shown in blue and, retrieved by AERONET — in dark and light green, respectively.**

AERONET observations used for the combined retrieval on 15–16 April 2015 were performed at 17:15 UTC and 09:04 UTC

(next day). LILAS observations were performed and accumulated during ~17:00-18:30, 23:00–6:00 and 7:00–10:00 UTC overnight 15–16 April. The volume size distributions for fine and coarse modes of aerosol retrieved from the observations performed in the evening 15 April 2015, at night 15–16 April 2015 and in the morning of 16 April 2015 are shown in Fig. 12 together with the size distributions provided by AERONET for the same observation period. All distributions have significant domination of coarse particles, with almost no fine mode and demonstrate a significant dent in the concentration of particles

of ~2-3μm. The shape of size distribution changes insignificantly overnight with morning retrieval showing higher total concentration of coarse particles. At the same time, it could be seen that nighttime retrieval demonstrates a shape that is in between the evening and morning retrievals. A notable overestimation of concentrations of coarser particles provided by GRASP combined multi-temporal retrievals in comparison with AERONET could be observed, similarly to the distributions shown in Section 3.1.1 (see for e.g. Fig. 4). As it was mentioned above such changes apparently are caused by the inclusion of

lidar observations. Although the sensitivity of sky radiometry to the particles in this size range (Dubovik et al., 2000b) as well as the sensitivity of lidar observation to the particle mean radii (Veselovskii et al., 2016) is rather limited, observed differences don't contradict the indications that desert dust particles may have a bigger mean radius (for e.g. Ryder et al., 2019 or Adebiyi et al., 2020) than reported earlier.

The estimation of the spherical particles fraction is almost constant during the observation period and is close to 0%, providing

a strong indication on the dominance of the desert dust.

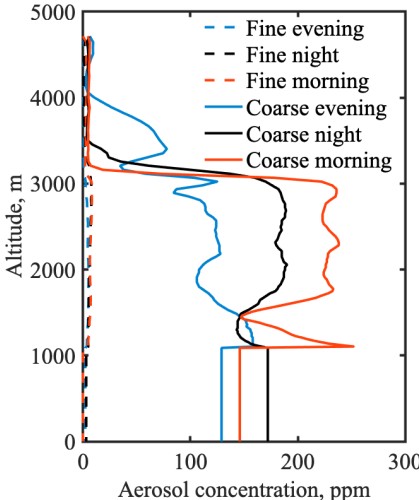

**Figure 13: Aerosol vertical distribution for fine (dashed) and coarse (solid) aerosol components retrieved on 15–16 of April 2015. Distributions retrieved before and after the advanced lidar observation in combination with photometer data are shown in blue and red, respectively.**

The aerosol vertical distributions for fine and coarse aerosol modes retrieved from the observations performed in the evening, at night and in the morning on 15–16 April 2015 are shown in Fig. 13. The vertical profiles show similar tendency as was seen for size distribution with almost no presence of fine mode, and noticeable increase of coarse particles concentrations overnight. All profiles demonstrate thick aerosol layer below 3 km with no particular vertical structure. The profile retrieved in the evening stretch up to 4 km as compared to the morning and nighttime retrieval limited at 4 km. A sharp change in aerosol concentration that is observed in the evening and morning profiles around 1 km is most likely related to the implied assumption on the profiles by Eq. (4). Indeed, radiative transfer calculations require information about aerosol distribution in the whole atmospheric column, which, unfortunately, can't be usually observed using lidars. For such situation, an extrapolation of the profile, to fill the gap between bottom of the atmosphere ($h_{BOA}$) and minimum lidar sounding distance ($z_{min}$) as well as between its top ($h_{TOA}$) and maximum range available from observations ($z_{max}$), is used as described in (Lopatin et al., 2013). The assumption of close to zero aerosol presence at top of the atmosphere should have no significant impact on the retrievals when $z_{max}$ is adequately selected so that most of the aerosol layers with significant load are included. At the same time, most of lidar systems suffer from high overlap regions, and a simple assumption of aerosol concentration at the bottom of the atmosphere (or at ground level) to be equal to the value at lowest range of observation distance ($z_{min}$), may introduce significant inconsistency between lidar and photometric data. In case of LILAS, the overlap range is higher than 1km, i.e. unseen aerosol layer represents significant part of the planetary boundary layer (PBL), where most of the aerosol is usually located. Therefore, the assumption of constant aerosol concentration in first 1 km may cause an overestimation of the lidar derived total aerosol concentration compared to the sun-photometric observations. In order to diminish the artifact and to adjust the columnar value to the AOD observations, the lowest point of the retrieved aerosol vertical distribution profile could be lowered, however this results in some degradation of the lidar data fit at lowest altitude. An approach to address such issues was suggested in (Bovchaliyuk et al., 2016) attempting to retrieve aerosol concentration at ground level, instead of intrinsically assuming its value. However, retrieval of this parameter for the night segment is particularly challenging in the described AERONET-LILAS data set, as no observations that may constrain it through total columnar aerosol concentration are available at night. In future efforts, it is planned to include the lunar photometry data in order to assure sufficient constrains for the nighttime segments for overnight synergy retrievals.

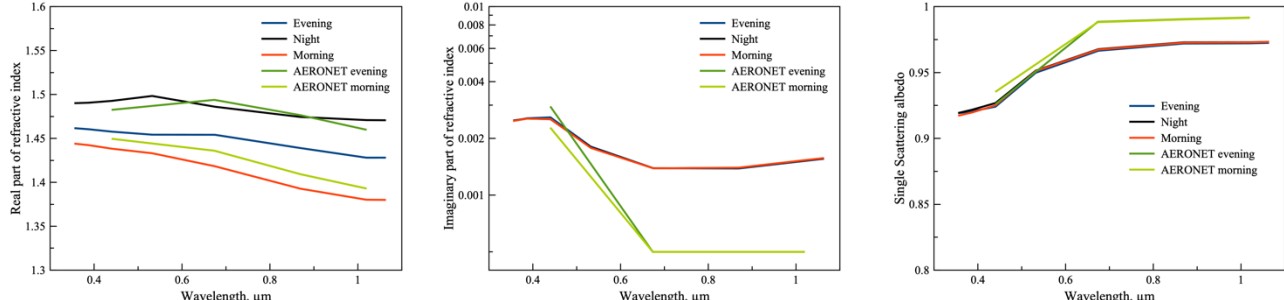

**Figure 14: Real (left) and imaginary (center) parts of aerosol complex refractive indices and corresponding single scattering albedos (right) retrieved on 15–16 of April 2015. Values retrieved before and after the advanced lidar observation in a combination with sun photometer data are shown in blue and red, retrieved by AERONET — in dark and light green, respectively.**

The retrievals of complex refractive index are shown in Fig. 14. Imaginary part demonstrates no significant change overnight with small increase of absorption in ultraviolet (355 nm), blue (440 nm) and green (532 nm) channels. The values of absorption at 400 nm and onward are similar to those reported by Dubovik et al. (2002) for desert dust. At the same time, significant temporal variation of the real part of the refractive index could be observed, with nighttime retrieval having values higher than retrieved in the evening and in the morning. Similar temporal variation of the real part of refractive index was observed by AERONET, with the values of multi-temporal retrievals demonstrating comparable variability. At the same time a more significant differences could be observed in values of imaginary part (central panel of Fig. 14), which meanwhile cause a less dramatic change in the estimations of single scattering albedos provided by two retrievals (left part of Fig. 14), both indicating no change of these parameters overnight. Apparently, aerosol with such parameters reproduces better the joined set of inverted radiometric and lidar data.

The achieved measurements fits are shown in Figs. 15–17 and similarly to the Section 3.1.1 are grouped and combined by seven different measurement types used in the retrieval (see Table 4).

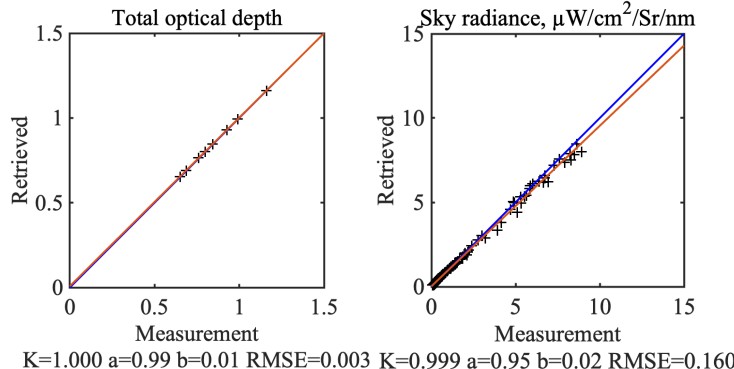

**Figure 15: Achieved total optical thickness (left) and sky radiances (right) fits for the multi-temporal retrievals performed on 15–16 of April 2015.**

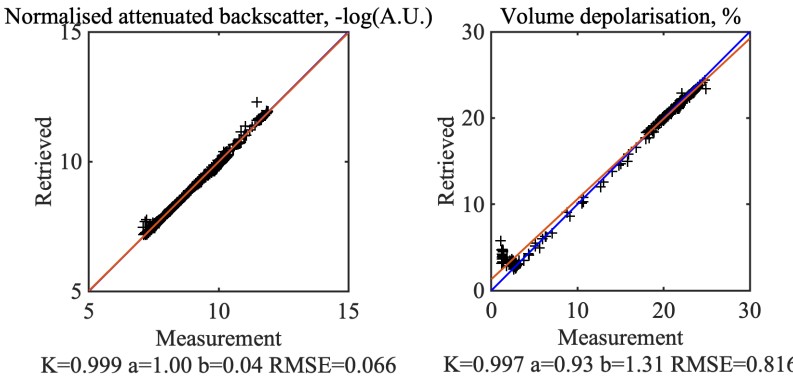

**Figure 16: Achieved attenuated normalized backscatter (left) and volume depolarization (right) fits for the multi-temporal retrievals performed on 15-16 of April 2015.**

Similarly to the results described above in Section 3.1.1 total optical thickness and sky radiances in Fig. 12 demonstrate almost ideal fits achieved with slope close to 1, bias close to 0 and negligible RMSE.

The results of elastic lidar data fits are presented in Fig. 16. Both statistics show very good fits, with slope close to unity and negligible biases. The regression for normalized attenuated backscatter (combined day, night and morning observations at 355, 532 and 1064 nm, 800 measurements in total) is presented in log scale due to high dynamic range of the attenuated signals. It should be particularly outlined that unlike combined sun-photometric/MPL and COBALD retrievals described in Section 3.1.1, depolarization fit of LILAS data does not demonstrate any notable negative bias in the area 5-25%, having very good slope of

0.93, despite of generally higher values of observed volume depolarization ratios, almost reaching 30%. Such difference, when exactly the same aerosol modeling in comparable conditions fits one data perfectly and another with a notable bias, may indicate differences in depolarization calibration quality of LILAS and MPL lidars. At the same time, it should be outlined that there is a group of overestimated points, that contributes to slope and bias values. This group comes from the upper (4-5 km) layers of the atmosphere, where quite low values (1-3%) of volume depolarization were registered. Unfortunately, the

forward model cannot reproduce such low values, due to the limitations in aerosol modelling. Indeed, following Eqs. (6) and (22) the vertical variation of volume depolarization could be achieved only by changing the relative proportion between fine and coarse particles, each having vertically constant depolarization ratios. At the same time as it can be seen from Figs. 13 and 16, the small total concentration of the fine mode couldn't provide enough impact to lower down volume depolarization at 4–5 km, rendering the retrieval unable to properly fit the data. One of the possible ways to address this issue is to use another

aerosol model that allows changing vertically the effective properties of the coarse mode, one of such approaches will be discussed in Section 3.2.

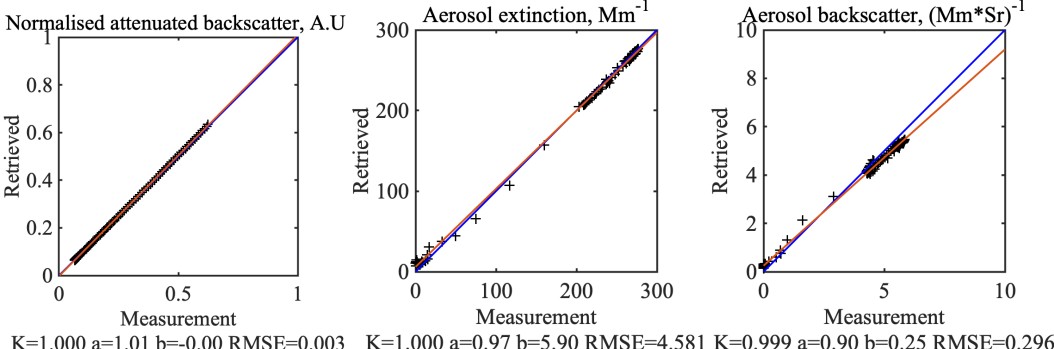

**Figure 17: Achieved attenuated normalized inelastic backscatter (left), aerosol extinction (middle) and aerosol backscatter (right) fits for the synergy retrievals performed on 15–16 of April 2015.**

The fits for LILAS lidar data, acquired during nighttime observations are shown in Fig. 17 and include normalized attenuated inelastic backscatter at 387 nm together with aerosol extinction and backscatter profiles at 532. All regressions demonstrate accurate fits that are particularly good in case of inelastic backscatter and aerosol extinction. A minor bias observed in aerosol backscatter profiles is most probably related to a small, but nonetheless not negligible difference of 2 nm between sounding and receiving wavelengths in the green channel, which were considered to be the same to reduce the complexity of the retrieval.

Thus, this section demonstrates capabilities of GRASP for realizing synergy processing of the ground-based sun-photometric and advanced lidar observations including Raman and depolarization data and data of the sun-photometric and micro-pulse lidar combined with in situ radiosonde backscatter observations collected during SHADOW and KAUST field campaigns. It was shown that the multi-pixel approach developed within GRASP concept could be efficiently used for combining not co-incident but fully co-located and close in time observations of various origins, for e.g. day- and night measurements, by

providing sufficient constraints on aerosol columnar properties variability to provide additional benefits to the retrievals of the night-time observations that are usually lacking sensitivity to qualitatively retrieve these parameters.

## 3.2 Stand-alone instrumental retrievals

All earlier application of GRASP used vertically resolved lidar data only in combination with co-incident radiometric observations, with the idea of deriving additional details about aerosols from the complementary lidar and radiometric data (e.g., Lopatin et al., 2013; Bovchaliuk, et al., 2016; Tsekeri et al., 2016, 2017a, 2017b; Roman et al., 2018; Benavent-Oltra et al., 2019; Hu et al., 2019). As a result, the possibility of explicit single-instrument inversion of vertically resolved observations was not initially offered in frame of GRASP developments while there is always significant interest to such applications in active remote sensing community. Such option was included in the recently updated version of GRASP and offered to the community. Another advantage of this option is higher data availability for the nighttime retrievals, which rely on the on the quality and availability of additional observations (see Table 3 for e.g.). Here, unlike a multi-temporal approach described in Section 3.1, only single sets of measurements were inverted. In these regards the retrieval could be considered as single pixel, following the terminology used by Dubovik et al. (2011) and above in Section 3.1.

The main difference of the stand-alone lidar or backscatter sonde retrievals from the combined GARRLiC like inversions is significantly lower information content regarding columnar properties of aerosol. Therefore, a number of corresponding parameters retrieved from a stand-alone lidar or backscatter sonde may need to be reduced. For example, retrieval of sphericity fraction, the details of size distributions and spectral variability of refractive index may be challenging. At the same time, there is clear some sensitivity to these properties in most vertically resolved measurements. Therefore, an external mixture of several aerosol components (see Eqs. (8–9) and associated discussion) seems to be appropriate for single-instrument lidar or backscatter sonde retrievals. Indeed, such an approach uses a smaller number of parameters, while allowing for retaining implicit sensitivity of the retrieval to the variability of nearly all aerosol properties.

Thus, the external mixture of several aerosol components is used to model aerosol single scattering properties following Eqs. (8–9 and 15–20) and employed to fit available LILAS and COBALD observations. Therefore, the set of retrieved parameters includes the vertical profiles of the concentrations for each aerosol component. Thus, the outcome of such retrieval provides a set of vertical profiles describing a fraction of each aerosol component in the total aerosol volume concentration. Then any other optical or microphysical properties can be recalculated using these fractions and assumed properties of each aerosol component. In the present study the four following aerosol components were used: fine absorbing, fine non-absorbing, coarse spherical non-absorbing and coarse non-spherical components. Based on preliminary analysis and the sensitivity analysis the proposed set allows for accounting for variability of the aerosol size, absorption and shape leaving additional opportunity to provide information in the format convenient for comparison and for assimilation with the global transport models (Chen et al., 2018, 2019).

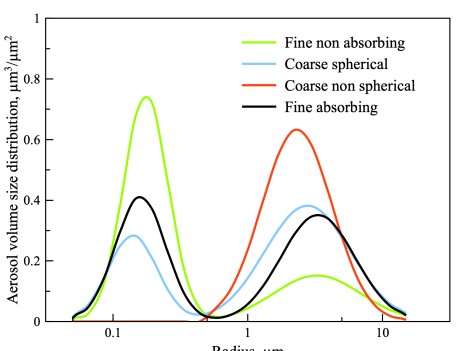

**Figure 18: Normalized volume size distributions of different aerosol components.**

The detailed microphysical properties of the proposed aerosol types, including parameters of bi-lognormal size distributions, values of complex refractive indices and fraction of spherical particles were selected based on the global analysis of abundant aerosol species over selected AERONET sites listed in Table 1 in (Dubovik et al., 2002). For the fine absorbing component, averaged aerosol properties over Mexico City site were selected. For the fine non-absorbing component the aerosol routinely

observed over GSFC (Goddard Space Flight Center) site was taken, for the coarse non-spherical type the properties of the desert dust aerosol abundant over Solar Village site in Saudi Arabia were chosen. For the coarse spherical component the microphysical properties on maritime model by (Smirnov et al., 2002) were adapted.

Such AERONET climatology set of aerosol compounds was used for processing 10-years of MERIS/Envisat passive observations (Dubovik et al., 2021). The archives of developed data products can be found at ([https://www.grasp-open.com/products/meris-data-release/](https://www.grasp-open.com/products/meris-data-release/)). Similar set of components but with additional fine medium absorbing component was used for processing the 9-year global PODLER-3/PARASOL data archive (see Chen et al., 2020) and Airborne Hyper-Angular Rainbow Polarimeter (AirHARP) observations during ACEPOL 2017 campaign (Puthukkudy et al., 2020). Following the

analysis of the above results some modifications were done to the chosen aerosol components for lidar/backscatter sonde applications. Specifically, two main changes were done. First, the increase of the absorption of fine absorbing aerosol type, for capturing the whole range in aerosol absorption variability. Second, the coarse-non spherical component is represented by only coarse mode in order to improve the allowed ranges of Angström exponent and depolarization ratio variability. All parameters including parameters describing aerosol volume size distribution, complex refractive index and sphericity could be

found in Table 5, where the values modified in comparison with the originals are marked in bold. Figure 18 illustrates the comparison of normalized size distributions with the parameters described in Table 5 for the selected aerosol types.

**Table 5: Particle volume concentrations ($c_v$), modal radii ($r_m$), geometric std dev ($\sigma_g$) of bi- lognormal size distributions, and refractive indices at 500 nm (adopted with modifications from (Dubovik et al., 2002 and Smirnov et al., 2002) modifications shown in bold).**

| Aerosol type | Fine mode | | | Coarse mode | | | Refractive index at 500nm |
|---|---|---|---|---|---|---|---|
| | $c_v^f$ | $r_m^f$ | $\sigma_g^f$ | $c_v^c$ | $r_m^c$ | $\sigma_g^c$ | |
| Fine non-absorbing | 0.72 | 0.175 | 0.38 | 0.28 | 3.275 | 0.75 | 1.395–0.003$i$ |
| Coarse spherical | *0.30* | *0.140* | *0.42* | *0.70* | *2.780* | *0.73* | *1.37–0.0001$i$* [*] |
| Coarse non-spherical | **0** | — | — | **1.0** | 2.320 | 0.60 | 1.56–$k(\lambda)i$ [**] |
| Fine absorbing | 0.52 | 0.160 | 0.43 | 0.48 | 3.320 | 0.63 | 1.47–**0.15**$i$ |

[*] Adapted from (Smirnov et al., 2002).
[**] Coarse non-spherical aerosol has spectrally dependent imaginary part of refractive index, extrapolated from (Dubovik et al., 2002). Fine mode was completely removed for this study. Detailed values at wavelengths concerning presented study can be found in Table 6.

As can be seen from Table 5 all aerosol components proposed except coarse non-spherical component have spectrally

independent complex refractive index. Indeed, the desert dust observations were used as a basis for defining microphysical properties of the coarse non-spherical aerosol component, that demonstrates a notable growth in absorption in the blue channel (Dubovik et al., 2002). Since AERONET climatology provided the refractive indices only at wavelengths 440, 670, 870 and 1020 nm a linear inter- or extra- polation was performed in order to define the values of the imaginary part of refractive index of the coarse non-spherical component at the wavelengths that different from above four. This includes the channels of a multi-

wavelength lidar system equipped with a commonly used YAG laser, notably 355, 532 and 1064 nm together with widely used spectral window of Raman scattering on atmospheric nitrogen of 387 nm and wavelengths of the COBALD backscatter sonde of 455 and 940 nm. The values of imaginary part of coarse non-spherical component are given in Table 6. It can be mentioned, that for the convenience of applying GRASP to diverse multi-instrumental retrievals, the properties of the components were calculated for a significantly larger set of wavelengths (~30) that covers most of the common observation wavelengths

including both satellite and ground-based instruments used, but the details are given only for specified channels for the brevity of Table 6. This extended data set of aerosol components should help users to reduce the efforts if they decide to try the concept for applications somewhat different from the ones performed in this paper.

It should be noted, however, that utilization of such external mixture of aerosol component does not assume that ambient aerosol microphysics is closely described by this model, instead it is expected that such multi-component aerosol mixture can

rather adequately describe the effective optical properties of the aerosol such as total scattering, absorption and phase matrices.

**Table 6: Spectral dependence of the complex refractive index of the coarse non-spherical aerosol type.**

| Wavelength, nm | 355 | 387 | 440 | 455 | 532 | 670 | 870 | 940 | 1020 | 1064 |
|---|---|---|---|---|---|---|---|---|---|---|
| $k(\lambda)$ | 0.0037 | 0.0034 | 0.0029 | 0.0028 | 0.0021 | 0.0013 | 0.001 | 0.001 | 0.001 | 0.001 |

### 3.2.1 Stand-alone COBALD retrievals

The measurements by COBALD provide two backscatter profiles at 455 and 940, a mixture of components described above at each given altitude is expected to provide sufficient flexibility for describing these aerosol properties following Eq. (13). As mentioned above, each aerosol component is described by its own vertical distribution profile, defining the contribution of this component at specific altitude to the total mixture of the observed layer, and a value of its total columnar concentration. Therefore, 101 for each aerosol component and 404 parameters for total aerosol mixture are retrieved from COBALD observations. The list of the parameters retrieved is presented in Table 7. Additional constraints on the vertical variability of the retrieved profiles were applied by limiting the 3-rd derivatives of vertical distribution of aerosol concentration over height, in a similar manner as proposed by Lopatin et al. (2013). Indeed, a retrieval of 4 aerosol component concentrations at each altitude layer, from only two observations is an ill-posed problem, therefore additional a priori restrictions are needed to assure sufficient information for making the retrieval feasible.

**Table 7: Summary of the aerosol properties retrieved and provided by GRASP implying stand-alone retrieval scheme for COBALD and LILAS instruments and the corresponding parameters of the applied constraints.**

| Aerosol characteristic | | Constraints | | | |
|---|---|---|---|---|---|
| | | Smoothness | | Multi-temporal | |
| | | Order of finite difference | Lagrange parameter | Order of finite difference | Lagrange parameter |
| $c_k$ | total columnar volume concentrations for each aerosol component (k=1, …, 4) | — | — | — | — |
| $\dfrac{dV_k(h_i)}{dh}$ | Normalized vertical distribution of aerosol concentration for each aerosol component ( k=1, …, 4; $i$=1, …, 100) | 3 | 1.0e-5 | — | — |

### Sensitivity study

A small set of sensitivity studies was conducted in order to evaluate limitations and capabilities of the standalone COBALD retrievals. All the tests were done according to the following scheme. First, a set of concentration profiles of predefined aerosol components (same set of components as used in the retrievals) was used to simulate the backscatter profiles at 455 and 940 nm. Second, these profiles were inverted and the profiles of retrieved optical properties were compared to the ones used in simulation. Such approach allowed a rather transparent approach to assess the feasibility of the retrieval, as well as to tune up the inversion set-up if needed. Indeed, the utilization of the same forward calculation in generation of the data and inversion allows one to eliminate possible uncertainties that may exist in the real data and to focus on fundamental limitations. Additionally, the sensitivities of the retrieval to random and systematic noises can be checked in a controlled environment, since usually the real data lack the detailed estimations of measurements accuracy.

For keeping the description compact, only one example of sensitivity tests will be demonstrated and discussed in details. The same two-layer aerosol distribution that included a fine non-absorbing layer close to the ground and a coarse non-spherical layer above as described in section 3.1 was used in forward simulations (see Fig. 1). The used altitude range and vertical resolution were assumed similar to the real observations as described above in Section 3.1.1. The total concentration of each aerosol type was selected to provide AOT at 455 nm close to 0.5 with virtually equal contributions of each type.

The retrieval of all aerosol components as described in Table 7 was performed using the simulated data. Specifically, two types of the retrievals were realized: with no noise added and with 5% of random noise added to the backscatter profiles. Figure 17 shows the comparison between modelled and retrieved aerosol extinction profiles for all four components combined in noise free and with noise added conditions. Figure 19 compares the modelled and retrieved profiles of Angström exponent (AE) at

1115     455/940 nm for the same conditions. Comparison of AE profiles was made in order to demonstrate that not only good reproduction of modeled extinction and backscatter but also the correct estimation of aerosol spectral properties is possible if aerosol is represented by the employed aerosol mixture.

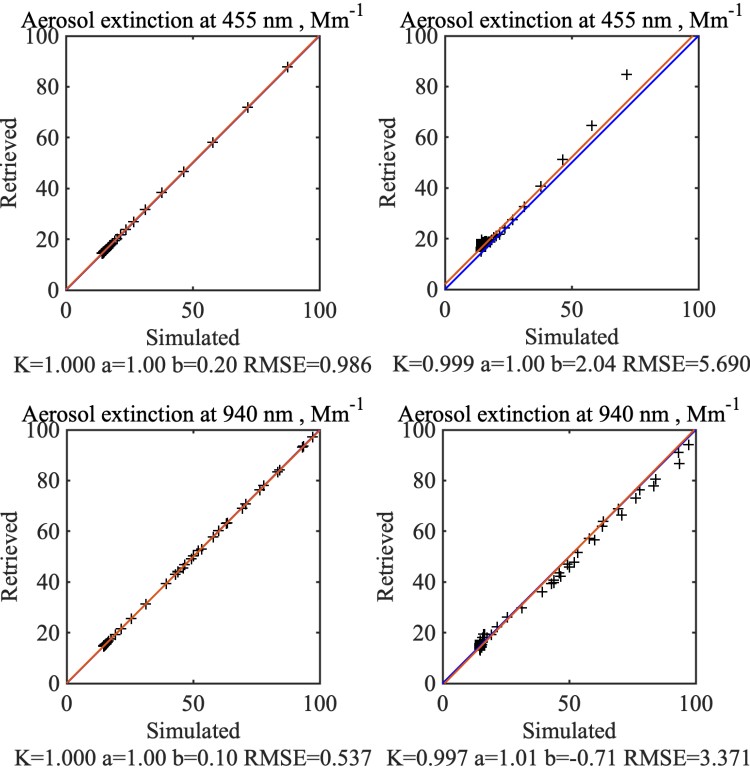

**Figure 19: Comparison between simulated and retrieved aerosol extinction profiles at 455 (top) and 940 (bottom) nm in noise free (left) and noisy (right) conditions.**

Comparison of aerosol backscatter fits are shown in lower part of Fig. 21. It should be mentioned that the fits achieved in the noisy conditions had a resulting residual close to the expected noise level (5%).

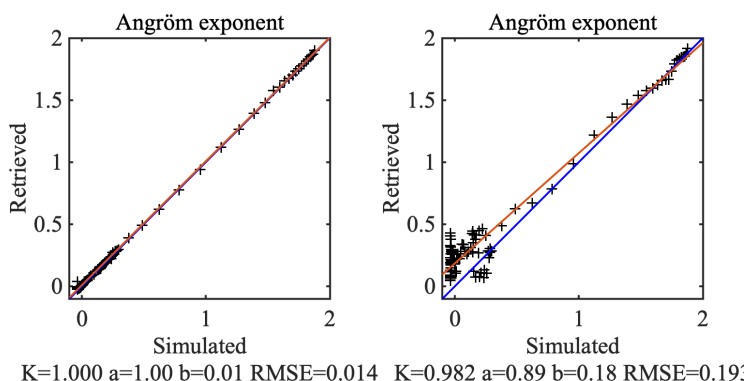

**Figure 20: Comparison between simulated and retrieved Angström exponent profiles at 455/940 nm in noise free (left) and noisy (right) conditions.**

Thus, from analysis illustrated by Figs. 19–21 it could be concluded that despite of using a rather limited data set (backscattering at only two wavelengths) and a quite complex model (for given application), the stand-alone COBALD instrument retrievals using GRASP can provide rather reliable profiles of aerosol optical properties even in the conditions with

1130     the presence of random noise. It should be underlined, that an external mixture of aerosol components is an approximate model that is expected to adequately mimic total optical properties while the presence of each component may not correspond to the

reality. Therefore, the total vertical extinction profile as well as their Angström exponent could be expected to be robust even though affected by the presence of the measurement noise. Indeed, in the retrievals considered here, the retrieval errors remain reasonable, not exceeding 10 Mm$^{-1}$, with a RMSE of ~5Mm$^{-1}$ for aerosol vertical extinction, and less than 0.5 with an RMSE of ~0.2 for Angström exponent. It should be additionally emphasized that the biggest errors in the AE estimation are associated with the layers characterized by low extinction values.

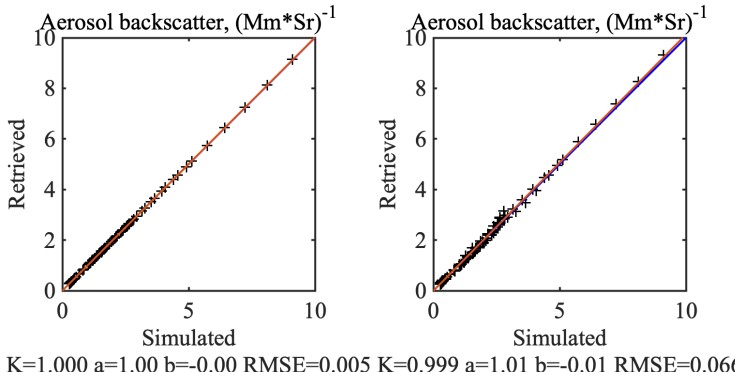

**Figure 21: Comparison between simulated and retrieved aerosol backscatter profiles at 455 and 940 nm combined in noise free (left) and noisy (right) conditions.**

**Application to real observations**

Overall, 9 profiles provided by COBALD instrument flights were considered. The stand-alone instrument inversions do not need other instrument data to be present, therefore more COBALD data were available for the considerations here compared to the analysis of combined retrieval considered in the Section 3.1.1 (all marked in bold and italic in Table 2). The retrievals were performed applying a slightly shifted altitude range as compared to the combined sun-photometer/MPL/COBALD retrievals described in Section 3.1.1 in order to benefit from the sensitivity of radiosonde measurements to the lower atmospheric layers, where no lidar data is available due to the overlap. This way all available and quality assured COBALD observations were processed with the altitude range of 140–7560 m and 65–7485 m for the data of KAUST.15 and KAUST.16 campaigns respectively. As mentioned previously the altitude sampling of COBALD profiles was aligned with the one of MPL observations, in order to make inter-comparisons and analysis easier. Based on the fixed altitude sampling grid, the number of points in vertical profiles of the retrievals were set to 100. For the consistency the same radiosonde flights as in the Section 3.1.1 were analyzed and compared with the results from combined retrievals. The details of the aerosol component vertical distributions for the dates of 5, 11 August of 2015 and 12 August of 2016 could be found in Figs. 22 – 29 Achieved backscatter vertical profiles fits are combined and shown in Fig. 29.

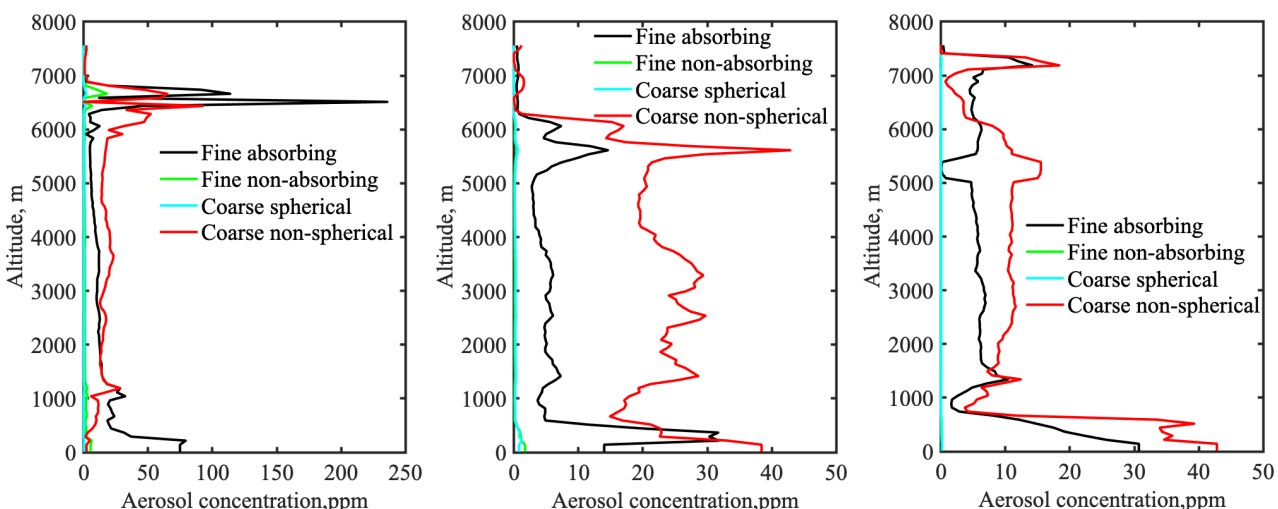

**Figure 22: Aerosol vertical distributions retrieved from stand-alone COBALD observations performed on 5 August 2015, 11 of August 2015 and 12 of August 2016.**

The retrieval performed on measurements of 5 August 2015 and shown in ther left panel of Fig. 22 demonstrates a general presence of coarse non-spherical component, which is mixed with a fine absorbing aerosol in the range of 1–8 km. The retrieval also indicates pronounced layers of fine absorbing particles below 1 km that could represent layers containing aerosol pollution.

There is no significant presence of fine non-absorbing or coarse spherical particles at any layer. A strong elevated thin layer containing both types of particles can be observed at ~6500 m, which is in agreement with the combined multi-temporal retrievals for this night (see left panel of Fig. 5), which also has a mixture of fine and coarse particles in this atmospheric layer. Thus, while here much simpler aerosol model is used than in the combined retrieval in Section 3.1.1, the comparison demonstrates encouraging consistency.

For quantitative comparison vertical profiles of extinction, Angström exponent and backscatter profiles at 455 and 940 nm provided by combined and single instrument retrieval for the COBALD flights on 5 of August 2015 are shown in Figs. 23–24. The extinction and backscatter profiles show excellent agreement. While such behavior could be expected from backscatter profiles that are directly constrained by COBALD observations, the fact that stand alone COBALD inversion can provide a vertical extinction profile of comparable accuracy compared with much more complex combined observations processing is

encouraging and agrees with the findings of the sensitivity study presented above. A reasonable agreement between retrieved Angström exponent profiles could be observed in Fig. 23, the average difference of 0.4 between the profiles is not exceeding the estimations of AE retrievals provided by the sensitivity study above. The closest observations of AE 440/870 performed by AERONET give estimations of ~0.3–0.5 change overnight, as compared to averaged values of ~0.6 and ~1.0 for multi-temporal and component approaches correspondingly. Indeed, the latter is significantly better constrained due to the inclusion

of high accuracy direct measurements of aerosol optical thickness. This in certain extent confirms the stability and consistency of GRASP inversions of COBALD observation while a somewhat simplified methodology was employed and a different observation set was used. Some differences, observed in the area close to the ground could be explained by a different altitude range used in both retrievals. While for combined retrievals an extrapolation should be performed to fill the gap between the ground level and minimal range of observation, radiosonde usually starts its observations at a much lower altitudes, therefore

providing a more accurate information on aerosol structure in the lower layers.

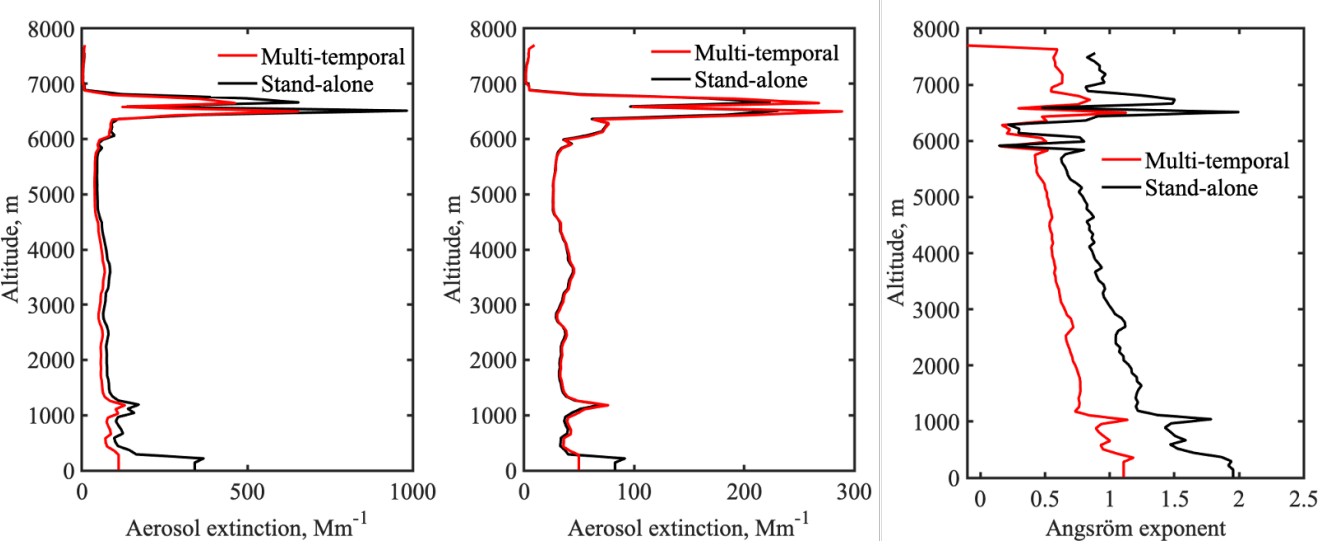

**Figure 23: Aerosol extinction profiles at 455nm (left) and 940 nm (center) and corresponding Angström exponent (left) estimated from COBALD profiles acquired on 5 August 2015 using a stand-alone (black) and multi-temporal (red) retrieval schemes.**

The retrieval results from COBALD observations on 11 of August 2015 are shown in the center panel of Fig. 22. The aerosol

below ~6000 m is dominated by coarse non-spherical particles. The second most abundant aerosol type is the fine absorbing component, with negligible (almost zero) presence of other aerosol components. The shape of vertical profile of fine absorbing particles replicates the shape of the coarse non-spherical component profile within the altitude range 1000–6000 m. This could indicate a well-mixed layer of desert dust with fine mode or a presence of stronger absorption than is assumed by the properties

of coarse non-spherical component. A strong presence of fine absorbing particles in the layer close to the ground is similar to
the other examples analyzed in this study. Generally, such behavior is observed for all the cases with pronounced presence of fine absorbing aerosol type (see Fig. 22).

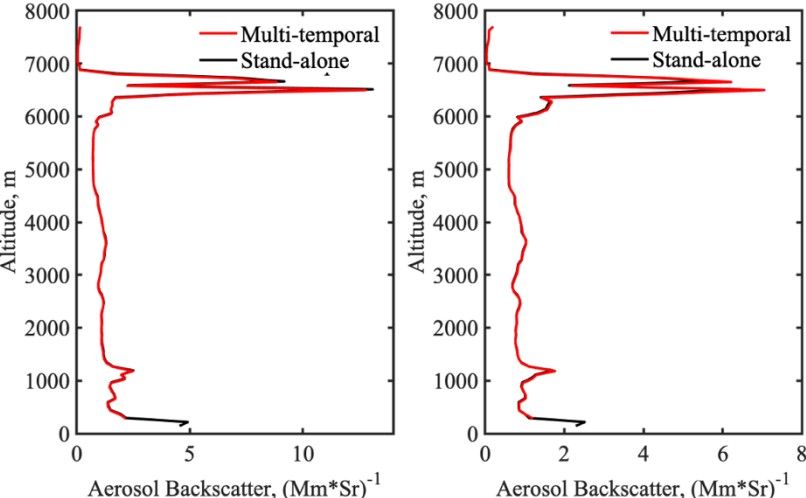

**Figure 24: Aerosol backscatter profiles at 455nm (left) and 940 nm (right) estimated from COBALD profiles acquired on 5 August 2015 using a stand-alone (black) and multi-temporal (red) retrieval schemes.**

The comparison of vertical profiles of extinction, Angström exponent and backscatter at 455 and 940 nm provided by multi-temporal and predefined component methods for the COBALD flights on 11 of August 2015 are shown in Fig. 25–26. The aerosol extinction profile estimated using stand-alone COBALD retrieval demonstrates almost exact coincidence with the profile provided by combined observation approach. At the same time, similarly to the previous example, some significant differences could be observed in the lower part of the extinction profiles at 455 nm below 500 m coming from a different
altitude range of profiles used in both retrievals.

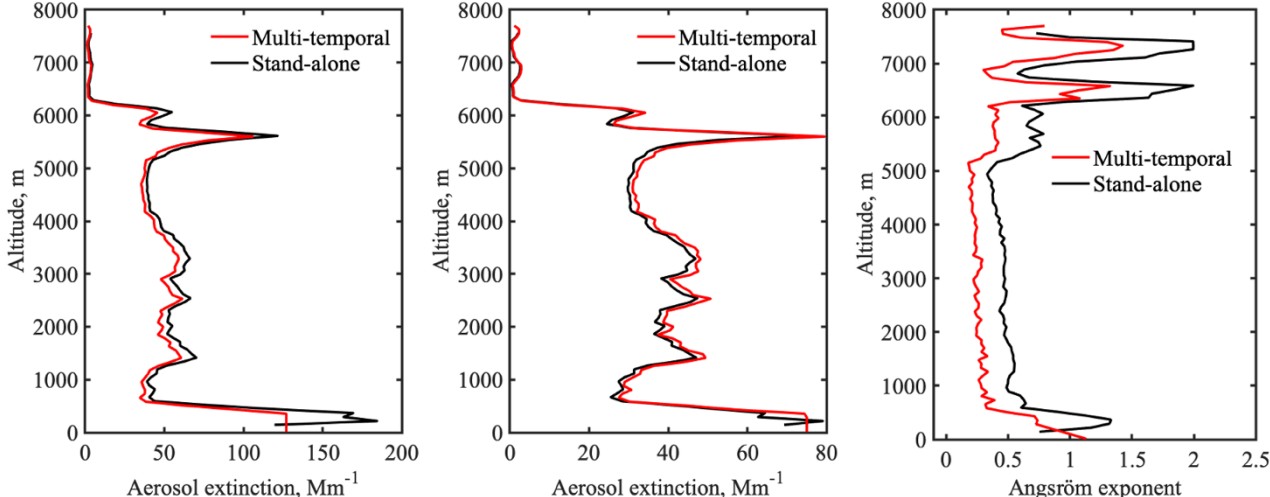

**Figure 25: Aerosol extinction profiles at 455nm (left) and 940 nm (center) and corresponding Angström exponent (left) estimated from COBALD profiles acquired on 11 August 2015 using a stand-alone (black) and multi-temporal (red) retrieval schemes.**

Angström exponent profiles at 455/950 nm estimated from COBALD observation using different retrieval approaches are
shown in Fig. 25. Profiles are generally in agreement with the average difference between the profiles below 0.25. The closest observations of AE 440/870 performed by AERONET give estimations of ~0.35 change overnight, as compared to the values of ~0.3 and ~0.54 averaged below 6000 m for multi-temporal and component approaches correspondingly. Higher values as well as differences could be observed above 6 km, where the concentration of aerosol is negligible and therefore higher AE errors are expected. Higher values of AE at ~6 km also supports the findings that the aerosols at this height consist of fine-
mode transported dust (Parajuli et al., 2020).

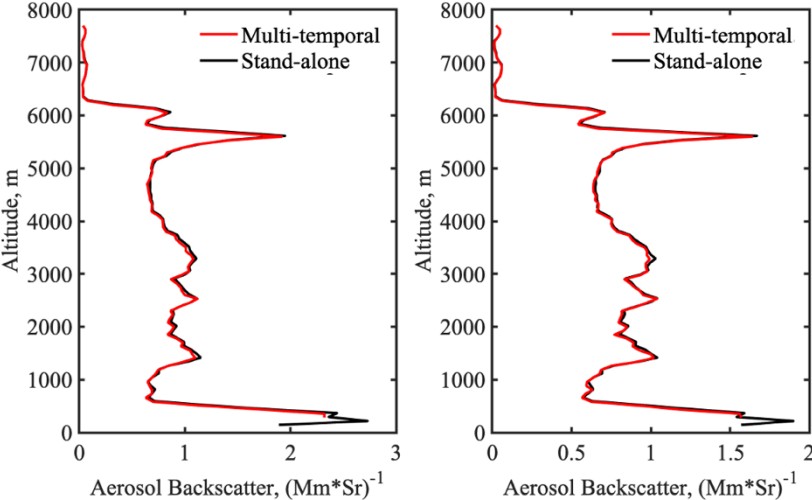

**Figure 26: Aerosol backscatter profiles at 455nm (left) and 940 nm (right) estimated from COBALD profiles acquired on 11 August 2015 using a stand-alone (black) and multi-temporal (red) retrieval schemes.**

The retrieval results from COBALD observations on 12 August 2016 are shown in the left panel of Fig. 22. The aerosol layers below ~6000 m are dominated by coarse non-spherical particles with thick layers close to the ground with pronounced layers at 5000–5500 m and 7000–7500 m. The second most abundant aerosol type is fine absorbing, with close to zero presence of other aerosol components. The shape of vertical profile of fine absorbing particles replicates the shape of the profile of coarse non-spherical below 5000 m. These are most probably the layers of desert dust with absorption or fine mode fraction bigger than the assumptions put to the coarse non-spherical component. A strong presence of coarse non-spherical particles is observed in the layer 5000–5500 m, which is most likely a layer of pure desert dust. It should be noted that in comparison with the combined retrievals presented in Section 3.1.1, only vertical structure of coarse particles demonstrates similar behavior, while no significant layers of fine particles could be observed.

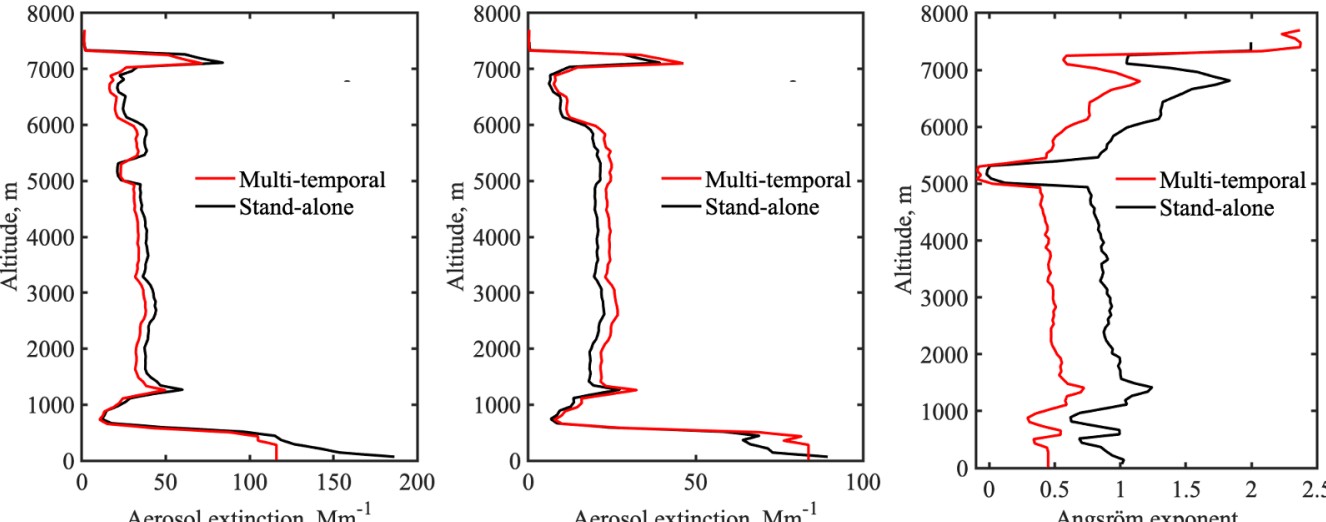

**Figure 27: Aerosol extinction profiles at 455nm (left) and 940 nm (center) and corresponding Angström exponent (left) estimated from COBALD profiles acquired on 12 of August 2016 using a component (black) and multi-temporal retrieval schemes.**

The comparison of extinction and backscatter profiles retrieved using different methodologies are presented in Figs. 27–28. The aerosol extinction profile estimated using stand-alone COBALD retrieval slightly underestimates but remains very close (within 10Mm$^{-1}$) to the profile provided by combined retrieval. Such behavior is most probably related to a limitation of the mixture of aerosol components to reproduce spectral properties of aerosols compared to a more complete model employed in the combined approach. An encouraging agreement between Angström exponent profiles could be observed in Fig. 27, the average difference between the profiles is below 0.4. The closest observations of AE 440/870 performed by AERONET give

estimations of ~0.5–0.7 change overnight, as compared to averaged values of ~0.55 and ~0.9 for multi-temporal and component approaches correspondingly (averaged below 7.5 km).

The fits achieved for all COBALD data observed for presented days presented in Fig. 29 are exceptionally good, with root-mean square error between measure and fitted backscatter below 0.02 (Mm*Sr)$^{-1}$, zero bias and slope equal to 1. It is worth mentioning that compared to combined retrievals presented in Section 3.1.1, component approach allows more flexibility in aerosol vertical structure, proposing a more complex model that has almost twice as many parameters describing aerosol vertical distribution as the one used in combined, which results in generally better fits achieved (compare with Fig. 11).

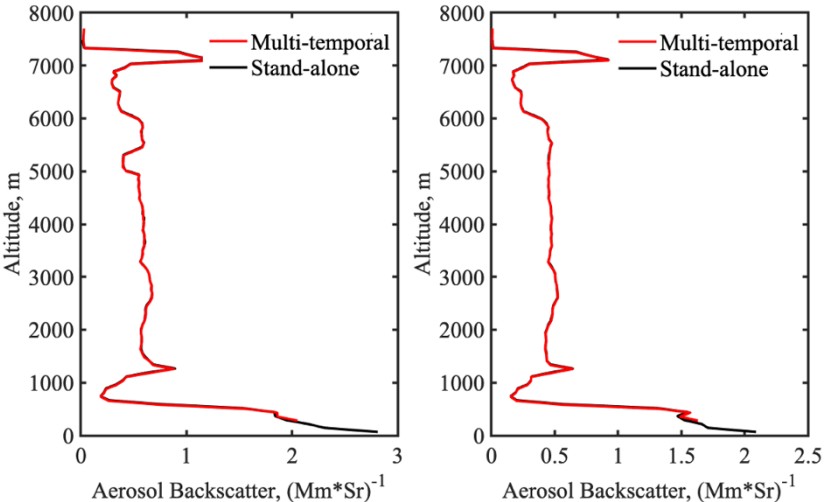

**Figure 28: Aerosol backscatter profiles at 455nm (left) and 940 nm (right) estimated from COBALD profiles acquired on 12 of August 2016 using a stand-alone (black) and multi-temporal (black) retrieval schemes.**

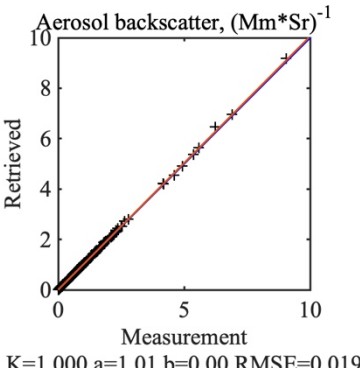

**Figure 29: Achieved vertical profile backscatter fits at 455 and 940 nm for data acquired on 5, 11 of August 2015 and 12 of August 2016.**

Thus, the analyzed cases overall show encouraging stability, disregarding the data set or approach used, both for reproducing observations and retrieval of aerosol optical properties. Several differences observed are either within the estimated accuracy of the retrieval of 10Mm$^{-1}$, or located in the lower atmosphere, where correct comparison is difficult due to the limitations of lidar observations. Also, it should be noted that despite a simplification in modeling of aerosol single scattering properties using component approach the single instrument provided better fits than a more flexible (in terms of columnar properties) approach in the combined retrieval of better constrained observations.

### 3.2.2 Stand-alone LILAS retrievals

LILAS lidar provides significant amount of spectral information including three backscatter, two extinction and one depolarization profiles to perform advanced retrievals, that provide a limited yet descriptive set of aerosol microphysical properties, including effective radius and spectrally independent value of complex refractive index (Veselovskii et al., 2017). Therefore, a nearly comparable performance is expected from LIALS stand-alone retrievals, providing enough constraints to

estimate at least four parameters describing aerosol properties in each layer. In these regards the same set of components as used in COBALD stand-alone retrievals described in Section 3.2.1 was also proposed for LILAS stand-alone retrievals (see Table 7 for details).

**Sensitivity study**

A sensitivity study was conducted in order to evaluate limitations and capabilities of the standalone LILAS retrievals. The sensitivity tests followed the same approach as discussed in Section 3.2.1 but in application to advanced LILAS nighttime observations. A set of concentration profiles of predefined aerosol components (same set of components as used in the retrievals, described in Table 7) was used to simulate the attenuated elastic backscatter profiles at 355 and 1064 nm, attenuated inelastic backscatter profiles at 387 together with vertical profiles of volume depolarization, aerosol extinction and backscatter

at 532 nm, as defined by Eqs. (11–13), (17) and (21), respectively. Generally, the simulated measurements configuration is mimicking one of the real-life nighttime observations described earlier in Section 3.1.2 for retrievals in combination with sun-photometric measurements (see the information for night measurements in Table 4).

Similarly as for COBALD stand-alone retrieval considerations only one sensitivity test will be shown and discussed in details. At the same time, a more complex scenario, to certain extent pushed to the extreme in terms of information content, was

considered. A four-layer aerosol distribution that included a fine absorbing layer close to the ground with three normally distributed overlapping layers above, each containing one of the remaining aerosol components. Three Gaussian distributions with geometrical standard deviations of 500 m were used to model vertical distributions of fine non-absorbing, coarse spherical and non-spherical components with median heights of 4000, 6000 and 8000 m correspondingly. An exponential distribution with the scale height of 1000 m was used to describe the vertical distribution of fine absorbing aerosol. It should be noted,

however, that this used simulation configuration is artificial, complex and is expected to be quite challenging even for advanced lidar retrievals while the sensitivity tests (not shown here) demonstrated outstanding results for simpler cases as the one presented above in sensitivity study performed for COBALD instrument. Such complex configuration was chosen in order to verify the potential of very high capabilities of advanced lidar observation techniques, and to certain extent to identify their advantages over simpler observations with lower information content that cannot characterize such complex scenes correctly.

The total concentrations of each component were selected to provide comparable contributions to the total optical thickness of ~0.25 at 532 nm. The described aerosol configuration shown in Fig. 30 was used to perform forward simulations of the vertical profiles of advanced LILAS observation with simplified geometry describing nadir observation in the range 0–10 km with 100 altitude grid points with a constant resolution of 100 m. Then these profiles were inverted and the profiles retrieved were compared to the ones used in simulation.

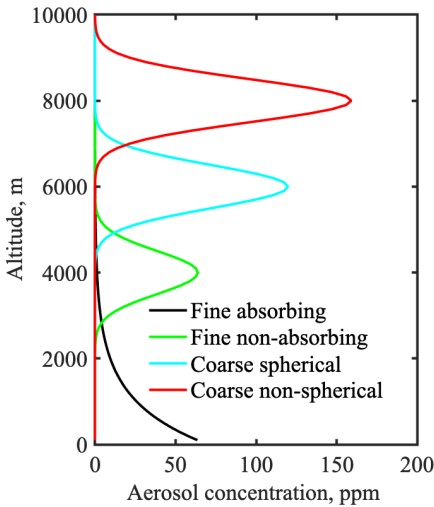


**Figure 30: Aerosol vertical distribution used for the simulation in the LILAS sensitivity test.**

Similar to the test in Section 3.2.1, the retrievals were performed in noise free conditions and with 5%, 10% and 0.5% of random noises added to the attenuated backscatter profiles, profiles of extinction and backscatter and volume depolarization, respectively. The noise levels added are expected to reflect realistic estimates on the quality of the data acquired at night, therefore allowing to estimate the influence of the presence of the random noises on the retrieval results. Figure 31 shows the comparison between modelled and retrieved aerosol concentration profiles for all four components combined in noise free and with noise added conditions (with the noise assumptions at the levels as mentioned above).

The achieved simulated observations fits are comparable to the random noise added to each type of the observation, and are not shown for brevity.

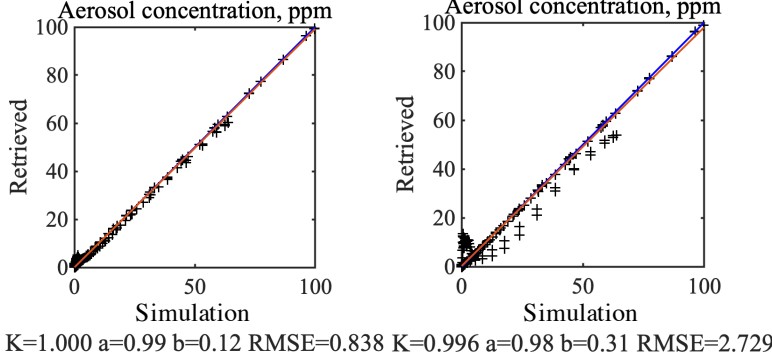

K=1.000 a=0.99 b=0.12 RMSE=0.838 K=0.996 a=0.98 b=0.31 RMSE=2.729

**Figure 31: Comparison between simulated and retrieved aerosol vertical distributions in noise free (left) and noisy (right) conditions.**

Thus, from results shown in Fig. 31 it could be concluded that despite the presence of multiple random noises of realistic magnitudes stand-alone LILAS/GRASP retrievals should provide reliable profiles of total aerosol properties even for sophisticated cases of aerosol mixtures. Notable shift in one of the profiles retrieved, in case of retrievals from data with noise added originates from underestimation of the total concentration of the fine non-absorbing mode. This underestimation most probably relates to the approach of adding noise to the extinction profile. Indeed, the advanced lidar systems provide both extinction and backscattering at the same time. In the present test, the noises added to the extinction and backscattering are absolutely uncorrelated because it is assumed that these two values are measured by different sensors. However, in reality both values depend on the same physical characteristic (e.g. aerosol concentration), i.e. variability of their values naturally correlates and most likely even their errors. Those correlations may have some positive effect in retrieval by canceling out some of measurement noise, while in absence of those correlations the noise effects can only be magnified.

**Application to real data**

The same advanced lidar observation profiles as processed in combination with radiometer in Section 3.1.2 will be analyzed and compared below. The details of the aerosol components vertical distributions for 16 of April 2015 could be found in Figs. 32–34. The achieved vertical profiles fits for elastic and inelastic attenuated backscatter together with aerosol extinction and backscatter profiles are shown in Figs. 35 and 36. As can be seen in Fig. 32 the aerosol below ~3000 m is dominated by coarse non-spherical particles with thick layer close to the ground without any pronounced layers structure. The second most abundant aerosol type is coarse spherical, with close to zero presence of fine aerosol components. Coarse non-spherical particles dominate in the upper layer above 3500 m, indicating aerosol layers free of desert dust. Another noticeable presence of coarse non-spherical particles is located at 2000–3000 m layer. These are likely the layers of desert dust with the properties that differ from the assumptions that were put to the coarse non-spherical component. A layer from 1000 to 2000 m, has also a slight presence of fine absorbing particles, most likely indicating layers with higher absorption. In general, one can conclude that the whole observed atmospheric column is most likely a single well-mixed layer of pure desert dust, which qualitatively corresponds to the results presented in Section 3.1.2, also indicating a well-mixed layer of desert dust particles on 15–16 April 2015.

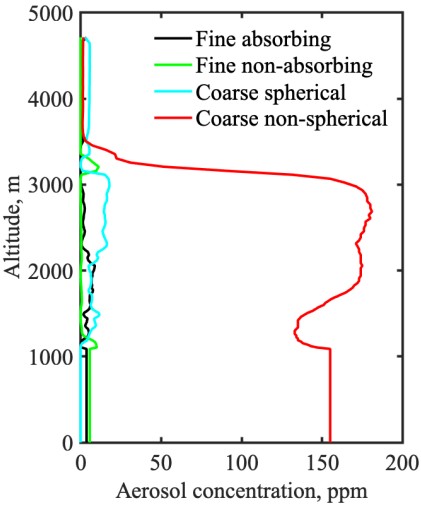

**Figure 32: Aerosol vertical distributions retrieved from stand-alone LILAS observations performed on 16 of April 2015.**

Figure 33 shows a comparison of vertical profiles of aerosol extinction at 355 nm retrieved from LILAS nighttime observation using a multi-temporal approach as described in Section 3.1.2 and direct estimation from inelastic attenuated backscatter using
method proposed by Ansmann et al. (1992). All three derived profiles demonstrate good agreement, disregarding the differences in methodologies that were applied to estimate them. It should be additionally outlined, that in a contrast with vertical extinction at 532 nm, that was used as an input for both combined and single-instrument retrievals measured data at 387 nm were used in a form of inelastic attenuated backscatter, i.e. without deriving extinction profile in advance. This demonstrates the capabilities of processing of such type of advanced observations constraining aerosol extinction, without
applying additional lidar data preprocessing. This detail could also introduce some of the observed differences in the final retrieval results, for e.g. aerosol extinction directly derived from Raman observations demonstrates very low and sometimes negative values upper than 3500 m.

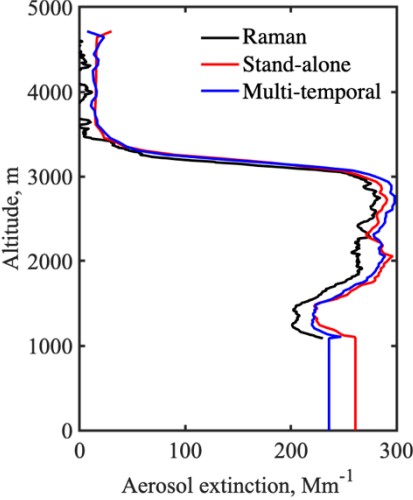

**Figure 33: Vertical profile of aerosol extinction at 355 retrieved from LILAS observations using stand-alone (red) and multi-**
**temporal approaches (blue) performed on 16 of April 2015 in comparison with estimation from 355/387 inelastic backscatter measurements (black).**

A convincing agreement between Angström exponent profiles at 355/532 nm estimated from LILAS nighttime observations using different approaches (multi-temporal, component and direct estimation form inelastic observations) could be observed in Fig. 34, the average difference between the profiles is below 0.2, with the average difference between GRASP provided
profiles below 0.02. The closest observations of AE 380/500 performed by AERONET provide estimations of ~0.12–0.15 overnight, as compared to the averaged values of ~0.2, ~0.18 and ~0 for multi-temporal, component and Raman approaches correspondingly (averaged below 3500 m).

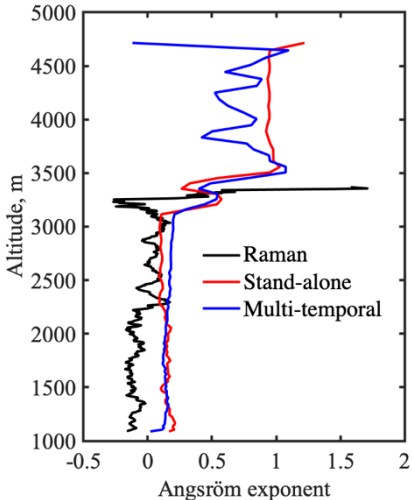

**Figure 34: Vertical profile of Angström exponent at 355/532 nm retrieved from LILAS observations using stand-alone (red) and multi-temporal approaches (blue) performed on 16 of April 2015 in comparison with estimation from 355/532 inelastic backscatter measurements (black).**

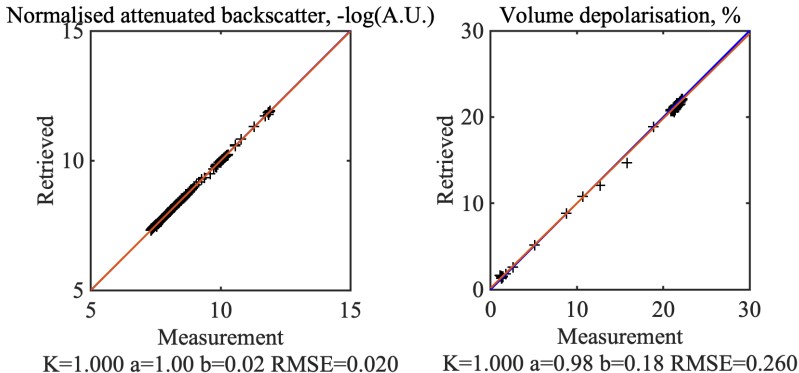

**Figure 35: Achieved fits for attenuated elastic backscatter at 355 and 1064 nm (right) and volume depolarization at 532 nm, data acquired on 16 of April 2015.**

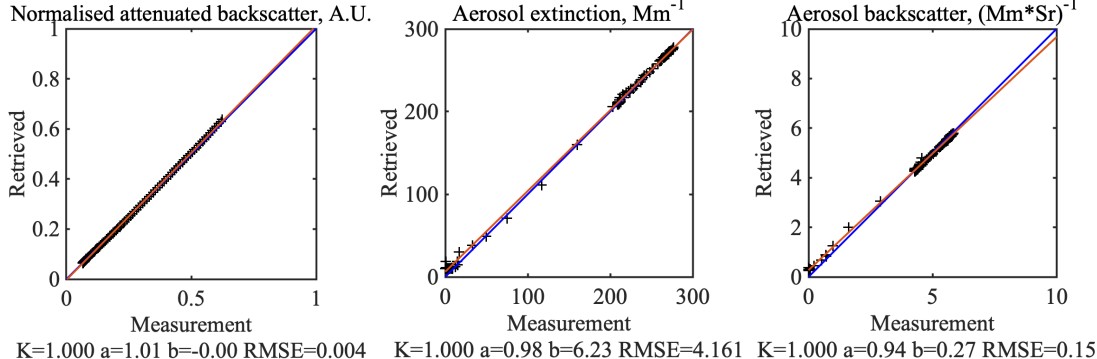

**Figure 36: Achieved fits for attenuated inelastic backscatter at 387 (left) aerosol extinction at 532 nm (middle) and aerosol backscatter at 532 nm (right), data acquired on 16 of April 2015.**

The fits achieved for all LILAS data observed at night on 16 April 2015 presented in Figs. 35 and 36 are exceptionally good. Both observations available during the whole observation period (i.e. evening, night and morning) are shown in Fig. 35 and advanced ones available only during nighttime are shown in Fig. 36. The elastic backscatter fits for 355 and 1064 nm channels combined and shown in log scale also demonstrate good fit, as well as volume depolarization at 532 nm: almost perfect slopes with the smallest value of 0.94, low RMSEs and absence of notable bias characterize all fits, including values of aerosol extinction and backscatter at 532 nm (see Fig. 36). In contrast with combined retrievals, stand-alone LILAS retrievals using predefined components approach demonstrate much better volume depolarization fits (see Fig. 16). As discussed above additional freedom in describing vertical properties of aerosol, to be precise, the ability to vary the effective sphericity of the

particles by mixing spherical and non-spherical components with separate vertical distributions, provided sufficient flexibility to perfectly fit the low values of volume depolarization data, additionally improving the fits of other observations (compare Figs. 35 and 16).

Thus, this section demonstrated a methodology to process single instrument vertically resolved data. The retrieval was demonstrated on both synthetic and real data to be efficient for achieving reliable retrievals of aerosol properties, allowing consistent and stable retrievals from processing stand-alone advanced lidar or radiosonde observations.

It should be noted that in all demonstrations the aerosol was modeled as an external mixture with pre-determined sizes, shapes and compositions. At the same time, it is one of many possibilities of applying GRASP to single instruments retrievals and other assumptions and constraints can be used. For example, aerosol mixture can only be represented by only one or two components in case of single or dual wavelength lidar and many other assumption settings can be used.

## 4 Conclusions

This paper discussed in details the evolution of GARRLiC/GRASP lidar/radiometer combined retrieval approach by Lopatin et al. (2013) and demonstrated new possibilities realized in GRASP algorithm for synergetic aerosol retrievals from various combinations of passive ground-based sun-photometric measurements with co-located ground-based lidar or airborne vertically resolved observations. The three following updates of GRASP were introduced and demonstrated in the paper:

1. The new possibilities of processing advanced vertically resolved observations as those provided by advanced lidars including Raman and other lidar systems providing information about backscattering and extinction profiles independently, the lidar system with polarimetric capabilities measuring returned signal depolarization, as well as in situ observations, as those from backscatter sondes, providing backscatter and other measurements at different altitudes;

2. A multi-temporal approach to invert simultaneously a set of diverse passive and/or active ground-based observations or in situ observations that are not necessarily co-incident. For example, records of both day and night passive and active observations can be inverted simultaneously in the frame of GRASP multi-pixel approach (Dubovik et al., 2011) using a priori constraints on the temporal or/and spatial variability of retrieved parameters;

3. A possibility to process vertically resolved data from a single instrument such as a lidar or backscatter sonde.

The above new functionalities were achieved relying on the improvements in forward model by extending the capabilities of simulating new types of observations and in numerical inversion part by adapting the multi-pixel retrieval concept to new types of ground-based and in situ observations.

The new functionalities of GRASP were demonstrated for a synergy processing of the ground-based sun-photometric and advanced lidar observations including Raman and depolarization data and data of the sun-photometric and micro-pulse lidar combined with in situ radiosonde backscatter observations collected during SHADOW and KAUST field campaigns in 2015 and 2016. The multi-temporal approach has shown to be rather efficient for combining non coincident but close in time such as diverse day- and night- observations during SHADOW and KAUST field campaigns. As a result, the observations obtained by different techniques can be inverted together and provide a complete set of fully consistent aerosol parameters including detailed size distributions, complex refractive indices, information about particle shape and vertical profiles. This provides a unique tool for combining various coordinated observations that contain information that is clearly complementary but not evident for straightforward fusion.

Secondly, a methodology to model aerosol single scattering properties that considers aerosol as an external mixture of several aerosol components with predetermined sizes, shapes and compositions has been described and demonstrated to be efficient for achieving reliable retrievals of aerosol properties in several situations. Specifically, the approach allows consistent and

stable retrievals of aerosol from processing stand-alone advanced lidar or radiosonde observations by reducing information content about aerosol columnar properties.

The new possibilities of processing vertically resolved observations from a single instrument was illustrated by processing of the observation from lidar systems with Raman and depolarization capabilities and in situ backscatter sonde. The single instrument retrievals were relying on the concept considering aerosol as an external mixture of several aerosol components with predetermined sizes, shapes and compositions. This model has been described and demonstrated to be efficient for achieving reliable retrievals of aerosol properties in several situations. Both numerical tests and applications to the data collected during SHADOW and KAUST field campaigns have demonstrated reliable quality of GRASP single instrument inversion results.

In general, both approaches discussed in this study, i.e. the advanced combination of diverse observations and single instrument retrieval using additional constraints in the forward model, help to address significant limitations in processing of single- and multi-instrument observations, allowing to exploit the most of the sensitivities of the instrumentation available. Generally speaking, both approaches rely on using additional constraints on aerosol columnar properties, either by including the observations with missing sensitivities or by employing direct a priori assumptions.

The shown new functionalities were achieved relying on the improvements in forward model by extending the capabilities in simulating new types of observations and in numerical inversion part by adapting the multi-temporal retrieval concept to new types of ground-based and in situ observations. Since, the inversion and forward model are practically independent modules of GRASP, the number of new retrieval possibilities is not limited by the examples demonstrated here. This altogether makes GRASP a very flexible software with exceptional possibilities of data treatment provided by various types of remote sensing and in situ observations that could be applied with very limited efforts to nearly arbitrary combinations of observations available. Therefore, GRASP offers significantly extended possibilities for processing observations collected during field campaigns obtaining compact records of diverse passive and active observations.

**Code availability.** GRASP is an open-source software, available upon registration from https://access-request.grasp-cloud.com/service/gitlab (last access: 25 January 2021).

**Data availability.** Data available upon request. AERONET data used for this study can be downloaded from https://aeronet.gsfc.nasa.gov/ (last access: 25 January 2021). MPL data used for this study can be downloaded from https://mplnet.gsfc.nasa.gov (last access: 25 January 2021). COBALD and LILAS data are available upon request.

**Author contribution.** OD contributed to the development of the overall algorithm methodology, research planning and article writing; GS participated in organisation of field campaigns, data acquisition, in the analysis and discussion of the results; TL, DF, QH and AL contributed in realization of the algorithms and different parts of supporting retrieval software; IV and FW contributed to the development of the instruments and their data preparation and analysis; IS contributed to the maintenance of the instruments and data acquisition; SP contributed to the retrieval data preparation and analysis, and the polishing article text; AL prepared the manuscript with contributions from all co-authors and conducted main part of the presented research.

**Competing interests.** The authors declare that they have no conflict of interest.

**Acknowledgments.** The research reported in this publication was partially supported by funding from King Abdullah University of Science and Technology (KAUST) through the Competitive Research Grant (URF/1/2180-01-01) "Combined Radiative and Air Quality Effects of Anthropogenic Air Pollution and Dust over the Arabian Peninsula". We are also thankful to KAUST Core Lab and IT department for supporting the data collection and archiving. Authors also pay the tribute to

1445 exhausting and, sometimes, sleepless labor of the teams that make operation of field campaigns possible, notably the teams of AERONET, MPL, IRD (Institut de Recherche pour le Développement) in Dakar and LOA (Laboratoire d'Optique Atmosphérique) in Lille for SHADOW campaigns and Jean-Paul Vernier for the operation of radiosonde flights during KAUST campaigns. O. Dubovik and T. Lapyonok were supported by the Labex CaPPA project, which is funded by the French National Research Agency under the contract "ANR-11-LABX0005-01".

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
