# Peer review of "Synergy processing of diverse ground-based remote sensing and in situ data using GRASP algorithm: applications to radiometer, lidar and radiosonde observations"

_Atmospheric Measurement Techniques, 2020_

## Referee Comment (RC1) · Lorraine Remer (Referee) · 9 Dec 2020

This paper is a comprehensive report of taking GRASP aerosol retrievals of lidar and lidar-like measurements to the next level. It had been previously established that GRASP retrieval software could be adapted to retrieve vertical aerosol properties when applied to simultaneous measurements of lidar and AERONET. These previous retrievals used the total column angular sky-scattering measurements and the total aerosol optical depth from AERONET to constrain the vertically-resolved backscattering lidar measurements. The present paper shows two things: 1) AERONET and lidar

do not need to be coincident in time to produce viable retrievals. This allows nighttime lidar measurements to be inverted together with day time AERONET measurements. 2) There is sufficient information in the vertically-resolved measurements of spectral backscattering alone to produce reasonable retrievals in a simplified GRASP inversion. Furthermore, if the lidar is more complex with measurements of profiles of multi-wavelength backscattering and extinction with depolarization, and this is adequate to resolve complex aerosol layering in the atmosphere, even without benefit of AERONET measurements for constraint.

My overall assessment is that the study is worthwhile, even though i have concerns, and the paper eventually publishable. I have no need to remain anonymous. This is Lorraine Remer writing.

Length, organization and readability: Right now the manuscript is very long with over 40 figures. Each case is presented methodically, with repetitive description elements. This systematic approach in some ways keeps the presentation consistent and clear but reading it does become tedious. I'm not going to insist on a re-organization, but I am going to suggest considering alternative presentation ideas. Perhaps combining the plots of each case into a single 4-panel figure, so that you have one plot per case study: Figures 1,2 and 3 become a 4-panel figure 1; Figures 16 and 17 become another 4-panel figure; Figures 24, 25 and 26 become a 4-panel figure, and then maybe you don't also need Figure 27 and it can eliminated it. And so on.

But overall the writing is satisfactory, mostly the descriptions are clear and the English is fine. The figures are clear and informative.

Questions and concerns: 1. There is an underlying assumption necessary to use evening and morning (daylight) AERONET measurements with the middle-of-the-night COBALD measurements. That is, there is no change in aerosol properties from evening to morning (line 524). Then I look at Figure 1. How can that assumption be justified when the evening and morning AERONET size distributions have shifted a

couple of microns to coarser particles over night? Not only has a dust event arrived over night, but the particles are different sizes. Do the refractive indices change also? Why aren't the AERONET values shown in Figure 3? Something similar happens in Figure 4. Maybe also in Figure 7. Not so much in Figure 13.

Exactly which parameters are constrained and which are free parameters to be retrieved? Is it just a temporal smoothing? I just don't see the assumption holding in some of these cases given the comparison of size distributions with AERONET.

I am not convinced that the assumption holds. It also bothers me that all three size distributions that include vertical profiles look alike in Figure 1, but do not look like either AERONET size distribution retrievals. The authors assume that the retrievals are accurately retrieving the aerosol properties and then describe the figures as though the "evening", "night" and "morning" retrieved properties are describing changes in the real aerosol that is happening overnight (lines 641 – 646). To me, I see no validation of this "multi-pixel" technique applied to these data when vertical profile measurements are not coincident with AERONET.

2. Is there MPL at night? Wouldn't it make sense to take baby steps? Joint retrievals of MPL and AERONET during the day. Non-simultaneous retrievals of MPL at night with AERONET morning and evening. Maybe at several times during the night. I would show this before I jumped in to do a non-simultaneous retrieval with a new instrument-data type (COBALD) that I could not compare with a simultaneous joint retrieval.

3. Figure 2, Figure 5, etc. Is there no lidar during night? Does it make sense to compare the COBALD multi-instrument retrieval with the same time lidar backscattering? That would help to validate what is going on here. Maybe the lidar at night would pick up that high altitude layer that the evening and morning miss. Right now it looks to me like a retrieval artifact.

4. Why do the size distributions in Figure 1 created with vertical profiles (MPL and COBALD both) look the same, but that AERONET alone does not? What is the vertical profile information doing to the retrieval that accentuates the coarser range of the coarse mode and diminishes the finer range of the coarse mode?

5. In the end, I am much more comfortable with the single instrument retrievals than I am with the non-simultaneous multi-instrument retrievals. It is somewhat surprising how well the unconstrained COBALD measurements perform, but not so surprising about the complex lidar. Still by combining instruments at different times you could be introducing more uncertainty than it is worth.

6. Still, all of the actual examples are in dust-dominated regimes. Before the authors conclude that all is well, they need a paragraph in their conclusion expressing that fine-mode dominated aerosols or multiple layers with different aerosols could pose problems. This is especially so since the retrieval was set up to constrain real and imaginary parts of the refractive index to be the same in the fine and coarse modes.

7. Table 2 needs to be more informative somehow. It needs to clarify which data sets are combined in retrieval to create what is termed "evening", "night" and "morning" in the figures. For example, "night" involves both COBALD and the AERONET measurements. That is unclear in Table 2. This is really important because the paper is so long it will be read in multiple sittings. So by the time somebody is looking at the figures and results, they have forgotten how the retrievals are set up. There needs to be an easily referenceable table to pull it all together.

Minor comments

Page 2. Need references to describe the advanced lidars

Page 2. Need references to describe the "blind zone".

Line 118. Contraction. "didn't" and other places. Usually in formal journal articles we do not use contractions, but that is a stylistics thing with the journal. I use contractions all the time.

Line 249. "component" should be plural "components"

Lines 274-275. BRDF and BPDF need to be written out and/or defined

Line 276. Make sure Dubovik et al. 2020 is in the reference list. Right now I believe the citation refers to Dubovik et al. 2019 in the list.

Line 307. Insert "the" into "of THE retrieval"

Line 329. "could defined" needs to be either "could be defined" or "could define"

Line 471. "visible-IR" should be "visible-NIR", but NIR needs to be defined some place as near infrared.

Line 471. "daily perform" should be "perform daily"

Line 473. Besides fixed elevation angle, if you include principal plane inversions, then you have to add "fixed azimuth angle"

Lines 630 – 633. I did not understand. Try saying, "Not shown here, but . . ." What is meant by "GRASP retrievals of only almucantar and TOD data do not demonstrate such difference"? Isn't this what the AERONET green lines in the figure are? Is there something different between "GRASP retrievals of AERONET measurements" and AERONET retrievals as displayed in the figures?

Line 736. Strongly should be stronger.

Lines 912- - 915. I'm still unconvinced.

Line 1063. Figures range should be 24-36, not 23-35.

Line 1172 – 1173. Maybe because the multi-instrument retrieval introduces noise because assumptions are not being met.

Figure 38b. how much noise was introduced in the panel shown?

Lines 1235 – 1238. I think the authors mean "coarse spherical" not "coarse non-spherical" when they discuss the second most abundant aerosol type.

---

## Referee Comment (RC2) · Feng Xu (Referee) · 9 Dec 2020

The paper by Lopatin et al. employed the GRASP algorithm to investigate the benefits of combining multiple ground-based observations (namely sun-photometric, lidar and radiosonde observations) to constrain aerosol retrievals in terms of their vertical concentration distribution, refractive index, size distribution and spherical particle fraction. With the assumption of temporal continuity of aerosol properties, the synergetic retrieval mitigates the insufficient information content in lidar only night-time retrieval and brings up the retrieval accuracy. The full-length paper covers a large amount of

valuable work on algorithm development, test and validation. My comments below are mainly to suggest clarifications on some implementation details:

1. Section 3.1.1 discusses the multi-temporal retrieval of combined COBALD, AERONET and MPL observations. Regarding the balloon-borne COBALD data, does the retrieval underlying Figs.1-3 use a series of COBALD temporal measurement during the night of Aug 05, 2015 ? If multiple COBALD measurements are inverted, then it will be helpful to add a time series plot showing the temporal evolution of certain retrieval aerosol quantity (e.g. fine aerosol concentration or other properties). Following the use of multiple COBALD measurement (if this is true), does the "Fine night" and "Coarse night" in Figs. 1-3 averages the retrievals of night measurements from 13:21UTC to 05:16UTC of next day ?

2. In some figures (e.g. Figs. 25-31), I saw terms "multi-pixel" in legend, but "multi-temporal" in figure caption. To clarify, did the balloon-borne COBALD measurement resolve both multiple pixel and multiple-temporal measurements so that smoothness constraints were imposed in both dimensions ?

3. Table 2: it might be helpful to add another column providing the measurement uncertainties for MPL, COBALD and sun-photometer which are used in retrieval.

4. In Section 3.1.1, demonstration of retrieval results were provided for refractive index, size distribution and vertical profiles of aerosol concentration. How about the spherical aerosol fraction ? Is it also retrieved retrieved and worthwhile to demonstrate ?

5. Section 3.2.1 and Section 3.2.2, do the stand-alone COBALD and LILAS retrieval involve the use of any multi-temporal/mullti-pixel constraints ? Please clarify.

6. Figs. 25, 26, 27, 29, 30 and 31, it took me a while to confirm the meaning of "components" in the figure legends. For clarify, authors might mention it again in Fig. 25 caption that "components retrieval" here mean "stand-alone COBALD" (or "stand-alone COBALD" retrieval) with a) turning off temporal constraint and b) using pre-determined

size distribution for all aerosol types (Table 5), while "multi-pixel retrieval" means a) using all types of instrumental data and b) imposing temporal constraint in retrieval.

7. As the authors pointed out, Fig. 29 indicates "some significant differences ... in the lower part of the extinction profiles at 455 nm below 500 m." Could this be due to the impact of measurement uncertainties, or neglecting the multiple scattering in the model, or others ? Is there any constraints imposed on the vertical variations of aerosol concentration or properties ? If so, maybe it's worthwhile to try relaxing the constraint and see if one can observe more consistency in the two types of retrievals.

---

## Author Comment (AC1) · 28 Jan 2021

**Reply to reviewer 1 (Lorraine Remer).**

Authors are grateful for the valuable comments that help to improve the paper. In the revised version of manuscript, we tried to address all the questions raised. The detailed responses and clarifications are given below.

Overall comments: This paper is a comprehensive report of taking GRASP aerosol retrievals of lidar and lidar-like measurements to the next level. It had been previously established that GRASP retrieval software could be adapted to retrieve vertical aerosol properties when applied to simultaneous measurements of lidar and AERONET. These previous retrievals used the total column angular sky-scattering measurements and the total aerosol optical depth from AERONET to constrain the vertically-resolved backscatter- ing lidar measurements. The present paper shows two things: 1) AERONET and lidar do not need to be coincident in time to produce viable retrievals. This allows nighttime lidar measurements to be inverted together with day time AERONET measurements. 2) There is sufficient information in the vertically-resolved measurements of spectral backscattering alone to produce reasonable retrievals in a simplified GRASP inver- sion. Furthermore, if the lidar is more complex with measurements of profiles of multi- wavelength backscattering and extinction with depolarization, and this is adequate to resolve complex aerosol layering in the atmosphere, even without benefit of AERONET measurements for constraint.

My overall assessment is that the study is worthwhile, even though i have concerns, and the paper eventually publishable. I have no need to remain anonymous. This is Lorraine Remer writing.

Reply: We are thankful for overall rather positive evaluation of our paper. The replies to the concerns are provided in point-by-point responses.

**The point-by-point responses:**

Comment 1: Length, organization and readability: Right now the manuscript is very long with over 40 figures. Each case is presented methodically, with repetitive description elements. This systematic approach in some ways keeps the presentation consistent and clear but reading it does become tedious. I'm not going to insist on a re-organization, but I am going to suggest considering alternative presentation ideas. Perhaps combining the plots of each case into a single 4-panel figure, so that you have one plot per case study: Figures 1,2 and 3 become a 4-panel figure 1; Figures 16 and 17 become another 4-panel figure; Figures 24, 25 and 26 become a 4-panel figure, and then maybe you don't also need Figure 27 and it can eliminated it. And so on.

Reply: Number of figures was reduced. Figures were grouped by retrieved parameters, with different panels corresponding to different retrieval dates in section 3.1.1 and 3.2.1. Additionally, figures containing vertical profile of extinction were combined with figures showing the corresponding Angström exponent. This allowed us to exclude 10 figures. After extension with extra materials, total number of figures does not surpass 36.

Comment 2: There is an underlying assumption necessary to use evening and morning (daylight) AERONET measurements with the middle-of-the-night COBALD measurements. That is, there is no change in aerosol properties from evening to morning (line 524). Then I look at Figure 1. How can that assumption be justified when the evening and morning AERONET size distributions have shifted a couple of microns to coarser particles over night? Not only has a dust event arrived over night, but the particles are different sizes.

Reply: We realized that that our description was quite unclear and therefore misleading. In fact, it is assumed that change between different observations exists, but it is smooth, i.e. provides a smooth transition from evening to morning overnight without a sharp and drastic change in aerosol columnar properties (this includes volume size distribution, complex refractive index and sphericity fraction). The only way the values constrained is the application of a priori smoothness constraints on variability of each aerosol parameter between evening, night and morning retrievals. Neither direct assumptions or constraints on values of aerosol properties nor assumption of them being constant in time are applied. The described change observed by AERONET should comply with the proposed temporal smoothing assumptions. In order to make this clear and demonstrate the effect of such constraints a specific subsection 3.1.1.1 describing a dedicated sensitivity study was added.

Comment 3: Do the refractive indices change also? Why aren't the AERONET values shown in Figure 3? Something similar happens in Figure 4. Maybe also in Figure 7. Not so much in Figure 13.

Reply: Refractive indices provided by AERONET were added to figures 6, 7 and 14. The following discussions on comparison of these parameters with AERONET retrieved values were added:

- lines 734–737: "All retrievals including AERONET shown in the left panel of Fig. 7 demonstrate a notable decrease of imaginary part with wavelength. The corresponding single scattering albedos are shown in Fig. 8, and demonstrate acceptable agreement, with both retrievals indicating no significant change of these parameters overnight."

- lines 775–779: "AERONET retrievals performed at the same time demonstrate stronger variability of this parameter overnight, meanwhile a more obvious variability could be seen in the values of SSA (central panel of Fig. 8) provided by the multi-temporal retrieval, with both imaginary part of refractive index and SSA values provided by multi-temporal retrieval located in between the evening and morning AERONET retrievals."

- lines 800–803: "AERONET retrievals performed at the same time demonstrate stronger variability of imaginary part of refractive index overnight together with SSA retrievals (right panel of Fig. 8), with both imaginary part of refractive index and SSA values provided by multi-temporal retrieval located in between the evening and morning AERONET retrievals, being generally closer to the estimations from morning observations."

- lines 959–964: "Similar temporal variation of the real part of refractive index was observed by AERONET, with the values of multi-temporal retrievals demonstrating comparable variability. At the same time a more significant differences could be observed in values of imaginary part (central panel of Fig. 14), which meanwhile cause a less dramatic change in the estimations of single scattering albedos provided by two retrievals (left part of Fig. 14), both indicating no change of these parameters overnight."

Additionally, to further expand the refractive index retrievals, figures showing retrievals of Single scattering Albedo and their comparison with AERONET were added at figure 8 and 14.

Comment 4: Exactly which parameters are constrained and which are free parameters to be retrieved? Is it just a temporal smoothing? I just don't see the assumption holding in some of these cases given the comparison of size distributions with AERONET.

Reply: All the parameters are retrieved for evening, night and morning with a priori constraints applied on variability of columnar aerosol properties, including size distribution, complex refractive index and sphericity fraction. There are no direct constraints applied on the values of the parameters. The detailed description of the constraints applied have been added to Table 1 and 7. Also changes were introduced to Table 1 to clarify that every set of parameters is retrieved for evening, night and morning.

Based on the results of the sensitivity tests and practical experience we are concluding that observed differences are mainly related to the need of the retrieval to fit additional measurements, as those from radiosonde or lidar, while the applied temporal constraints generally do not contradict AERONET observations while uses the trends imposed by AERONET data.

Comment 5: I am not convinced that the assumption holds. It also bothers me that all three size distributions that include vertical profiles look alike in Figure 1, but do not look like either AERONET size distribution retrievals. The authors assume that the retrievals are accurately retrieving the aerosol properties and then describe the figures as though the "evening", "night" and "morning" retrieved properties are describing changes in the real aerosol that is happening overnight (lines 641 – 646). To me, I see no validation of this "multi-pixel" technique applied to these data when vertical profile measurements are not coincident with AERONET.

Reply: An additional section of sensitivity tests (Section 3.1.1.1) was added to clarify this issue. The results of the sensitivity tests demonstrate the impact on the retrievals from temporal constraints). It was shown how the multi-temporal approach can capture the changes in the aerosol properties, even if they change significantly overnight. Both, the strength and limitations of the approach were demonstrated.

Comment 6: Is there MPL at night? Wouldn't it make sense to take baby steps? Joint retrievals of MPL and AERONET during the day. Non-simultaneous retrievals of MPL at night with AERONET morning and evening. Maybe at several times during the night. I would show this before I jumped in to do a non-simultaneous retrieval with a new instrument- data type (COBALD) that I could not compare with a simultaneous joint retrieval.

Reply: Results of multi-temporal MPL+AERONET retrievals were previously published by Parajuli et al., 2020. This fact was additionally emphasized in the text:

- lines 729–733: "It should be noted that a similar behavior of nighttime high-altitude layers was also observed on a regular basis during summer time in a previous study by Parajuli et al., (2020). The authors analyzed two years of MPL data retrieved at night with a similar multi-temporal approach at KAUST site, and concluded that the high altitude dust was possibly associated with long-range transport of dust from distant sources."

Comment 7: Figure 2, Figure 5, etc. Is there no lidar during night? Does it make sense to compare the COBALD multi-instrument retrieval with the same time lidar backscattering? That would help to validate what is going on here. Maybe the lidar at night would pick up that high altitude layer that the evening and morning miss. Right now it looks to me like a retrieval artifact.

Reply: The similar artifact was observed on multiple days by Parajuli et al, 2020, a study that uses similar multi-temporal approach but in a combination with MPL instead of COBALD data at nighttime. This fact was emphasized in the text:

- lines 729–733: "It should be noted that a similar behavior of nighttime high-altitude layers was also observed on a regular basis during summer time in a previous study by Parajuli et al., (2020). The authors analyzed two years of MPL data retrieved at night with a similar multi-temporal approach at KAUST site, and concluded that the high-altitude dust was possibly associated with long-range transport of dust from distant sources."

It also should be outlined that direct comparison of COBALD and MPL retrieved profiles is not possible, due to the limitation in minimum and maximum altitudes of MPL sounding, and due to the fact that balloon carrying COBALD instrument may significantly divert from the atmospheric volume observed by lidar. Possibility of this and similar artifacts to be related to cloud contamination was also discussed in Parajuli et al., 2020.

Comment 8: Why do the size distributions in Figure 1 created with vertical profiles (MPL and COBALD both) look the same, but that AERONET alone does not? What is the vertical profile information doing to the retrieval that accentuates the coarser range of the coarse mode and diminishes the finer range of the coarse mode?

Reply: The authors assume that these differences are introduced when algorithm needs to fit simultaneously additional data (vertical backscatter, attenuated backscatter and volume depolarization). In fact, it should be noticed that retrievals which do not fit vertical profiles and fit only radiometer data show much closer agreement with AERONET size distributions. The temporal constraints may also induce some similarities in the shapes of size distributions retrieved for different time periods but only those that do not contradict observations.

The following clarifications were added in the text:

- lines 710–716: "At the same time, it should be mentioned that to understand better the reason of such significant differences between results of GRASP multi-temporal and AERONET standard retrieval, GRASP inversion of almucantar + TOD only data was performed, as well as MPL/AERONET retrievals for evening and morning data, both with no use of temporal constraints. These retrievals (not shown here for keeping size of the paper reasonable) demonstrated a significantly better agreement in volume size distributions of GRASP almucantar + TOD results with standard AERONET results, allowing one to conclude that changes observed above are caused mainly by the inclusion of lidar observations."

Comment 9: In the end, I am much more comfortable with the single instrument retrievals than I am with the non-simultaneous multi-instrument retrievals. It is somewhat surprising how well the unconstrained COBALD measurements perform, but not so surprising about the complex lidar. Still by combining instruments at different times you could be introducing more uncertainty than it is worth.

Reply: It should be noted that these retrievals are also constrained: the profiles retrieved are considered to be vertically smooth, Table 7 was extended to make this fact clearer. Authors admit that without additional constraints meaningful retrievals would not be possible since the problem in such formulation is ill posed. Additionally, authors emphasize in the text that only optical properties are expected to be retrieved from stand-alone COBALD retrievals, while the component and microphysical interpretation of the results should be done with extreme caution.

Comment 10: Still, all of the actual examples are in dust-dominated regimes. Before the authors conclude that all is well, they need a paragraph in their conclusion expressing that fine-mode dominated aerosols or multiple layers with different aerosols could pose problems. This is especially so since the retrieval was set up to constrain real and imaginary parts of the refractive index to be the same in the fine and coarse modes.

Reply: The possibility and limitations to distinguish 2 aerosol refractive indices are demonstrated in the sensitivity study added in Section 3.1.1.1. At the same time such separation could be correctly performed in the situations with sufficient load of both aerosol modes that make distinguishable contributions to the measurements. The studied real cases are characterized by coarse mode domination, and do not demonstrate any sign of substantial presence of fine mode of another origin. Due to this reason, in order to constrain the retrievals, they were performed under assumption of the same refractive index for both fine and coarse mode. Additional clarifications were added to the text

- lines 673–675: "Moreover, the extra retrieval test for the similar scenario as in Fig. 3 showed that using an extra assumption of common complex refractive index for fine and coarse aerosol modes may significantly improve the retrieval in the cases when aerosol is dominated by one of the modes."

Comment 11: Table 2 needs to be more informative somehow. It needs to clarify which data sets are combined in retrieval to create what is termed "evening", "night" and "morning" in the figures. For example, "night" involves both COBALD and the AERONET measure- ments. That is unclear in Table 2. This is really important because the paper is so long it will be read in multiple sittings. So by the time somebody is looking at the figures and results, they have forgotten how the retrievals are set up. There needs to be an easily referenceable table to pull it all together.

Reply: Table 2 was extended to make it more clear which combinations of the data were used in different diurnal periods.

**Minor comments**

Comment: Page 2. Need references to describe the advanced lidars Page 2. Need references to describe the "blind zone".

Reply: Reference to a book chapter by Wandinger, 2005, summarizing Raman lidar techniques was added. Reference to Freudentaler et al., 2018 describing in detail technical limitation of lidar observation including the overlap was added.

Comment: Line 118. Contraction. "didn't" and other places. Usually in formal journal articles we do not use contractions, but that is a stylistics thing with the journal. I use contractions all the time.

Reply: the text was changed to the "did not".

Comment: Line 249. "component" should be plural "components"

Reply: The text was replaced with "where components are characterized"

Comment: Lines 274-275. BRDF and BPDF need to be written out and/or defined

Reply: the definitions were added "with surface bidirectional reflectance distribution function (BRDF) and bidirectional polarization distribution function (BPDF)."

Comment: Line 276. Make sure Dubovik et al. 2020 is in the reference list. Right now I believe the citation refers to Dubovik et al. 2019 in the list.

Reply: references was added "Dubovik, O., et al.: Multi-term LSM for applying multiple a priori constraints in problems of atmospheric remote sensing: GRASP algorithm - concept and applications, in preparation, 2021."

References were additionally sorted by year within the same first author to make it easier to find

Comment: Line 307. Insert "the" into "of THE retrieval"

Reply: The text was changed to "of the retrieval"

Comment: Line 329. "could defined" needs to be either "could be defined" or "could define"

Reply: Text changed to "could be defined"

Comment: Line 471. "visible-IR" should be "visible-NIR", but NIR needs to be defined some place as near infrared.

Reply: Text was changed to "visible and near infrared"

Comment: Line 471. "daily perform" should be "perform daily"

Reply: text was changed to "perform daily"

Comment: Line 473. Besides fixed elevation angle, if you include principal plane inversions, then you have to add "fixed azimuth angle"

Reply: The phrase was added: "measurements at fixed solar elevations (almucantars) or azimuth angles (principal plane) during the day"

Comment: Lines 630 – 633. I did not understand. Try saying, "Not shown here, but . . ." What is meant by "GRASP retrievals of only almucantar and TOD data do not demonstrate such difference"? Isn't this what the AERONET green lines in the figure are? Is there something different between "GRASP retrievals of AERONET measurements" and AERONET retrievals as displayed in the figures?

Reply. The clarifications were added to the text:

- lines 710–716: "At the same time, it should be mentioned that to understand better the reason of such significant differences between results of GRASP multi-temporal and AERONET standard retrieval, GRASP inversion of almucantar + TOD only data was performed, as well as MPL/AERONET retrievals for evening and morning data, both with no use of temporal constraints. These retrievals (not shown here for keeping size of the paper reasonable) demonstrated a significantly better agreement in volume size distributions of GRASP almucantar + TOD results with standard AERONET results, allowing one to conclude that changes observed above are caused mainly by the inclusion of lidar observations."

Comment: Line 736. Strongly should be stronger.

Reply: Text was changed to "stronger"

Comment: Lines 912- - 915. I'm still unconvinced.

Reply: The Section 3.1 was updated with extra materials to address the issues. Authors hope that the conclusions are better supported now.

Comment: Line 1063. Figures range should be 24-36, not 23-35.

Reply: Figure numbers were corrected.

Comment: Line 1172 – 1173. Maybe because the multi-instrument retrieval introduces noise because assumptions are not being met.

Reply: The possibility of introducing noise to the retrieval due to the unmet assumptions were studied in the sensitivity study added to section 3.1.1.1. No tendency to worsen the fits of the observations were observed. The following clarifications were added in the text:

- lines 666–669: "For example, as it can be seen from Fig. 3 the retrieval of evening and morning aerosol may be affected negatively if evening, night and morning observations are inverted simultaneously in the case with such unlikely sharp temporal variability of aerosol, when a perturbation in temporal variability of one parameter may manifest in another while ideally fitting the observation data. "

Comment: Figure 38b. how much noise was introduced in the panel shown?

Reply: The values of noise added are introduced in the text:

- line 1287: "the retrievals were performed in noise free conditions and with 5%, 10% and 0.5% of random noises added to the attenuated backscatter profiles, profiles of extinction and backscatter and volume depolarization, respectively"

- line 1292: "with noise added conditions (with the noise levels as mentioned above)."

Comment: Lines 1235 – 1238. I think the authors mean "coarse spherical" not "coarse non- spherical" when they discuss the second most abundant aerosol type.

Reply: the phrase was corrected to "coarse spherical"

---

## Author Comment (AC2) · 28 Jan 2021

**Reply to reviewer 2 (Feng Xu).**

Authors are grateful for the valuable comments that help to improve the paper. In the revised version of manuscript, we tried to address all the questions raised. The detailed responses and clarifications are given below.

Overall comments: The paper by Lopatin et al. employed the GRASP algorithm to investigate the benefits of combining multiple ground-based observations (namely sun-photometric, lidar and radiosonde observations) to constrain aerosol retrievals in terms of their vertical concentration distribution, refractive index, size distribution and spherical particle fraction. With the assumption of temporal continuity of aerosol properties, the synergetic retrieval mitigates the insufficient information content in lidar only night-time retrieval and brings up the retrieval accuracy. The full-length paper covers a large amount of valuable work on algorithm development, test and validation

Reply: We are thankful for overall positive evaluation of our paper.

The point-by-point responses:

Comment 1. Section 3.1.1 discusses the multi-temporal retrieval of combined COBALD, AERONET and MPL observations. Regarding the balloon-borne COBALD data, does the retrieval underlying Figs.1-3 use a series of COBALD temporal measurement during the night of Aug 05, 2015 ? If multiple COBALD measurements are inverted, then it will be helpful to add a time series plot showing the temporal evolution of certain retrieval aerosol quantity (e.g. fine aerosol concentration or other properties). Following the use of multiple COBALD measurement (if this is true), does the "Fine night" and "Coarse night" in Figs. 1-3 averages the retrievals of night measurements from 13:21UTC to 05:16UTC of next day ?

Reply: Only one COBALD profile per night (containing two simultaneous measurements at 455 and 940 nm) was used in the retrievals. The lidar profiles were averaged within ~30 min – the time that is needed for the balloon to reach the top of the profile (~8km). The following clarifications were put to the text:

- lines 610–614: "Thus, an observation set used in multi-temporal AERONET/COBALD/MPL retrievals consists of three diurnal sets including two (evening and morning) co-located observations of AERONET and MPL (see Table 2) and backscatter profiles provided by a single COBALD flight at night. All observations are considered to be instant and observing the aerosol properties averaged within a timeframe not exceeding 30 minutes."

Comment 2. In some figures (e.g. Figs. 25-31), I saw terms "multi-pixel" in legend, but "multi- temporal" in figure caption. To clarify, did the balloon-borne COBALD measurement resolve both multiple pixel and multiple-temporal measurements so that smoothness constraints were imposed in both dimensions ?

Reply: The constraints were applied only on temporal dimension. Term multi-pixel representing both time and spatial measurement discretization, was replaced in the text to avoid the confusion. The following clarifications were added:

- lines 133-143: "This approach uses a priori knowledge of limited time or spatial variability of the parameters retrieved from coordinated but not fully co-incident and/or simultaneous observations. For example, it is used in processing of satellite observations where observations for a large group of different satellite pixels are inverted simultaneously. In this study, it is demonstrated below that this principle can be rather efficient for combining non-coincident but close in time observations, e.g. day- and night- ground-based measurements".

- lines 564–567: "Therefore, a generalized multi-pixel approach realized in GRASP (Dubovik et al., 2011) that allows for applying constraints on variability of aerosol and surface parameters in three dimensions (latitude, longitude and time) is reduced to an application of only multi-temporal constraints in this study, as all provided observations are considered to be spatially co-located, and no spatial variability constraints were used in the retrievals."
- lines 1003–1006: "It was shown that the multi-pixel approach developed within GRASP concept could be efficiently used for combining not fully co-incident but co-located and close in time observations of various origins, for e.g. day- and night measurements, by providing sufficient constraints on aerosol columnar properties

variability to provide additional benefits to the retrievals of the night-time observations that are usually lacking sensitivity to qualitatively retrieve these parameters."

Additionally, descriptions of the constrains applied to aerosol properties during the retrieval were put in Tables 1 and 7.

Comment 3. Table 2: it might be helpful to add another column providing the measurement uncertainties for MPL, COBALD and sun-photometer which are used in retrieval.

Reply: Measurement uncertainties were added into table 2.

Comment 4. In Section 3.1.1, demonstration of retrieval results were provided for refractive index, size distribution and vertical profiles of aerosol concentration. How about the spherical aerosol fraction ? Is it also retrieved retrieved and worthwhile to demonstrate ?

Reply: Spherical fraction is retrieved. However, the studied cases are dominated by desert dust, and do not demonstrate any significant change of this parameter overnight. The changes are below one percent. This is why the retrieved values of sphericity fraction were not illustrated in the figures. Instead, the retrieval results of sphericity fraction were additionally discussed in the text. For better consistency of presentation, the discussions of sphericity fraction retrieval were placed after a size distribution description for each retrieval date both in sections 3.1 and 3.2.

Comment 5. Section 3.2.1 and Section 3.2.2, do the stand-alone COBALD and LILAS retrieval involve the use of any multi-temporal/mullti-pixel constraints ? Please clarify.

Reply: None multi-temporal or multi-pixel constraints were used in stand-alone retrieval. Clarifications were added in the text:

- lines 1014–1017: "Here, unlike a multi-temporal approach described in Section 3.1, only single sets of measurements were inverted. In with these regards the retrieval; could be considered as single pixel, following the terminology used by Dubovik et al. (2011) and above in in Section 3.1"

Additionally, the description of all the constraints applied to retrieved properties were added to Table 1 and 7.

Comment 6. Figs. 25, 26, 27, 29, 30 and 31, it took me a while to confirm the meaning of "components" in the figure legends. For clarify, authors might mention it again in Fig. 25 caption that "components retrieval" here mean "stand-alone COBALD" (or "stand-alone COBALD" retrieval) with a) turning off temporal constraint and b) using pre-determined size distribution for all aerosol types (Table 5), while "multi-pixel retrieval" means a) using all types of instrumental data and b) imposing temporal constraint in retrieval.

Reply: Figure captions and legends were corrected to correspond with the terminology used in the article and to avoid the term "components".

Comment 7. As the authors pointed out, Fig. 29 indicates "some significant differences ... in the lower part of the extinction profiles at 455 nm below 500 m." Could this be due to the impact of measurement uncertainties, or neglecting the multiple scattering in the model, or others ? Is there any constraints imposed on the vertical variations of aerosol concentration or properties ? If so, maybe it's worthwhile to try relaxing the constraint and see if one can observe more consistency in the two types of retrievals.

Reply: The differences observed are coming from different vertical profile cropping, since multi-temporal retrieval requires all three evening/night/morning profiles to be cropped within same altitude range, COBALD data was cropped higher to correspond to MPL profiles provided in the evening and morning, in a stand-alone retrieval a much lower cropping could be applied. The following clarifications were put to the text:
- lines 1199–1200: "coming from a different altitude range of profiles used in both retrievals.".